# KoNODE: Koopman-Driven Neural Ordinary Differential Equations with Evolving Parameters for Time Series Analysis

Hanru Bai [1]   Weiyang Ding [1]

## Abstract

Neural ordinary differential equations (NODEs) have demonstrated strong capabilities in modeling time series. However, existing NODE-based methods often focus solely on the surface-level dynamics derived from observed states, which limits their ability to capture more complex underlying behaviors. To overcome this challenge, we propose KoNODE, a Koopman-driven NODE framework that explicitly models the evolution of ODE parameters over time to encode deep-level information. KoNODE captures the essential yet simple intrinsic linear dynamics that govern the surface dynamics by employing Koopman operators. Our framework operates at three hierarchical levels: the observed state dynamics, the parameter dynamics, and the Koopman linear dynamics, representing the fundamental driving rules of the state dynamics. The proposed approach offers significant improvements in two critical time series tasks: long-term prediction (enabled by the simple linear dynamics) and generalization to new data (driven by the evolving ODE parameters). We validate KoNODE through experiments on synthetic data from complex dynamic systems and real-world datasets, demonstrating its effectiveness in practical scenarios. Our code is available at https://github.com/Baitie00/KoNODE.

## 1. Introduction

Time-series data from various domains are typically sampled from underlying continuous dynamic systems (Voss et al., 2004). Conventional deep learning approaches, such as Recurrent Neural Networks (RNNs) (Rumelhart et al., 1986; Medsker & Jain, 1999), and Long Short-Term Memory networks (LSTMs) (Hochreiter & Schmidhuber, 1997), primarily focus on learning discrete-time transition patterns from these observations. However, these methods do not explicitly model the continuous dynamics that govern the system. Since the available observations are often partial and noisy representations of the true dynamics, overlooking the intrinsic temporal continuity can lead to suboptimal performance in time-series analysis (Abeliuk et al., 2020).

Ordinary Differential Equations (ODEs) provide a continuous framework for modeling time-series data. When the explicit form of the ODEs is unknown and only discrete observations are available, Neural ODEs (NODEs) (Chen et al., 2018), denoted by $\frac{d\boldsymbol{x}(t)}{dt} = \boldsymbol{f}(\boldsymbol{x}(t), t; \boldsymbol{\theta})$, where $\boldsymbol{f}$ is a neural network parameterized by $\boldsymbol{\theta}$, offer a data-driven alternative to learn the continuous dynamics. Various NODE-based methods, such as HBNODEs (Xia et al., 2021) and MoNODEs (Auzina et al., 2024), have been proposed to improve efficiency and predictive performance. However, most existing NODE-based approaches assume a static parameter $\boldsymbol{\theta}$, limiting their adapting ability to constantly changing environments. To overcome this limitation, ANODEV2 (Zhang et al., 2019) introduced time-evolving parameters, enabling the model to adapt to different temporal evolution patterns and enhance its generalization ability. However, ANODEV2 primarily focused on the improvement of model generalization by evolving parameters, without fully exploring their potential to model more complex dynamical behaviors.

In contrast, we argue that $\boldsymbol{\theta}(t)$, which directly drives the state dynamics, inherently encodes the underlying dynamics of the evolution (namely ***deep-level*** information). By explicitly modeling the evolution of $\boldsymbol{\theta}(t)$, we not only enhance the model's generalization ability but also capture the intrinsic dynamics beneath the surface-level behaviors, offering a more comprehensive understanding of the complex dynamic system. This perspective is further supported by the following observations. In systems like robotic arms, internal physical properties such as frictional torque fundamentally shape the system's evolution over time. As time progresses, factors like joint lubrication changes and frictional wear cause gradual variations in the frictional torque, potentially leading to trajectory deviations, motion delays, and other dynamic effects. These changes, in turn, modify

[1]Institute of Science and Technology for Brain-Inspired Intelligence, Fudan University, Shanghai, China. Correspondence to: Weiyang Ding <dingwy@fudan.edu.cn>.

*Proceedings of the 42^{nd} International Conference on Machine Learning*, Vancouver, Canada. PMLR 267, 2025. Copyright 2025 by the author(s).

the system's underlying dynamical behavior and influence its overall performance and adaptability. Thus, this point of view expands the NODE models beyond the application of data-driven state predictors, positioning them as a framework for uncovering the fundamental principles that govern system evolution.

Based on this insight, we further observe that the most fundamental evolution of the system is governed by simple yet essential ***intrinsic linear dynamics***. This is inspired by observations in robotic arms, where complex robotic trajectories are inherently driven by linear dynamics (see App. B.1 for the theoretical analysis). This suggests that although $\boldsymbol{\theta}$, the neural network parameters, may exhibit nonlinear dynamical behavior, its true underlying dynamics can be fundamentally linear. In other words, the apparent nonlinearity in parameter dynamics arises from observation limitations, which is consistent with the principles of the Koopman theory (Kurdila & Bobade, 2018). The Koopman operators transform complex nonlinear dynamics into linear representations by expanding the system dimensions. To this end, we use the Koopman operators as the dominating parameters for the dynamics of $\boldsymbol{\theta}$, capturing the intrinsic linear space and revealing dynamics beneath the surface. In summary, we conceptualize complex dynamics as a three-level hierarchical structure as illustrated in Fig. 1: the trajectory from the ***observed state dynamics***, influenced by the ***parameter dynamics***, and ultimately governed by the underlying ***Koopman linear dynamics***.

Specifically, we introduce KoNODE, a Koopman-driven NODE framework that captures the intrinsic linear dynamics by applying the Koopman model to the time-evolving ODE parameters. Our framework enhances two key tasks in time series analysis: *generalization to new data*, by explicitly modeling parameter evolution, and *long-term prediction*, which is guaranteed by effectively learning global structures through capturing the underlying simple linear dynamics. Notably, our approach diverges from ANODEV2 (Zhang et al., 2019) in several ways. While ANODEV2 treats parameter dynamics as an ODE and uses Green's functions for time convolution, primarily focusing on image data, we model parameter dynamics as nonlinear observations of intrinsic linear dynamics that uncover the deeper driving rules of the system, providing a richer understanding of the dynamic systems beneath time series data. Our main contributions are as follows:

(1) We introduce a novel approach to encoding deep-level, time-dependent information from $\boldsymbol{\theta}(t)$, transforming NODEs from simple data-driven state predictors into a powerful framework that enables uncovering the fundamental principles driving system evolution.

(2) Applying Koopman theory on $\boldsymbol{\theta}(t)$, we capture the intrinsic linear dynamics that underlie complex systems. We

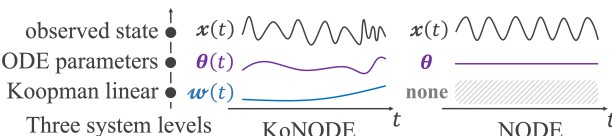

*Figure 1.* The roadmap for the proposed framework and its comparison to NODE.

thus propose a three-level hierarchical structure—spanning the observed state dynamics, ODE parameter evolution, and Koopman linear dynamics. The Koopman operators, which govern the deepest dynamics, could help identify the system's dominant driving modes and frequencies and offer insights into stability, periodicity, or other inherent characteristics, thus gaining a mathematically interpretable understanding of the system's evolution and governing laws.

(3) Our framework excels in two critical time series tasks: long-term prediction and generalization to new data. Experiments on two synthetic datasets from complex dynamic systems demonstrated its superior performance. Additionally, we evaluated the framework's applicability in real-world scenarios through robot motion prediction across three datasets and multivariate forecasting on six datasets.

## 2. Preliminaries

### 2.1. Neural ODEs

Neural ODEs (NODEs) (Chen et al., 2018) extend discrete traditional neural network architectures by modeling hidden states as continuous trajectories governed by an ODE

$$\frac{d\boldsymbol{x}(t)}{dt} = \boldsymbol{f}(\boldsymbol{x}(t), t; \boldsymbol{\theta}), \tag{1}$$

where $\boldsymbol{f}$ is a neural network parameterized by $\boldsymbol{\theta}$, and $\boldsymbol{x}(t) \in \mathbb{R}^n$ is the system state at $t \in [t_0, t_1]$. Given the initial state $\boldsymbol{x}(t_0)$, the solution at any time $t \in (t_0, t_1]$ is obtained by integrating the differential equation: $\boldsymbol{x}(t) = \boldsymbol{x}(t_0) + \int_{t_0}^{t} \boldsymbol{f}(\boldsymbol{x}(t), t; \boldsymbol{\theta}) \, dt$, rather than propagating activations through discrete layers as in recurrent or deep networks. Neural ODEs employ numerical solvers, such as the Euler method (Gear & Petzold, 1984), Runge-Kutta methods (Runge, 1895; Kutta, 1901), or solvers with adaptive step size such as Dormand-Prince (Prince & Dormand, 1981), to calculate this integration efficiently.

Since the trajectory of $\boldsymbol{x}(t)$ is governed by a neural network parameterized by $\boldsymbol{\theta}$, learning the system dynamics requires optimizing these parameters. This is formulated as an ODE-constrained optimization problem:

$$\min_{\boldsymbol{\theta}} \mathcal{L}(\boldsymbol{x}), \text{ s.t. } \frac{d\boldsymbol{x}}{dt} = \boldsymbol{f}(\boldsymbol{x}(t), t; \boldsymbol{\theta}), \ \boldsymbol{x}(t_0) = \boldsymbol{x}_0, \tag{2}$$

where $\mathcal{L}(\boldsymbol{x}) \triangleq \frac{1}{|T|} \sum_{t \in T} \mathcal{E}[\boldsymbol{x}(t)]$ is the average loss function over the sampled time points and $\mathcal{E}$ is the error term at

a specific time that quantifies the discrepancy between the predicted and the actual system states.

To optimize the parameters, the adjoint method (Chen et al., 2018) is commonly used to compute $\nabla_{\boldsymbol{\theta}}\mathcal{L}$ in NODE formulation. It efficiently computes gradients by solving a backward ODE, avoiding storing intermediate states during the forward pass. This memory-efficient approach makes NODEs particularly attractive for long-time-series modeling. Please refer to App. A.3 for the details about the adjoint method utilized in NODEs. Related work about Neural ODEs can be found in App. A.1.

## 2.2. Koopman Operator

The Koopman theory is a powerful tool for nonlinear dynamical system analysis. It acts on the observable functions of the system state, which evolve linearly, rather than the nonlinear ones in the state space itself (Koopman, 1931).

**Definition 2.1** (Koopman Operator, (Mezić, 2005)). For a finite-dimensional state space $\mathcal{X} \subseteq \mathbb{R}^n$ in which the state evolves according to $\boldsymbol{x}_{t+\Delta t} = \Phi(\boldsymbol{x}_t), t \in [t_0, t_1]$ where $\Phi : \mathcal{X} \to \mathcal{X}$ is the operator from the state to the next time. The Koopman operator $\mathcal{K}$ is a linear operator that reveals such evolution by acting on an infinite-dimensional function space of observable functions $g : \mathbb{R}^n \to \mathbb{R}$ such that

$$(\mathcal{K}g)(\boldsymbol{x}) = g(\Phi(\boldsymbol{x})).$$

In numerical practice, the infinite-dimensional Koopman space is commonly approximated by modeling the observable function space through a finite number of bases $\boldsymbol{u} = [u_1, u_2, \cdots, u_m]^\top$ (Brunton et al., 2021). The Koopman operators are estimated by $\boldsymbol{u}(\Phi(\boldsymbol{x})) = [\mathcal{K}u_1, \mathcal{K}u_2, \cdots, \mathcal{K}u_m]^\top \approx \boldsymbol{K}\boldsymbol{u}(\boldsymbol{x})$ where $\boldsymbol{K} \in \mathbb{R}^{m \times m}$ is an approximated Koopman matrix. Hence, for an observable function $g$, let $\boldsymbol{\xi} \triangleq [\langle g, u_1 \rangle, \langle g, u_2 \rangle, \cdots, \langle g, u_m \rangle]^\top$ then $g(\Phi(\boldsymbol{x})) \approx \boldsymbol{\xi}^\top \boldsymbol{K}\boldsymbol{u}(\boldsymbol{x})$ for the $m$-dimensional subspace. Specifically, for a vector of linearly uncorrelated observable functions $\boldsymbol{g} \approx \boldsymbol{P}\boldsymbol{u}$ and $\boldsymbol{P}$ is a nonsingular matrix, we have $\boldsymbol{g}(\Phi(\boldsymbol{x})) \approx \boldsymbol{P}\boldsymbol{K}\boldsymbol{P}^{-1}\boldsymbol{g}(\boldsymbol{x})$ for the evolution of the observables. Related work about Koopman Operators can be found in App. A.1.

## 3. KoNODE Framework

In this section, we present the theoretical framework of KoNODE. We begin by formalizing the proposed model (Sec. 3.1) into an optimization problem and solving it using the Lagrangian approach. The succeeding adjoint method is used to perform backpropagation (Sec. 3.2). Finally, we provide theoretical results to present both the guidance for choosing the Koopman space dimension and the analysis of the error introduced by approximating the Koopman operators (Sec. 3.3).

### 3.1. Problem Formulation

Our framework, KoNODE, is proposed within the NODE setups. As outlined in the introduction, the core idea of KoNODE is to uncover the simple intrinsic linear dynamics underlying the complex observed systems. We denote the observed system as $\frac{d\boldsymbol{x}(t)}{dt} = \boldsymbol{f}(\boldsymbol{x}(t), t; \boldsymbol{\theta}(t))$, where $\boldsymbol{x}(t) \in \mathbb{R}^n$ is the trajectory of the system, and $\boldsymbol{\theta}(t)$ are the parameters of the neural network $\boldsymbol{f}$. We let the intrinsic linear dynamics be represented as $\boldsymbol{w}(t)$. They reveal the underlying driving rules of $\boldsymbol{x}(t)$. Next, we proceed with the detailed formulation of KoNODE.

We start by refining the structure of KoNODE, which is built on three hierarchical levels:

(1) **The observed dynamics**: The trajectory $\boldsymbol{x}(t)$ evolves according to $\frac{d\boldsymbol{x}(t)}{dt} = \boldsymbol{f}(\boldsymbol{x}(t), t; \boldsymbol{\theta}(t))$, where $\boldsymbol{\theta}(t)$ encodes the deep-level driving rules of the observed dynamic.

(2) **The parameter dynamics**: The parameter $\boldsymbol{\theta}(t)$, modeled by Koopman operators, evolves as an observation of the trajectory $\boldsymbol{w}(t)$ of an intrinsic linear system.

(3) **The Koopman linear dynamics**: At the deepest level, the dynamics of $\boldsymbol{w}(t)$, which underlies $\boldsymbol{\theta}(t)$, is simplified to a linear form, i.e., $\frac{d\boldsymbol{w}(t)}{dt} = \mathcal{A}\boldsymbol{w}(t)$, where $\mathcal{A}$ is a linear operator representing the essential structure of the system. In Koopman Theory, $\boldsymbol{w}(t)$ is inherently defined in an infinite-dimensional space. For practical implementation, we approximate $\boldsymbol{w}(t)$ in a finite-dimensional subspace, assuming $\boldsymbol{w}(t) \in \mathbb{R}^m$. Under this approximation, the dynamics of $\boldsymbol{w}(t)$ become $\frac{d\boldsymbol{w}(t)}{dt} = \boldsymbol{A}\boldsymbol{w}(t)$, where $\boldsymbol{A} \in \mathbb{R}^{m \times m}$ is a finite-dimensional matrix. Thm. 3.2 in Sec. 3.3 provides theoretical bounds for this approximation to confirm its validity.

Together, these three levels form the basic model of our KoNODE framework, which is expressed as the following formulation:

$$\min_{\boldsymbol{A}, \boldsymbol{w}_0, \boldsymbol{\psi}} \quad \mathcal{L}(\boldsymbol{x}) \tag{3a}$$

$$\text{s.t.} \quad \frac{d\boldsymbol{x}(t)}{dt} = \boldsymbol{f}(\boldsymbol{x}(t), t; \boldsymbol{\theta}(t)), \boldsymbol{x}(t_0) = \boldsymbol{x}_0, \tag{3b}$$

$$\boldsymbol{\theta}(t) = \boldsymbol{h}(\boldsymbol{w}(t); \boldsymbol{\psi}), \tag{3c}$$

$$\frac{d\boldsymbol{w}(t)}{dt} = \boldsymbol{A}\boldsymbol{w}(t), \boldsymbol{w}(t_0) = \boldsymbol{w}_0, t \in [t_0, t_1]. \tag{3d}$$

Here, $\boldsymbol{h} : \mathbb{R}^m \to \boldsymbol{\Theta}$ is a neural network parameterized by $\boldsymbol{\psi}$ to estimate the inverse mapping of the observable function vector $\boldsymbol{g}$, where $\boldsymbol{\Theta}$ is the parameter space with $\boldsymbol{\theta}(t)$ in. We model $\boldsymbol{h}$ using a neural network due to its flexibility.

Next, we give further modeling of $\boldsymbol{A}$. Numerically, we perform an eigen-decomposition on $\boldsymbol{A}$, i.e., $\boldsymbol{A} = \boldsymbol{P}\boldsymbol{D}\boldsymbol{P}^{-1}$ to a product of a non-singular matrix $\boldsymbol{P}$ and a block diagonal matrix $\boldsymbol{D}$ in a format of $\boldsymbol{D} = \text{diag}\{\lambda_1, \lambda_2, \cdots, \lambda_p, \boldsymbol{\Lambda}_1, \boldsymbol{\Lambda}_2, \cdots, \boldsymbol{\Lambda}_q\}$, where $p + 2q =$

$m$ and $\{\lambda_i\}_{i=1}^p$ are real eigenvalues of $\boldsymbol{A}$ and the matrix blocks $\boldsymbol{\Lambda}_i = \begin{bmatrix} \alpha_i & \beta_i \\ -\beta_i & \alpha_i \end{bmatrix}$, $i = 1, \cdots, q$ are real matrices that represent the imaginary eigenvalues of $\boldsymbol{A}$. To simplify the problem, we unify the treatment of all eigenvalues by allowing redundancy in the dimensions of $\boldsymbol{w}(t)$. Specifically, we assume that all eigenvalues are conjugate complex eigenvalues, that is, $\boldsymbol{D} = \text{diag}\left\{\boldsymbol{\Lambda}_1, \boldsymbol{\Lambda}_2, \cdots, \boldsymbol{\Lambda}_{m/2}\right\}$, where $m$ is even. Let $\tilde{\boldsymbol{g}}(\boldsymbol{\theta}) \triangleq \boldsymbol{P} \cdot \boldsymbol{g}(\boldsymbol{\theta})$ be a vector of alternative observable functions, the resulting trajectory $\boldsymbol{w}(t)$ also follows a linear dynamic driven by another approximated Koopman operator $\tilde{\boldsymbol{K}} = \mathrm{e}^{\Delta t \cdot \boldsymbol{D}}$. Hence, without loss of generality, we assume $\boldsymbol{A} \equiv \boldsymbol{D}$ using the observable function $\tilde{\boldsymbol{g}}$. See App. B.2 for the rigorous theoretical analysis of the above formulation of matrix $\boldsymbol{A}$.

## 3.2. Adjoint Model for the Optimization Problem

In this part, we apply the Lagrangian method to solve the constrained optimization problem defined in Eq. (3). Subsequently, the Karush-Kuhn-Tucker (KKT) conditions obtained from the Lagrangian function in Eq. (4) imply the restrictions in the gradient descent method. The resulting adjoint method was used to perform backpropagation.

### 3.2.1. LAGRANGIAN FUNCTION

The Lagrangian function of Eq. (3) is

$$
\begin{aligned}
\mathfrak{L} =&\, \mathcal{L}(\boldsymbol{x}) - \boldsymbol{\alpha}_0^\top \left( \boldsymbol{x}(t_0) - \boldsymbol{x}_0 \right) - \boldsymbol{\beta}_0^\top \left( \boldsymbol{w}(t_0) - \boldsymbol{w}_0 \right) \\
&- \int_{t_0}^{t_1} \boldsymbol{\alpha}(t)^\top \cdot \left( \frac{d\boldsymbol{x}(t)}{dt} - \boldsymbol{f}(\boldsymbol{x}(t), t; \boldsymbol{\theta}(t)) \right) dt \\
&- \int_{t_0}^{t_1} \boldsymbol{\beta}(t)^\top \cdot \left( \frac{d\boldsymbol{w}(t)}{dt} - \boldsymbol{A}\boldsymbol{w}(t) \right) dt \\
&- \int_{t_0}^{t_1} \boldsymbol{\gamma}(t)^\top \cdot \left( \boldsymbol{\theta}(t) - \boldsymbol{h}(\boldsymbol{w}(t); \boldsymbol{\psi}) \right) dt,
\end{aligned}
\tag{4}
$$

where $\boldsymbol{\alpha}(t)$, $\boldsymbol{\beta}(t)$, and $\boldsymbol{\gamma}(t)$ are the corresponding adjoint variables for the constraints in Eq. (3). We give the KKT conditions of Eq. (4) below.

### 3.2.2. OPTIMALITY CONDITIONS

The KKT conditions consist of (1) the given conditions in (3b), (3c) and (3d); (2) the gradients of $\mathfrak{L}$ regarding $\boldsymbol{A}$, $\boldsymbol{\psi}$, and $\boldsymbol{w}_0$ equaling to zeros; and (3) the variants of $\mathfrak{L}$ concerning $\boldsymbol{x}(t)$, $\boldsymbol{\theta}(t)$, $\boldsymbol{w}(t)$ being zeros, which result in relations for the three adjoints $\boldsymbol{\alpha}(t)$, $\boldsymbol{\beta}(t)$, $\boldsymbol{\gamma}(t)$. Specifically, taking the variants of $\mathfrak{L}$ with respect to $\boldsymbol{x}(t)$, $\boldsymbol{\theta}(t)$ and $\boldsymbol{w}(t)$ yield Eq. (5), Eq. (6), and Eq. (7) as follows (where the argument $t$ is omitted for simplicity),

$$
\frac{d\boldsymbol{\alpha}}{dt} + \frac{\partial \boldsymbol{f}(\boldsymbol{x}, t; \boldsymbol{\theta})}{\partial \boldsymbol{x}}^\top \boldsymbol{\alpha} + \nabla_t \mathcal{L} = \boldsymbol{0}, \nabla_{t_1} \mathcal{L} - \boldsymbol{\alpha}(t_1) = \boldsymbol{0}, \tag{5}
$$

$$
\frac{\partial \boldsymbol{f}(\boldsymbol{x}, t; \boldsymbol{\theta})}{\partial \boldsymbol{\theta}}^\top \boldsymbol{\alpha} - \boldsymbol{\gamma} = 0, \tag{6}
$$

$$
\frac{d\boldsymbol{\beta}}{dt} + \boldsymbol{A}^\top \boldsymbol{\beta} + \frac{\partial \boldsymbol{h}(\boldsymbol{w}; \boldsymbol{\psi})}{\partial \boldsymbol{w}}^\top \boldsymbol{\gamma} = 0, \boldsymbol{\beta}(t_1) = \boldsymbol{0}, \tag{7}
$$

where $\nabla_t \mathcal{L} \triangleq \frac{\partial}{\partial \boldsymbol{x}(t)} \mathcal{L}(\{\boldsymbol{x}(t)\}_{t \in T})$ is the direct partial derivative of $\mathcal{L}$. Among them, Eq. (5) is a backward-in-time ODE for the adjoint function $\boldsymbol{\alpha}(t)$, Eq. (6) reveals the algebraic relation between $\boldsymbol{\alpha}(t)$ and $\boldsymbol{\gamma}(t)$, and Eq. (7) is the backward-in-time ODE for the adjoint function $\boldsymbol{\beta}(t)$ with our unknown parameters $\boldsymbol{A}$ and $\boldsymbol{\psi}$.

The gradients of $\mathfrak{L}$ regarding $\boldsymbol{\psi}$, $\boldsymbol{A}$, and $\boldsymbol{w}_0$ are,

$$
\frac{\partial \mathfrak{L}}{\partial \boldsymbol{\psi}} = \int_{t_0}^{t_1} \frac{\partial \boldsymbol{h}(\boldsymbol{w}; \boldsymbol{\psi})^\top}{\partial \boldsymbol{\psi}} \cdot \frac{\partial \boldsymbol{f}(\boldsymbol{x}, t; \boldsymbol{\theta})^\top}{\partial \boldsymbol{\theta}} \cdot \boldsymbol{\alpha} \, dt, \tag{8}
$$

$$
\frac{\partial \mathfrak{L}}{\partial \boldsymbol{A}} = \int_{t_0}^{t_1} \boldsymbol{\beta} \cdot \boldsymbol{w}^\top \, dt, \quad \frac{\partial \mathfrak{L}}{\partial \boldsymbol{w}_0} = \boldsymbol{\beta}_0. \tag{9}
$$

Note that if optimality conditions are achieved with respect to $\boldsymbol{\psi}$, $\boldsymbol{A}$, and $\boldsymbol{w}_0$, then $\frac{\partial \mathfrak{L}}{\partial \boldsymbol{\psi}} = \boldsymbol{0}$, $\frac{\partial \mathfrak{L}}{\partial \boldsymbol{A}} = \boldsymbol{0}$, $\frac{\partial \mathfrak{L}}{\partial \boldsymbol{w}_0} = \boldsymbol{0}$. The detailed derivation of conditions is provided in App. B.3.

However, the unknown parameters $\boldsymbol{A}$ only appear in Eq. (7), which represents an ODE for the adjoint function $\boldsymbol{\beta}(t)$, making them unable to solve directly through the KKT conditions. Consequently, we turn to gradient-based optimization approaches to iteratively update these parameters.

**Theorem 3.1** (Equivalence of Gradient and Adjoint Trajectories)**.** *If the partial Jacobian matrices are Lipschitz continuous. The gradient trajectories of $\boldsymbol{x}(t)$ and $\boldsymbol{w}(t)$ are $\boldsymbol{\alpha}(t)$ and $\boldsymbol{\beta}(t)$ respectively, i.e., $\frac{\partial \mathcal{L}}{\partial \boldsymbol{x}(t)} = \boldsymbol{\alpha}(t)$, $\frac{\partial \mathcal{L}}{\partial \boldsymbol{w}(t)} = \boldsymbol{\beta}(t)$.*

Thm. 3.1 proves that the adjoint trajectories $\boldsymbol{\alpha}(t)$ and $\boldsymbol{\beta}(t)$ are actually continuous equivalents of the gradient trajectory during backpropagation through the KoNODE model. Consequently, the adjoint trajectory substitutes the backpropagation process for network training. This conclusion hence aligns with the adjoint method in NODE-based methods. Therefore, we utilize the adjoint method to compute the derivatives of the loss function $\mathcal{L}$ with respect to $\boldsymbol{\psi}$, $\boldsymbol{A}$, and $\boldsymbol{w}_0$ to perform optimization. The overall adjoint model is summarized in App. B.4, while the algorithm is detailed in Alg. 1, App. B.4.

By leveraging Koopman operators, our framework can use spectral tools for system analysis. Fundamental evolution principles are revealed by the spectrum of the Koopman operators, which correspond element-wise to matrix $\boldsymbol{A}$ and are given by $\lambda_j(\boldsymbol{K}) = e^{\alpha_j \Delta t}(\cos \beta_j \Delta t \pm \mathrm{i} \sin \beta_j \Delta t)$ in our framework. Specifically, the real part of the eigenvalues indicates the evolution speed, while the imaginary part corresponds to intrinsic frequencies for the periodicity. Analyzing the elements of $\boldsymbol{A}$ helps identify the system's dominant driving modes and frequencies, thus offering insights into stability and periodicity, respectively. Furthermore, KoNODE enables possible physics-informed constraints over $\boldsymbol{\theta}$ via regularization on $\boldsymbol{A}$, $\boldsymbol{\psi}$, or $\boldsymbol{w}_0$. We derive the adjoint results for stable systems by constraining $\boldsymbol{A}$'s eigenvalues (Mezic & Banaszuk, 2000). See App. B.5 for details on these constraints and the corresponding adjoint results.

### 3.3. Theoritical Results

This section presents a theoretical analysis of the error caused by the proposed framework. Sec. 3.3.1 aims at guiding the choice of dimension $m$, and Sec. 3.3.2 proves that the additional Koopman module causes only minor error compared to conventional ODE models, apart from the potential improvement due to the advanced fitting ability.

#### 3.3.1. THE FINITE-RANK APPROXIMATION OF THE KOOPMAN MODEL

We first provide the theoretical upper bound of the approximation error of the Koopman model regarding $\boldsymbol{w}(t)$ with dimension $m$ under the guidance of Kurdila & Bobade (2018). $g(\boldsymbol{\theta}) = \boldsymbol{\xi}^\top \boldsymbol{u}(\boldsymbol{\theta})$ is the vector of observable functions and $\boldsymbol{K} \triangleq e^{\boldsymbol{A}} \in \mathbb{R}^{m \times m}$ is the order-$m$ Koopman matrix.

**Theorem 3.2** (Regression Error for the Order-$m$ Koopman Operator). *The estimation of the Koopman space by $m$-dimensional observables has a relative error of,*

$$\int_{\boldsymbol{\Theta}} \left[ \boldsymbol{\xi}^\top \boldsymbol{K} \boldsymbol{u}(\boldsymbol{\theta}) - \mathcal{K} g(\boldsymbol{\theta}) \right]^2 d\boldsymbol{\theta} \Big/ \int_{\boldsymbol{\Theta}} [\mathcal{K} g(\boldsymbol{\theta})]^2 d\boldsymbol{\theta} \lesssim m^{-r}$$

*where $\boldsymbol{\xi} \triangleq \begin{bmatrix} \langle g, u_1 \rangle & \langle g, u_2 \rangle & \cdots & \langle g, u_m \rangle \end{bmatrix}$ and $r \triangleq \frac{1}{2} \left[ -\max_{m < i \leq N} \log_i \langle g, u_i \rangle - 1 \right]$ where $N$ is the number of data and $\{u_i\}_{i=1}^\infty$ are the bases of true Koopman space.*

The conclusion ensures that the relative error of the regression results is inversely bounded by the approximated dimension of the Koopman space.

**Proposition 3.3.** *If the data trajectories are in a space of rank $s$ and function $f$ satisfies the Lipschitz condition and is differentiable, we have $\lceil \frac{D}{D-s} \rceil \leq m \leq s$.*

If we take the data rank into account, we can further provide the possible range of the Koopman dimension $m$. Prop. 3.3 provides the result, as a reference for the choice of hidden dimension. The ideal dimension in practice can be various due to (1) the auxiliary dimension design of the Koopman matrix, (2) the multiple choice of dynamics for $\boldsymbol{\theta}(t)$, and (3) more accurate dynamic regression by multiple frontier sets. In subsequent sections, we further address the error caused by the proposed method with respect to the chosen $m$.

#### 3.3.2. THE ESTIMATION ACCURACY OF THE DATA-DRIVEN METHOD

Noted that the error bound regarding $\boldsymbol{w}(t)$ is better controlled by the number of data $N$, we further refer to the influence of data amount in the accuracy of estimation. In the data-driven scenario, we provide bounds for the errors in the estimation of the Koopman operator and Koopman regression method (Bevanda et al., 2024) regarding $N$ randomly drawn data. If the dataset was drawn independently and identically, the theorem for the error bound of the approximated Koopman operator, as a corollary of Thm. 12, Nüske et al. (2023) claims as follows.

**Theorem 3.4** (Probabilistic Error for Data-Driven Koopman Operator). *For $N$ data drawn independently and identically from a Hilbert space at a single time point and any probabilistic tolerance $\delta \in (0,1)$ we have $\mathbb{P}(\|\boldsymbol{K} - \hat{\boldsymbol{K}}_N\|_F \leq \varepsilon) \geq 1 - \delta$, s.t. $\varepsilon = 2\sqrt{3}\, \sigma m^2 \|\boldsymbol{K}\| / (\min\{1, \|\boldsymbol{K}\|\} \cdot \sqrt{\delta N} - \sqrt{3}\, \sigma m^2)$ where $\boldsymbol{K} \triangleq e^{\boldsymbol{A}}$ is the theoretical Koopman matrix, $\hat{\boldsymbol{K}}_N$ is the matrix estimated by $N$ data and $\sigma$ is the isotropic standard deviation in the space of $\boldsymbol{w}(t)$.*

The result of Thm. 3.4 shows that the error bound $\varepsilon \sim \mathcal{O}(1/\sqrt{N})$ as $N \to \infty$. Thm. C.15 in App. C.4 shows that the regression error is also controlled by $\mathcal{O}(1/\sqrt{N})$ which ensures the feasibility of the batch loss. Furthermore, Thms. 3.2 and 3.4 together reveal the theoretical error bound of the proposed model in real-world data, as shown in Thm. 3.5.

**Theorem 3.5.** *For $N$ data drawn independently and identically from a Hilbert space at a single time point and any observable function $g$ and its corresponding $\boldsymbol{\xi}$, with any probabilistic tolerance $\delta \in (0,1)$ we have*

$$\mathbb{P}\left( \frac{|\boldsymbol{\xi}^\top \hat{\boldsymbol{K}}_N \boldsymbol{w}(t) - \mathcal{K} g(\boldsymbol{\theta}(t))|}{\|\boldsymbol{\xi}\| \cdot \|\boldsymbol{w}(t)\|} \leq \frac{2r(\mathcal{K})}{p \cdot \min\{1, r(\mathcal{K})\} - 1} + \varepsilon \right) \geq 1 - \delta,$$

*where $\hat{\boldsymbol{K}}_N$ is the order-$m$ Koopman matrix estimated by the $N$ data, $p = \sqrt{N(\delta - 2m^{-\frac{r}{3}})}/(\sqrt{3}\, \sigma m^2)$ for the data standard deviation $\sigma$, $\varepsilon = o(m^{-\frac{r}{3}})$, and $r(\mathcal{K})$ is the spectral radius of operator $\mathcal{K}$.*

The presented error provides guidance for the choice of dimension $m$, and a proof for the minority of errors caused by the additional Koopman module compared to conventional ODE models, apart from the potential improvement due to the advanced fitting ability. Thm. C.22, App. C.6 further presents the conclusion by comparing the proposed and ODE models. The proofs of all theorems can be found in App. C.

## 4. Practical Implementation

This section outlines the practical implementation of our framework, which consists of three levels marked as $\boldsymbol{x}, \boldsymbol{\theta}$, and $\boldsymbol{w}$ in Fig. 2. The forward propagation in Fig. 2 requires the given input states $\{\boldsymbol{x}(t)\}_{t \in T_{\text{input}}}$ shown by the red line in Fig. 2 and the initial Koopman observables obtained by random initialization (if given initial point) or IKOE (if given sequence). Then we deduce the trajectory $\boldsymbol{w}_t$ by the trainable matrix $\boldsymbol{A}$. The observables $\boldsymbol{w}(t)$ are then passed through the neural network $\boldsymbol{h}$ to obtain the parameters $\boldsymbol{\theta}(t)$. Subsequently, we utilize the ODE solver to predict $\boldsymbol{x}(t)$ at any time after $T_{\text{input}}$ shown by the green dashed line in Fig. 2. Using the true trajectory and the predicted trajectory, we can calculate the loss. For the backward propagation, we use the adjoint model results in Sec. 3.2 to update the trainable parameters, including $\boldsymbol{\psi}$ in $\boldsymbol{h}$, $\boldsymbol{A}$, and network parameters

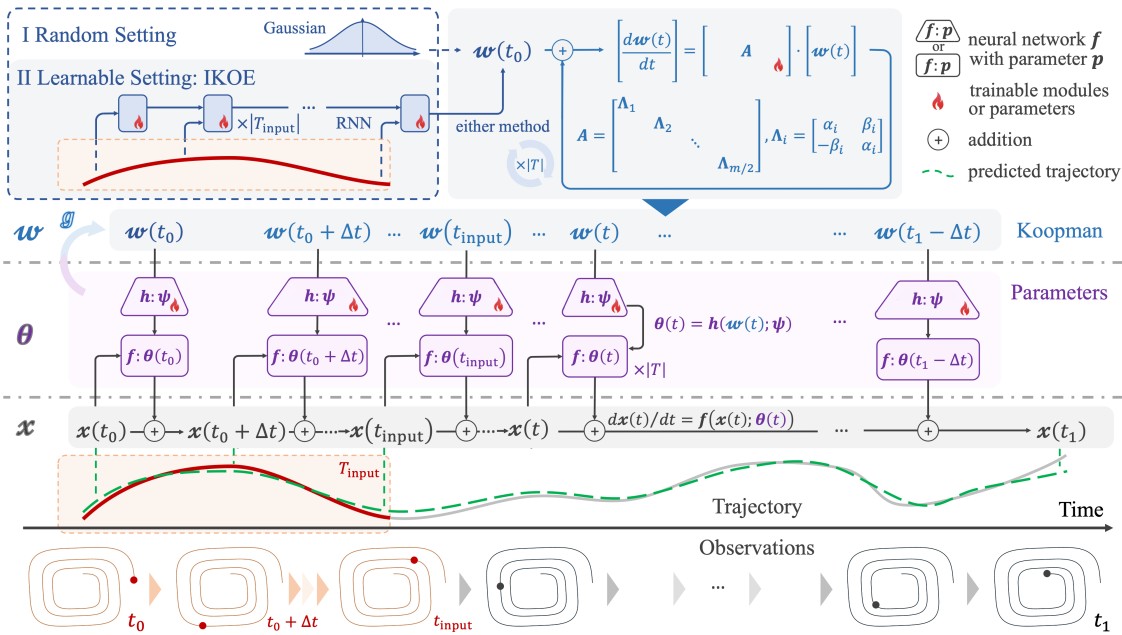

*Figure 2.* **Our proposed framework of system estimation.** The overall structure is separated into three levels, i.e., the Koopman linear dynamics, the parameter dynamics, and the observed state dynamics, which directly leads to observations. The initial Koopman observable $\boldsymbol{w}(t_0)$ serves as an encoding of the system over $N$ time points which is either obtained randomly for simple systems (method I) or predicted by the Initial Koopman Observable Estimator (IKOE) module from the first $N_{\text{input}}$ time points (method II).

in IKOE. Our framework is trained in an end-to-end manner, allowing for simultaneous optimization of all the levels in the model.

We now discuss the design of $\boldsymbol{w}_0$. $\boldsymbol{w}_0$ is an encoding of a specific system as it identifies the trajectory of $\boldsymbol{w}(t)$ under a determined matrix $\boldsymbol{A}$ and thus influences the estimated ODE. There are two settings for the initial observables. (I) **Learnable setting** when given a sequence as input: We estimate $\boldsymbol{w}_0$ in a data-driven fashion using the Initial Koopman Observable Estimator (IKOE) $\Xi$, which is based on an RNN architecture (Hochreiter & Schmidhuber, 1997; Paszke et al., 2017), i.e., $\boldsymbol{w}_0 = \Xi[\{\boldsymbol{x}(t)\}_{t \in T_{\text{input}}}]$. (II) **Random setting** when given initial point as input: We draw $\boldsymbol{w}_0 \sim \mathcal{N}(\boldsymbol{0}, \sigma^2 \boldsymbol{I})$ from a centered normal distribution. In the latter part of the article, we experimentally demonstrate that when the initial point is provided as input, the random initialization setting remains effective.

# 5. Experiments

We set up the experiments to answer the following problems regarding our proposed framework:

(1) Global Structure Representation: Does our method effectively capture the global structure, thereby ensuring robust long-term prediction performance? (Sec. 5.1)

(2) Validation of Major Capability: Does our method im-

prove the ability to model complex systems, particularly in terms of long-term prediction and generalization (Sec. 5.2), and what empirical insights into the governing rules of system evolution does our framework reveal through spectral analysis? (Sec. 5.3)

(3) Real-World Applications: How does the model perform in real-world applications? (Sec. 5.4 and Sec. 5.5)

(4) Ablation Study: What is the contribution of each component in the framework, specifically regarding parameter evolution and Koopman operators (App. D.1.1)?

## 5.1. Spiral Curve Fitting: Validating the Global Representation Capacity of KoNODE

In this experiment, we adapted the ***single trajectory*** fitting task proposed in Chen et al. (2018) to validate the global representation capacity of KoNODE. We added a time dependency to the matrix $A$ using sines and cosines to produce the new time-dependent matrix $\boldsymbol{A}(t) = \begin{bmatrix} -0.1 + 0.5\sin(t) & 2.0 + \cos(t) \\ -2.0 + 0.5\cos(2t) & -0.1 - 0.5\sin(t) \end{bmatrix}$. The trajectory data were then generated from ODE $\left[\frac{dx}{dt}, \frac{dy}{dt}\right]^{\top} = \boldsymbol{A}(t) \left[x^3, y^3\right]^{\top}, [x_0, y_0]^{\top} = [2, 0]^{\top}$. We extend the matrix $A$ from Chen et al. (2018) by making it time-dependent, using sine and cosine functions to define its temporal variation. The results of the KoNODE framework were compared with

*Table 1.* **Test MSEs for the Oscillator and Robot Motion datasets in Secs. 5.2 and 5.4.** The experiments for the Oscillator dataset include the Non-linear Oscillator and Damped Oscillator Family. The experiments for the robot motion dataset include Imitation C, Imitation cube pick, and Imitation S. Note that the MSE values for the Damped Oscillator Family are rescaled by a factor of $10^2$. Results are presented as mean±std across 5 runs with different seeds. The best-performing results are highlighted in **bold**, while the second-best results are underlined.

| | (SEC. 5.2) OSCILLATOR | | (SEC. 5.4) ROBOT MOTION | | |
| --- | --- | --- | --- | --- | --- |
| METHOD | NON-LINEAR OSCILLATOR | DAMPED OSCILLATOR FAMILY ($\times 10^{-2}$) | IMITATION C | IMITATION CUBE PICK | IMITATION S |
| LATENT ODE | 0.711±0.098 | 12.522±1.533 | 0.134±0.028 | 0.156±0.047 | 0.204±0.075 |
| NEURAL PROCESS | 0.460±0.013 | 13.844±0.007 | 0.094±0.004 | 0.087±0.006 | 0.142±0.062 |
| NODE | 0.497±0.039 | 14.906±0.204 | 0.136±0.056 | 0.084±0.035 | 0.171±0.055 |
| HYPERNET | 0.381±0.104 | 2.252±1.505 | 0.031±0.007 | 0.041±0.009 | 0.063±0.023 |
| ANODEV2 | 0.419±0.109 | 0.895±0.126 | 0.046±0.008 | 0.071±0.058 | 0.035±0.045 |
| HBNODE | 0.447±0.019 | 0.912±0.004 | 0.021±0.003 | 0.030±0.012 | 0.015±0.009 |
| LEADS | 0.120±0.049 | 0.907±0.073 | 0.050±0.010 | 0.060±0.017 | 0.018±0.002 |
| CoDA | 0.123±0.017 | 0.914±0.036 | 0.039±0.004 | 0.069±0.016 | 0.010±0.000 |
| OINNs | 0.558±0.026 | 23.512±6.428 | 0.022±0.010 | 0.042±0.017 | 0.016±0.015 |
| MoNODE | 0.449±0.039 | 0.936±0.019 | 0.166±0.052 | 0.039±0.022 | 0.127±0.049 |
| **KoNODE** | **0.085±0.031** | **0.810±0.009** | **0.017±0.003** | **0.022±0.001** | **0.005±0.002** |

the standard Neural ODE approach (Chen et al., 2018). App. D.2 details the implementation, architecture, and hyperparameters in the experiment.

In Fig. 3, KoNODE effectively fits the dynamic systems, whereas NODE fails due to the complexity of the dynamics. The results demonstrate that the proposed framework effectively captures the underlying global structure of complex dynamics by leveraging the linear dynamics modeled through the Koopman operator. The NODE model, only focusing on local non-linearities, struggles to represent this global pattern. The presented superiority suggests that KoNODE allows for improved long-term prediction on complex trajectories.

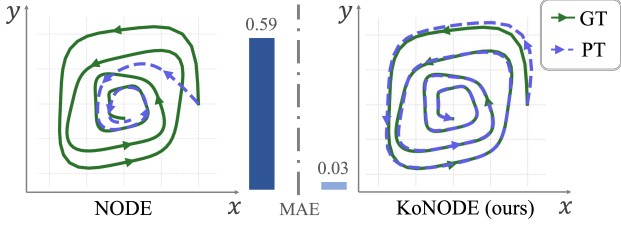

*Figure 3.* **The test MAE and the phase portrait of the test trajectory obtained from NODE and KoNODE, respectively.** The green lines represent the ground truth (GT) trajectories and the blue dashed lines represent the predicted trajectories (PT).

## 5.2. Dynamics Systems Learning: Validating Benefits for Long-Term Prediction and Generalization

In this section, we conducted two experiments on different dynamic systems: (I) *System modeling*, where all trajectories share the same differential equation parameterization

to evaluate the model generalization from one to another (Sec. 5.2.1); (II) *System family modeling*, where trajectories have different parameterizations to assess generalization to unseen systems (Sec. 5.2.2). As the training data covers a shorter period than the test data, both experiments simultaneously evaluate the long-term prediction capability.

**Dynamics systems.** The two chosen dynamic systems are both governed by the Duffing oscillator (Kovacic & Brennan, 2011), a generalized framework for oscillatory systems. The Duffing equation is $\frac{d\mathbf{q}}{dt} = \mathbf{p}$, $\frac{d\mathbf{p}}{dt} = -\alpha\mathbf{q} - \beta\mathbf{q}^3 - \gamma\mathbf{p}$, where $\alpha, \beta$, and $\gamma$ are scalar parameters that determine the linear stiffness, non-linear stiffness, and damping, respectively. The characteristics of the Duffing oscillator can be changed by adjusting parameters. Complete implementation details on the training setup, model architecture, and hyperparameters can be found in App. D.3.3 and App. D.3.4.

**Comparison methods.** We compared our framework with the following models: (1) baselines: **vanilla NODE** (Chen et al., 2018), **Latent ODE** (ODE enc) (Rubanova et al., 2019), and **Neural Process** (Garnelo et al., 2018), (2) **Hypernet** (Ha et al., 2016), (3) **ANODEV2** (Zhang et al., 2019), (4) **HBNODE** (Xia et al., 2021), (5) **LEADS** (Yin et al., 2021), (6) **CoDA** (Kirchmeyer et al., 2022), (7) **OINNs** (Zhi et al., 2022), and (8) **MoNODE** (Auzina et al., 2024). Among these methods, the ODE parameters in Hypernet and ANODEV2 vary over time. The other methods are achieved by their pubic codes. Refer to App. D.3.1 for details of these comparison methods.

**Performance metric.** Following prior work, such as Auzina et al. (2024), we evaluated model performance using the mean squared error (MSE) between the predicted phase space trajectories and the ground truth from the test set.

*Table 2.* **Test MSEs for long-term prediction.** Results are presented as mean±std across 5 runs with different seeds. The best-performing results are highlighted in **bold**, while the second-best results are underlined.

| METHOD | CUP | WEATHER | ETTH1 | ETTH2 | ETTM1 | ETTM2 |
|---|---|---|---|---|---|---|
| NODE | 0.993±0.008 | 0.553±0.017 | 1.064±0.067 | 0.597±0.035 | 1.084±0.036 | 0.545±0.023 |
| HBNODE | 0.879±0.069 | 0.513±0.031 | 0.905±0.039 | 0.592±0.041 | 0.914±0.017 | 0.493±0.068 |
| DEEPVAR | 0.962±0.000 | 0.561±0.011 | 0.947±0.223 | 0.592±0.032 | 0.905±0.017 | 0.398±0.056 |
| AUTOFORMER | 1.163±0.005 | 0.449±0.016 | 0.868±0.034 | 0.375±0.003 | 1.071±0.028 | 0.227±0.002 |
| PATCHTST | 1.066±0.000 | 0.421±0.001 | 0.722±0.007 | 0.366±0.000 | 1.008±0.002 | 0.238±0.000 |
| KOOPA | 1.139±0.007 | 0.427±0.004 | **0.605±0.009** | 0.357±0.002 | 0.759±0.007 | 0.229±0.001 |
| SST | 1.085±0.001 | 0.417±0.000 | 0.759±0.018 | 0.370±0.001 | 0.960±0.002 | 0.239±0.000 |
| **KoNODE** | **0.864±0.009** | **0.386±0.009** | 0.678±0.005 | **0.289±0.001** | **0.664±0.015** | **0.211±0.007** |

### 5.2.1. SYSTEM MODELING: NON-LINEAR OSCILLATOR

We first performed system modeling on the nonlinear oscillator system, setting $\alpha = -1, \beta = 1, \gamma = 0$, to test the ability of long-term prediction and generalization to new trajectories. The training set includes 600 trajectories with 30 time steps and was perturbed by Gaussian noise, while the test set consists of 200 trajectories with 300 time steps from the same system. See App. D.3.3 for more details about data generation. Note that in this part, $\boldsymbol{w}_0$ is randomly initialized to evaluate the effectiveness of our random initialization strategy.

The results in the Non-linear oscillator column, Tab. 1 show that our framework improves the test accuracy across all comparison methods. Notably, Hypernet and ANODEV2, which allow time-evolving parameters, perform better than baseline methods, highlighting the importance of modeling parameters evolving over time for complex systems. However, they still lag behind ours, highlighting the advantage of using the Koopman model for parameter dynamics, which thereby enhances long-term prediction accuracy. We showed the predicted results of our method in Fig. 4 (a).

**$\boldsymbol{w}$ dimension.** We also presented the test MSE across different $\boldsymbol{w}$ dimensions in Fig. 4 (c). Fig. 4 (c) shows the corresponding test MSE when $\boldsymbol{w}$ dimension increases from 10 to 50. Fig. 4 (c) shows a curve dropping in $[2, 10]$ and rising if $m > 10$ with a tolerable relative perturbation. Experimentally, more weights are introduced in networks, which may add extra uncertainty and instability during training when $m$ is large. Theoretically, the error bound in Thm. 3.5 is majorly dominated by two terms, one with a factor of $\frac{m^2}{\sqrt{N}}$ in the coefficient $p$ and the other with the order of $m^{-\frac{r}{3}}$, which does indicate the instability when $m$ is too large and may infer overfits. Additionally, we reported the results of our method using other ODE solvers in Tab. 4, App. D.1.2.

**Time and memory analysis.** We analyze the running time and memory assumption in this experiment. During the experiment, KoNODE converges in 109.19 seconds using 10 epochs, while NODE spends 215.50 seconds using 48

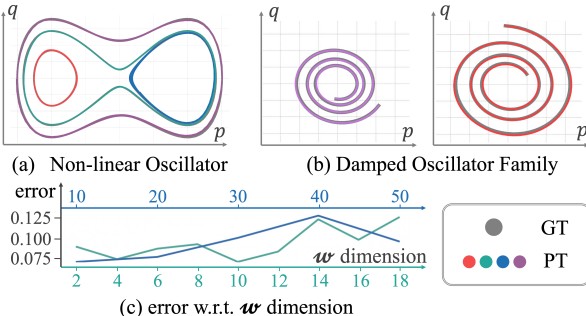

(a) Non-linear Oscillator    (b) Damped Oscillator Family

(c) error w.r.t. $\boldsymbol{w}$ dimension

*Figure 4.* **Visual results of KoNODE predictions.** (a) Non-linear oscillator trajectories, where different initial conditions produce four distinct patterns, shown in different colors. (b) Damped oscillator family, where different parameters result in varying trajectories, with predictions shown in purple and red. Gray lines represent ground truth (GT) trajectories in (a) and (b). (c) Test MSE across different $\boldsymbol{w}$ dimensions, where the blue polygonal line corresponds to the blue scale at the top, and the green line corresponds to the green scale at the bottom.

epochs. This concludes that the proposed method converges faster, though it spends longer for each iteration. Regarding the memory assumption, the proposed method uses $O(\max\{n, m, h\} \cdot D)$ while NODE uses $O(nD)$ where $n$, $m$, $h$ are the dimensions of $x(t), \theta(t)$, and the hidden layer. $D$ is the model size for the differential function $f$. The theoretical conclusion indicates that the memory assumption of the proposed method is comparable to the ODE-based models, which are superior to other methods. In practice, the auxiliary memory assumption would further narrow the gap between them. See Tab. 5, App. D.1.2 for more results.

### 5.2.2. SYSTEM FAMILY MODELING: DAMPED OSCILLATOR

In this section, we focus on system family modeling on the damped oscillator family, setting $\alpha \in [0.9, 1.1], \gamma \in [0.99, 1.01], \beta = 0$, to test the ability of long-term prediction and generalization to unseen systems. See App. D.3.4 for details on data generation and implementation. The results in the Damped oscillator family column, Tab. 1, and

Fig. 4 (b) demonstrate the effectiveness of our framework in generalizing to new systems.

## 5.3. Koopman Spectral Analysis: Showing How to Uncover the System's Evolution Laws Empirically

In this section, we provide a more detailed analysis and interpretation of the learned dynamics in Sec. 5.2 by visualizing the spectrum of the approximated Koopman operator for the Oscillator dataset. The results are shown in Fig. 5. First, the magnitudes of the spectrum are approximately one for two dynamic systems, indicating boundary stability, where the state remains on a stable trajectory without diverging or converging. Second, the dominant spectrum of the nonlinear oscillator (on the left) has similar imaginary parts (i.e., the frequency components are the same), indicating that the system exhibits clear periodicity. Modes with the same imaginary part have the same periodic components.

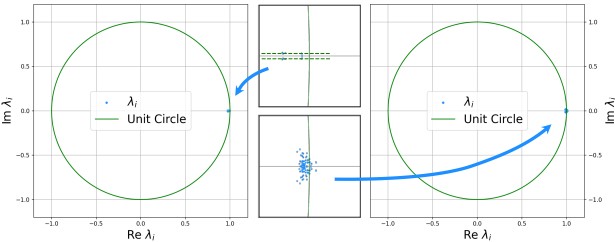

*Figure 5.* **Spectral Analysis of the Approximated Koopman Operator for Oscillatory Systems.** The plots illustrate the eigenvalue spectra of the approximated Koopman operator for two classes of oscillators: the nonlinear oscillator (left) and the damped oscillator family (right). The real and imaginary parts of the eigenvalues are plotted along the horizontal and vertical axes, respectively. The green unit circle serves as a reference for spectral stability, where eigenvalues inside the circle indicate dissipative behavior. Blue dots represent the computed spectrum from the approximation of our framework.

## 5.4. Application on Robot Motion Prediction: Validating Generalization in Real-World Scenarios

We evaluated our framework's generalization in real-world scenarios through robot motion prediction on three datasets. Unlike synthetic data from precise dynamics, real-world data comes from unstable systems in complex environments. The experiment demonstrates the method's ability to generalize from training to unseen test data. We compared the results with the same models as in Sec. 5.2, with implementation details in App. D.4.2.

**Data setup.** The trajectory datasets were obtained under three different robot tasks proposed by Khansari-Zadeh & Billard (2011), i.e., drawing "S" shapes (Imitation S), placing a cube on a shelf (Imitation cube pick), and drawing out large "C" shapes (Imitation C). See App. D.4.1 for more

details about data description and generation.

The results in the Robot Motion column of Tab. 1 highlight our method's superiority in accuracy and generalization for real-world applications. This demonstrates its effectiveness in modeling complex robot dynamics, with potential for robotic control and trajectory prediction. Additionally, HBNODE, which captures high-order information, and OINNs, which enhance linearity, ranked second and third, showcasing their ability to extract underlying dynamics from real-world data.

## 5.5. Application on Multivariate Forecasting: Validating Long-term Prediction Ability in Real-world Scenarios

In this experiment, we evaluated the long-term prediction capability of our framework on six real-world datasets: CUP, WEATHER, ETTH1, ETTH2, ETTM1, and ETTM2. We followed the preprocessing methods of datasets outlined in Liu et al. (2023) and split the datasets into training, validation, and test sets with a ratio of $7 : 1 : 2$. For further details on the datasets and implementation, please refer to App. D.5.1 and App. D.5.2 respectively.

**Comparison methods.** In this experiment, we compared the proposed method with (1) NODE-based methods: **NODE** and **HBNODE** (performs well in Sec. 5.4); (2) Statistical-based method: **DeepVAR** (Salinas et al., 2019); (3) Transformer-based methods: **Autoformer** (Wu et al., 2021) and **PatchTST** (Nie et al., 2022); (4) Koopman-based method: **Koopa** (Liu et al., 2023); and (5) SSM-based method: **SST** (Xu et al., 2024). See App. D.5.3 for more details about these comparison methods.

As shown in Tab. 2, KoNODE achieved the best performance on five out of six datasets, demonstrating its strong capability in long-term prediction in real-world applications. While Koopa performed best on ETTh1, KoNODE remained competitive. Both methods leveraged Koopman operators to capture global system dynamics, making them particularly effective for datasets with periodic or quasi-periodic structures such as ETT datasets.

**The ablation study** can be found in App. D.1.1 whose results confirm the effectiveness of parameter evolvement and the Koopman modeling.

## 6. Conclusions and Future Work

We propose a Koopman-driven hierarchical NODE framework, with theoretical and experimental results demonstrating its ability to provide a deep understanding of time series. In the future, we will explore accelerating the KoNODE algorithm to improve the overall modeling efficiency and adapting the model for irregularly sampled data.

# Acknowledgements

This work was partially supported by the National Natural Science Foundation of China (No. 12471481, U24A2001), the Science and Technology Commission of Shanghai Municipality (No. 23ZR1403000), and the Open Foundation of Key Laboratory Advanced Manufacturing for Optical Systems, CAS (No. KLMSKF202403).

# Impact Statement

This paper presents work whose goal is to advance the field of Machine Learning. There are many potential societal consequences of our work, particularly in domains that rely on accurate and interpretable time-series forecasting. By enhancing the ability to model continuous dynamics through Koopman-driven Neural ODEs, our approach has implications in fields such as robotics, environmental sciences, and so on. By improving prediction accuracy and interpretability, our framework enhances decision-making in these critical areas.

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

## Appendix Overview

The following lists the structure of the appendix, with links to the respective sections.

# A. Background Knowledge Supplement

## A.1. Related Work

### A.1.1. NEURAL ODEs

Neural ODEs (Chen et al., 2018) have been widely studied for modeling continuous-time dynamics, with various extensions improving their expressiveness and adaptability. Since their introduction, various NODE-based methods have been extensively developed to enhance efficiency, stability, and predictive performance in modeling continuous-time dynamics. To address stability and long-term prediction, HBNODEs (Xia et al., 2021) use second-order heavy ball dynamics, mitigating numerical instabilities and improving trajectory accuracy over extended time horizons. Modulated Neural ODEs (MoNODEs) (Auzina et al., 2024) enhance predictive adaptability by decoupling dynamic and static factors. ODEs via Invertible Neural Networks (Zhi et al., 2022) transform the dynamics to promote linearity, improving learning speed and robustness. However, the aforementioned methods assume that the ODE parameters remain static, which limits the generalization ability of the NODE framework. ANODEV2 (Zhang et al., 2019) was the first to propose modeling parameters to evolve over time and introduced a coupled neural ODE framework. Subsequently, ODEtoODE (Choromanski et al., 2020) proposed a nested NODE framework, where the parameter weights are constrained to be orthogonal, primarily focusing on the stability benefits of this approach. Our framework continues to focus on modeling the parameter evolution to encode the deep-level information that reveals the driving rules of the systems.

### A.1.2. KOOPMAN OPERATORS

Over the past two decades, Koopman operator theory has gained significant attention, facilitating advancements in various domains, including system analysis (Lusch et al., 2018; Azencot et al., 2019), control (Narasingam et al., 2023), optimization (Redman et al., 2021), and forecasting (Liu et al., 2023; Zheng et al.). These approaches leverage Koopman-based representations to extract linearizable global structures from nonlinear dynamics, enhancing interpretability and control.

Our method builds upon Koopman-driven frameworks, capturing underlying linear dynamical properties. Unlike traditional Koopman approaches that rely on predefined observables, our framework uses neural networks to improve its flexibility.

### A.1.3. NON-AUTONOMOUS DYNAMICAL SYSTEMS AND TIME-VARYING MODELS

In addition to Neural ODE-based paradigms, which have provided powerful tools for modeling continuous-time dynamics, another well-established line of work addresses non-stationarity by explicitly modeling time-dependent parameters. This approach is closely related to the theory of non-autonomous dynamical systems, where the evolution rules themselves vary over time (Carvalho et al., 2015). Such models have been applied to diverse domains, including climate modeling, neuroscience, and economics, where the assumption of stationarity is often violated. More recently, in machine learning and statistics, various time-varying parameter models have been proposed, such as time-varying autoregressive models (Haslbeck et al., 2021) and time-varying state-space models (Tong et al., 2023). In deep learning, architectures like recurrent neural networks with dynamic weights (Krauss et al., 2019). In addition, there has been a growing body of recent work exploring hierarchical latent variable models with time-varying structure, particularly in the context of learning representations that generalize across domains and tasks, such as Guo et al. (2018), Brenner et al. (2024). These approaches illustrate the importance of incorporating temporal variability directly into the model structure, aligning with our approach.

## A.2. Notation Convention For Gradients

We first define the notations for the derivatives in the appendices as follows.

**Definition A.1** (Notation Convention For Gradients). We denote the loss gradient w.r.t. a parameter (or activation) tensor $x$ as $\frac{d\mathcal{L}}{d\boldsymbol{x}}$ which has the same shape, i.e., $\left[\frac{d\mathcal{L}}{d\boldsymbol{x}}\right]_i = \frac{\partial \mathcal{L}}{\partial \boldsymbol{x}_i}$ for any index $i$ in the index set.

**Definition A.2** (Notation Convention For The Jacobian Matrix). We denote the Jacobian matrix of a vector $\boldsymbol{y} \in \mathbb{R}^n$ w.r.t. another $\boldsymbol{x} \in \mathbb{R}^m$ as $\frac{d\boldsymbol{y}}{d\boldsymbol{x}} \in \mathbb{R}^{n \times m}$.

The differentiations in the above conventions can be replaced by partial derivatives in certain occasions where the response is also considered a function of other fixed parameters.

Note that under conventions A.1 and A.2, a scalar-to-vector gradient w.r.t. $\boldsymbol{x} \in \mathbb{R}^m$ results in a vector $\frac{d\mathcal{L}}{d\boldsymbol{x}} \in \mathbb{R}^m$ while the vector-to-vector Jacobian matrix is $\frac{d\mathcal{L}}{d\boldsymbol{x}} \in \mathbb{R}^{n \times m}$ even for 1-dimensional response ($n = 1$).

### A.3. Additional Preliminaries on Neural ODEs: Ajoint Method

For the adjoint method of the Neural ODE structure, an auxiliary variable, $\boldsymbol{\lambda}(t) = \frac{\partial \mathcal{L}}{\partial \boldsymbol{x}(t)}$, is introduced to help compute $\frac{\partial \mathcal{L}}{\partial \boldsymbol{\theta}}$. The evolution of $\boldsymbol{\lambda}(t)$ is governed by

$$\frac{d\boldsymbol{\lambda}(t)}{dt} = -\frac{\partial \boldsymbol{f}(\boldsymbol{x}(t), t; \boldsymbol{\theta})}{\partial \boldsymbol{x}(t)}^{\top} \boldsymbol{\lambda}(t), \tag{10}$$

with the terminal condition

$$\boldsymbol{\lambda}(t_1) = \frac{\partial \mathcal{L}}{\partial \boldsymbol{x}(t_1)}. \tag{11}$$

Once the adjoint variable $\boldsymbol{\lambda}(t)$ is computed, the gradient of the loss with respect to $\boldsymbol{\theta}$ can be efficiently obtained by

$$\frac{\partial \mathcal{L}}{\partial \boldsymbol{\theta}} = \int_{t_0}^{t_1} \frac{\partial \boldsymbol{f}(\boldsymbol{x}(t), t; \boldsymbol{\theta})}{\partial \boldsymbol{\theta}}^{\top} \boldsymbol{\lambda}(t) dt. \tag{12}$$

This approach avoids storing intermediate states, reducing memory requirements and enabling efficient training of Neural ODEs in large-scale applications.

## B. Framework Supplement

### B.1. The Inspiration of KoNODE

If we refer to a simple mechanism in robotics that consists of two nodes, A and B, connected by a rod where both nodes have motors to control the mechanism. Assume the motor at node A is fastened to the basement. Let $\boldsymbol{x}_*$, $\boldsymbol{n}_*$ and $\theta_*$ be the location, the axis of rotation, and the phase of rotation of the motor at the node $*$ respectively. If the two motors rotate at angular velocities $\omega_A$ and $\omega_B$, we have the evolution formula,

$$
\begin{bmatrix} \boldsymbol{x}_A \\ \boldsymbol{n}_A \\ \theta_A \\ \boldsymbol{x}_B \\ \boldsymbol{n}_B \\ \theta_B \end{bmatrix}(t + \Delta t) =
\begin{bmatrix} \boldsymbol{I} & & & & & \\ & \boldsymbol{I} & & & & \\ & & 1 & & & \\ & & & \boldsymbol{R}(\Delta t) & & \\ & & & & \boldsymbol{R}(\Delta t) & \\ & & & & & 1 \end{bmatrix}
\begin{bmatrix} \boldsymbol{x}_A \\ \boldsymbol{n}_A \\ \theta_A \\ \boldsymbol{x}_B \\ \boldsymbol{n}_B \\ \theta_B \end{bmatrix}(t) +
\begin{bmatrix} \boldsymbol{0} \\ \boldsymbol{0} \\ \omega_A \Delta t \\ \boldsymbol{x}_A - \boldsymbol{R}(\Delta t)\boldsymbol{x}_A \\ \boldsymbol{0} \\ \omega_B \Delta t \end{bmatrix}, \tag{13}
$$

where $\boldsymbol{R}(\Delta t)$ is the rotation matrix caused by motor $A$ which is,

$$
\boldsymbol{R}(\Delta t) = \boldsymbol{P}_A \begin{bmatrix} \cos \omega_A \Delta t & \sin \omega_A \Delta t & 0 \\ -\sin \omega_A \Delta t & \cos \omega_A \Delta t & 0 \\ 0 & 0 & 1 \end{bmatrix} \boldsymbol{P}_A^{\top} \approx \boldsymbol{P}_A \begin{bmatrix} 1 & \omega_A & 0 \\ -\omega_A & 1 & 0 \\ 0 & 0 & 1 \end{bmatrix} \boldsymbol{P}_A^{\top} \Delta t, \tag{14}
$$

where $\boldsymbol{P}_A$ is a unit orthogonal matrix with the last column being $\boldsymbol{n}_A$. Consequently,

$$
\frac{d}{dt} \begin{bmatrix} \boldsymbol{x}_A \\ \boldsymbol{n}_A \\ \theta_A \\ \boldsymbol{x}_B \\ \boldsymbol{n}_B \\ \theta_B \end{bmatrix} = \boldsymbol{P}
\begin{bmatrix} \boldsymbol{I} & & & & & & & \\ & \boldsymbol{I} & & & & & & \\ & & 1 & & & & & \\ & & & 1 & \omega_A & & & \\ & & & -\omega_A & 1 & & & \\ & & & & & 1 & & \\ & & & & & & 1 & \omega_A \\ & & & & & & -\omega_A & 1 \\ & & & & & & & & 1 \\ & & & & & & & & & 1 \end{bmatrix}
\boldsymbol{P}^{\top} \begin{bmatrix} \boldsymbol{x}_A \\ \boldsymbol{n}_A \\ \theta_A \\ \boldsymbol{x}_B \\ \boldsymbol{n}_B \\ \theta_B \end{bmatrix} +
\begin{bmatrix} \boldsymbol{0} \\ \boldsymbol{0} \\ \omega_A \\ \boldsymbol{W}\boldsymbol{x}_A \\ \boldsymbol{0} \\ \omega_B \end{bmatrix}, \tag{15}
$$

where $\boldsymbol{P} = \text{diag}\{\boldsymbol{I}, \boldsymbol{I}, 1, \boldsymbol{P}_A, \boldsymbol{P}_A, 1\}$ and $\boldsymbol{W} = \begin{bmatrix} 0 & -\omega_A & 0 \\ \omega_A & 1 & 0 \\ 0 & 0 & 0 \end{bmatrix}$.

## B.2. Analysis for the Formulation of Matrix $A$

We formulate the real matrix $A \in \mathbb{R}^{n \times n}$ as $A = \lim\limits_{\varepsilon \to 0^+} P_\varepsilon D_\varepsilon P_\varepsilon^{-1} - \varepsilon \Sigma$ where $P_\varepsilon D_\varepsilon P_\varepsilon^{-1}$ is the real eigen decomposition of the matrix $A + \varepsilon \Sigma$ to the product of a non-singular matrix $P_\varepsilon$ and a block diagonal matrix $D_\varepsilon$ in a format of,

$$D_\varepsilon = \mathrm{diag}\left\{\lambda_1, \lambda_2, \cdots, \lambda_p, \Lambda_1, \Lambda_2, \cdots, \Lambda_q\right\},$$

where $p + 2q = n$; the matrix blocks $\Lambda_i = \begin{bmatrix} \alpha_i & \beta_i \\ -\beta_i & \alpha_i \end{bmatrix}$, $i = 1, 2, \cdots, q$ are real matrices in replace of the conjugated imaginary eigenvalues. The perturbation $\varepsilon \Sigma$ is a diagonal matrix that ensures the uniqueness of the eigenvalues of the matrix $A + \varepsilon \Sigma$.

Note that we can select the perturbation as $\Sigma = \mathrm{diag}\left\{1, 2, \cdots, n\right\}$ so that the matrix $A + \varepsilon \Sigma$ has no eigenvalue whose order is greater than 1 when $\varepsilon < \dfrac{1}{n} \min\limits_{1 \le i, j \le n, \rho_i \ne \rho_j} |\rho_i - \rho_j|$ for the $n$ eigenvalues $\{\rho_i\}_{i=1}^n$ for matrix $A$.

Under this formulation, we assume $D = \lim_{\varepsilon \to 0^+} D_\varepsilon$ and define the parameter vector for the matrix $\mathcal{A}$ as $a \triangleq [\lambda_1, \lambda_2, \cdots, \lambda_p, \alpha_1, \cdots, \alpha_q, \beta_1, \cdots, \beta_q]^\top$.

Note that the eigenvalues of $\Lambda_i$ are $\alpha_i \pm \beta_i \cdot \mathrm{i}$, thus $\Lambda_i = Q \, \mathrm{diag}\left\{\alpha_i + \beta_i \cdot \mathrm{i}, \alpha_i - \beta_i \cdot \mathrm{i}\right\} Q^{-1}$. As a result, $\mathrm{e}^{\Lambda_i} = \mathrm{e}^{\alpha_i} Q Q^{-1} = \mathrm{e}^{\alpha_i} I$ which leads to,

$$K = \mathrm{e}^{A\Delta t} = \lim_{\varepsilon \to 0^+} P_\varepsilon \mathrm{e}^{\Delta t \cdot D_\varepsilon} P_\varepsilon^{-1} - \Gamma_\varepsilon(A, \Sigma, \Delta t) \overset{\star}{=} \lim_{\varepsilon \to 0^+} P_\varepsilon \, \mathrm{e}^{\Delta t \cdot D} P_\varepsilon^{-1},$$

where $\mathrm{e}^{\Delta t \cdot D} = \mathrm{diag}\left\{\mathrm{e}^{\lambda_1 \Delta t}, \cdots, \mathrm{e}^{\lambda_p \Delta t}, \mathrm{e}^{\alpha_1 \Delta t} I_2, \cdots, \mathrm{e}^{\alpha_q \Delta t} I_2\right\}$ and $\Gamma_\varepsilon \sim o(1), \varepsilon \to 0^+$. The equality $\overset{\star}{=}$ holds as the limit $K$ is known to exist.

Let $\tilde{g}(\theta) \triangleq \tilde{P} \cdot g(\theta)$ be an alternative observable function, the resulting sequence $w(t)$ follows a linear dynamic with an finite-order operator $\tilde{K} = \mathrm{e}^{\Delta t \cdot D}$. Here, $\tilde{P}$ is an approximation of $\lim_{\varepsilon \to 0^+} P_\varepsilon^{-1}$. Hence, without loss of generality, we assume $A \equiv D$. That is to say, in our formulation, the matrix $A$ takes the following form:

$$A \triangleq \begin{bmatrix} \alpha_1 & \beta_1 & & & & & & \\ -\beta_1 & \alpha_1 & & & & & & \\ & & \alpha_2 & \beta_2 & & & & \\ & & -\beta_2 & \alpha_2 & & & & \\ & & & & \ddots & & & \\ & & & & & \alpha_{m/2} & \beta_{m/2} \\ & & & & & -\beta_{m/2} & \alpha_{m/2} \end{bmatrix}.$$

## B.3. The Derivation of Optimality Conditions

To start with, we define the one-order variations as a tool so that applying it to the Lagrangian function shown in Eq. 4, manuscript results in the KKT conditions.

**Definition B.1** (One-order variations). Consider a functional $\mathfrak{L}(x) : \mathcal{F} \to \mathbb{R}$ where $\mathcal{F}$ is the set for functions. For a function $x(t)$, denoted as $x$ and its variation $\delta x$, the one-order variation $\delta \mathfrak{L}(x)$ is

$$\delta \mathfrak{L}(x) = \frac{\partial \mathfrak{L}}{\partial x}^\top \delta x,$$

due to the Taylor's expansion $\mathfrak{L}(x + \delta x) = \mathfrak{L}(x) + \frac{\partial \mathfrak{L}}{\partial x}^\top \delta x + O(||\delta x||^2)$.

Some arguments of the functions are omitted for simplicity under the assumption of being time $t$ when they are not specified.

**Variations with respect to $x(t)$**

In order to satisfy the first optimality condition on $z$, we have $\delta \mathfrak{L}(x) = \frac{\partial \mathfrak{L}}{\partial x}^\top \delta x = 0$ for any variation $\delta x$ in space and time,

hence

$$\frac{\partial \mathfrak{L}}{\partial \boldsymbol{x}}^\top \delta \boldsymbol{x}$$

$$= \sum_{t \in T} \nabla_t^\top \mathcal{L} \, \delta \boldsymbol{x}(t) - \delta \left[ \int_{t_0}^{t_1} \boldsymbol{\alpha}^\top \, d\boldsymbol{x} \right] + \int_{t_0}^{t_1} \left( \frac{\partial \boldsymbol{f}(\boldsymbol{x}, t; \boldsymbol{\theta})}{\partial \boldsymbol{x}}^\top \boldsymbol{\alpha} \right)^\top \delta \boldsymbol{x} \, dt - \boldsymbol{\alpha}_0^\top \, \delta \boldsymbol{x}(t_0),$$

$$= \sum_{t \in T} \nabla_t^\top \mathcal{L} \, \delta \boldsymbol{x}(t) - \delta \left[ \boldsymbol{\alpha}(t_1)^\top \boldsymbol{x}(t_1) - \boldsymbol{\alpha}(t_0)^\top \boldsymbol{x}(t_0) - \int_{t_0}^{t_1} \boldsymbol{x}^\top \, d\boldsymbol{\alpha} \right] + \int_{t_0}^{t_1} \left( \frac{\partial \boldsymbol{f}(\boldsymbol{x}, t; \boldsymbol{\theta})}{\partial \boldsymbol{x}}^\top \boldsymbol{\alpha} \right)^\top \delta \boldsymbol{x} \, dt - \boldsymbol{\alpha}_0^\top \, \delta \boldsymbol{x}(t_0),$$

$$= \sum_{\substack{t \in T \\ t \neq t_1}} \nabla_t^\top \mathcal{L} \, \delta \boldsymbol{x}(t) + \int_{t_0}^{t_1} \left[ \frac{d\boldsymbol{\alpha}}{dt} + \frac{\partial \boldsymbol{f}(\boldsymbol{x}, t; \boldsymbol{\theta})}{\partial \boldsymbol{x}}^\top \boldsymbol{\alpha} \right]^\top \delta \boldsymbol{x} \, dt + [\nabla_{t_1} \mathcal{L} - \boldsymbol{\alpha}(t_1)]^\top \delta \boldsymbol{x}(t_1) + [\boldsymbol{\alpha}(t_0) - \boldsymbol{\alpha}_0]^\top \delta \boldsymbol{x}(t_0),$$

$$= \int_{t_0}^{t_1} \left[ \frac{d\boldsymbol{\alpha}}{dt} + \frac{\partial \boldsymbol{f}(\boldsymbol{x}, t; \boldsymbol{\theta})}{\partial \boldsymbol{x}}^\top \boldsymbol{\alpha} + \nabla_t \mathcal{L} \right]^\top \delta \boldsymbol{x} \, dt + [\nabla_{t_1} \mathcal{L} - \boldsymbol{\alpha}(t_1)]^\top \delta \boldsymbol{x}(t_1) + [\boldsymbol{\alpha}(t_0) - \boldsymbol{\alpha}_0]^\top \delta \boldsymbol{x}(t_0) = \mathbf{0},$$

holds for any variations $\delta \boldsymbol{x}$. Therefore, the first set of adjoint equations are as follows,

$$\frac{d\boldsymbol{\alpha}}{dt} + \frac{\partial \boldsymbol{f}(\boldsymbol{x}, t; \boldsymbol{\theta})}{\partial \boldsymbol{x}}^\top \boldsymbol{\alpha} + \nabla_t \mathcal{L} = \mathbf{0}, \nabla_{t_1} \mathcal{L} - \boldsymbol{\alpha}(t_1) = 0. \tag{16}$$

Note that simultaneously, we have $\boldsymbol{\alpha}_0 = \boldsymbol{\alpha}(t_0)$ for the solution of $\boldsymbol{\alpha}_0$.

**Variations with respect to $\boldsymbol{\theta}(t)$**

With $\frac{\partial \mathfrak{L}}{\partial \boldsymbol{\theta}}^\top \delta \boldsymbol{\theta} = \mathbf{0}$ we have

$$\frac{\partial \mathfrak{L}}{\partial \boldsymbol{\theta}}^\top \delta \boldsymbol{\theta} = \int_{t_0}^{t_1} \left( \frac{\partial \boldsymbol{f}(\boldsymbol{x}, t; \boldsymbol{\theta})}{\partial \boldsymbol{\theta}}^\top \boldsymbol{\alpha} \right)^\top \delta \boldsymbol{\theta} \, dt + \int_{t_0}^{t_1} \boldsymbol{\gamma}^\top \delta \boldsymbol{\theta} \, dt = \int_{t_0}^{t_1} \left[ \boldsymbol{\gamma} + \frac{\partial \boldsymbol{f}(\boldsymbol{x}, t; \boldsymbol{\theta})}{\partial \boldsymbol{\theta}}^\top \boldsymbol{\alpha} \right]^\top \delta \boldsymbol{\theta} \, dt = \mathbf{0}. \tag{17}$$

**Gradient with respect to $\boldsymbol{w}_0$**

The gradient w.r.t. $\boldsymbol{w}_0$ shows that

$$\frac{\partial \mathfrak{L}}{\partial \boldsymbol{w}_0} = \boldsymbol{\beta}_0 = \mathbf{0}. \tag{18}$$

**Variations with respect to $\boldsymbol{w}(t)$**

With $\frac{\partial \mathfrak{L}}{\partial \boldsymbol{w}}^\top \delta \boldsymbol{w} = \mathbf{0}$ we have

$$\frac{\partial \mathfrak{L}}{\partial \boldsymbol{w}}^\top \delta \boldsymbol{w} = \int_{t_0}^{t_1} \left[ \frac{d\boldsymbol{\beta}}{dt} + \boldsymbol{A}^\top \boldsymbol{\beta} \right]^\top \delta \boldsymbol{w} \, dt - \boldsymbol{\beta}(t_1)^\top \delta \boldsymbol{w}(t_1) + \boldsymbol{\beta}(t_0)^\top \delta \boldsymbol{w}(t_0) + \int_{t_0}^{t_1} \boldsymbol{\gamma}^\top \frac{\partial \boldsymbol{h}(\boldsymbol{w}; \boldsymbol{\psi})}{\partial \boldsymbol{w}} \delta \boldsymbol{w} \, dt - \boldsymbol{\beta}_0^\top \delta \boldsymbol{w}(t_0),$$

$$= \int_{t_0}^{t_1} \left[ \frac{d\boldsymbol{\beta}}{dt} + \boldsymbol{A}^\top \boldsymbol{\beta} + \frac{\partial \boldsymbol{h}(\boldsymbol{w}; \boldsymbol{\psi})}{\partial \boldsymbol{w}}^\top \boldsymbol{\gamma} \right]^\top \delta \boldsymbol{w} \, dt - \boldsymbol{\beta}(t_1)^\top \delta \boldsymbol{w}(t_1) + [\boldsymbol{\beta}(t_0) - \boldsymbol{\beta}_0]^\top \delta \boldsymbol{w}(t_0) = \mathbf{0},$$

$$\tag{19}$$

holds for any variation $\delta \boldsymbol{w}$. Together with Eqs. (17) and (18), we obtain the second set of adjoint equations.

$$\frac{d\boldsymbol{\beta}}{dt} + \boldsymbol{A}^\top \boldsymbol{\beta} + \frac{\partial \boldsymbol{h}(\boldsymbol{w}; \boldsymbol{\psi})}{\partial \boldsymbol{w}}^\top \cdot \frac{\partial \boldsymbol{f}(\boldsymbol{x}, t; \boldsymbol{\theta})}{\partial \boldsymbol{\theta}}^\top \boldsymbol{\alpha} = \mathbf{0}, \boldsymbol{\beta}(t_1) = \mathbf{0}.$$

Simultaneously, we have $\boldsymbol{\beta}(t_0) = \boldsymbol{\beta}_0 = \mathbf{0}$ which provides a second boundary condition for the ODE $\boldsymbol{\beta}$ satisfies which adds up to the complications of directly solving the KKT conditions. In practice, we omit the condition that $\boldsymbol{\beta}_0 = \mathbf{0}$ to use the gradient-based methods.

---

**Algorithm 1** Training of the KoNODE Model

---

**Input:** trajectory $\boldsymbol{x}(t)$, start time $t_0$, stop time $t_1$, network parameters $\boldsymbol{\psi}$, matrix $\boldsymbol{A}$, IKSE $\Xi$

Initialize $\boldsymbol{A} = \boldsymbol{I}$ by setting $\alpha_i = 1$ and $\beta_i = 0$.

Estimate $\boldsymbol{w}(t_0) = \Xi(\{\boldsymbol{x}(t)\}_{t \in T_{\text{input}}}))$

OR Draw $\boldsymbol{w}(t_0) \sim \mathcal{N}(\boldsymbol{0}, \sigma^2 \boldsymbol{I})$

**repeat**

{Forward process}

$init\_aug\_state = [\boldsymbol{x}(t_0), \boldsymbol{w}(t_0)]$

**def** $aug\_dynamic([\boldsymbol{x}, \boldsymbol{w}], t, \boldsymbol{\psi}, \boldsymbol{A})$ :

**return** $[\boldsymbol{f}(\boldsymbol{x}; \boldsymbol{h}(\boldsymbol{w}; \boldsymbol{\psi})), \boldsymbol{A}\boldsymbol{w}]$

$[\boldsymbol{x}(t), \cdot] = \text{ODESolver}(init\_aug\_state, aug\_dynamic, t_0, t, [\boldsymbol{\psi}, \boldsymbol{A}])$

{Backward process}

Compute $\mathcal{L}(\boldsymbol{x})$ and retrieve $\nabla_t \mathcal{L}, \forall t \in [t_0, t_1]$

$init\_aug\_state = [\boldsymbol{x}(t_1), \boldsymbol{w}(t_1), \nabla_{t_1} \mathcal{L}, \boldsymbol{0_w}, \boldsymbol{0_\psi}, \boldsymbol{0_A}]$

**def** $aug\_dynamic([\boldsymbol{x}, \boldsymbol{w}, \boldsymbol{\alpha}, \boldsymbol{\beta}, \cdot, \cdot], t, \boldsymbol{\psi}, \boldsymbol{A})$ :

**return** $[\boldsymbol{f}(\boldsymbol{x}; \boldsymbol{h}(\boldsymbol{w}; \boldsymbol{\psi})),\ \boldsymbol{A}\boldsymbol{w}, -\frac{\partial \boldsymbol{f}}{\partial \boldsymbol{x}}^\top \boldsymbol{\alpha} - \nabla_t \mathcal{L}, -\frac{\partial \boldsymbol{h}}{\partial \boldsymbol{w}}^\top \frac{\partial \boldsymbol{f}}{\partial \boldsymbol{\theta}}^\top \boldsymbol{\alpha} - \boldsymbol{A}^\top \boldsymbol{\beta}, -\frac{\partial \boldsymbol{h}}{\partial \boldsymbol{\psi}}^\top \frac{\partial \boldsymbol{f}}{\partial \boldsymbol{\theta}}^\top \boldsymbol{\alpha},\ \boldsymbol{\beta} \cdot \boldsymbol{w}^\top]$

$[\cdot, \cdot, \frac{\partial \mathcal{L}}{\partial \boldsymbol{x}(t_0)}, \frac{\partial \mathcal{L}}{\partial \boldsymbol{w}(t_0)}, \frac{\partial \mathcal{L}}{\partial \boldsymbol{\psi}}, \frac{\partial \mathcal{L}}{\partial \boldsymbol{A}}] = \text{ODESolver}(init\_aug\_state, aug\_dynamic, t_1, t_0, [\boldsymbol{\psi}, \boldsymbol{A}])$

Backpropagate from $\boldsymbol{x}(t_0)$ by $\frac{\partial \mathcal{L}}{\partial \boldsymbol{x}(t_0)}$

Backpropagate from $\boldsymbol{w}(t_0)$ by $\frac{\partial \mathcal{L}}{\partial \boldsymbol{w}(t_0)}$

Update $\boldsymbol{\psi}$ and $\boldsymbol{A}$ by $\frac{\partial \mathcal{L}}{\partial \boldsymbol{\psi}}$ and $\frac{\partial \mathcal{L}}{\partial \boldsymbol{A}}$ respectively

**until** Convergence

---

## Gradient with respect to $\boldsymbol{A}$

The gradient w.r.t. $\boldsymbol{A}$ shows that

$$\frac{\partial \mathfrak{L}}{\partial \boldsymbol{A}} = \int_{t_0}^{t_1} \boldsymbol{\beta} \cdot \boldsymbol{w}^\top \, \mathrm{d}t. \tag{20}$$

## B.4. The Overall Adjoint Model and the Detailed Algorithm

The summarization of the formulas from App. B.3 leads to the overall model for the adjoint method which is expressed as integrations in Eq. (21) and algorithm in Alg. 1.

The integration is formulated as

$$\begin{bmatrix} \boldsymbol{x}(t) \\ \boldsymbol{w}(t) \\ \boldsymbol{\alpha}(t) \\ \boldsymbol{\beta}(t) \\ \frac{\partial \mathcal{L}}{\partial \boldsymbol{\psi}} \\ \frac{\partial \mathcal{L}}{\partial \boldsymbol{A}} \end{bmatrix} = \begin{bmatrix} \boldsymbol{x}(t_1) \\ \boldsymbol{w}(t_1) \\ \nabla_{t_1} \mathcal{L} \\ \boldsymbol{0}_{\boldsymbol{w}(t_1)} \\ \boldsymbol{0}_\psi \\ \boldsymbol{0}_A \end{bmatrix} + \int_{t_1}^t \begin{bmatrix} \boldsymbol{f}(\boldsymbol{x}(s), s; \boldsymbol{\theta}(s)) \\ \boldsymbol{A}\boldsymbol{w}(s) \\ -\frac{\partial \boldsymbol{f}(\boldsymbol{x}(s), s; \boldsymbol{\theta}(s))}{\partial \boldsymbol{x}(s)}^\top \cdot \boldsymbol{\alpha}(s) - \delta_T(s) \cdot \nabla_s \mathcal{L} \\ -\frac{\partial \boldsymbol{h}(\boldsymbol{w}(s); \boldsymbol{\psi})}{\partial \boldsymbol{w}(s)}^\top \cdot \frac{\partial \boldsymbol{f}(\boldsymbol{x}(s), s; \boldsymbol{\theta}(s))}{\partial \boldsymbol{\theta}(s)}^\top \cdot \boldsymbol{\alpha}(s) - \boldsymbol{A}^\top \cdot \boldsymbol{\beta}(s) \\ -\frac{\partial \boldsymbol{h}(\boldsymbol{w}(s); \boldsymbol{\psi})}{\partial \boldsymbol{\psi}}^\top \cdot \frac{\partial \boldsymbol{f}^\top(z(s), s; \boldsymbol{\theta}(s))}{\partial \boldsymbol{\theta}(s)} \cdot \boldsymbol{\alpha}(s) \\ -\boldsymbol{\beta}(s) \cdot \boldsymbol{w}^\top(s) \end{bmatrix} \mathrm{d}s, \tag{21}$$

where $\boldsymbol{0_x}$ represents a zero vector with the same shape of tensor $\boldsymbol{x}$ and $\delta_T(s) = \sum_{\substack{t \in T \\ t \neq t_1}} \delta(s - t)$ is a comb function based on the Dirac function $\delta$. The detailed algorithm is listed in Alg. 1.

### B.5. Details about Our Framework on Regularization Term Regarding Physical-informed Constraints

By modeling the intrinsic simple system, our framework not only improves the accuracy and the generality of the prediction tasks but also allows leveraging spectral tools to analyze system behavior. Furthermore, physics-informed constraints can be imposed by adding regularization terms in the objective function in Eq. (3), manuscript. In our formulation, the regularization to $\boldsymbol{\theta}$ can transform to $\boldsymbol{A}$, $\boldsymbol{w}_0$ or $\boldsymbol{\psi}$. So, we reformulate Eq. (3) as,

$$\min_{\boldsymbol{a}, \boldsymbol{w}_0, \boldsymbol{\psi}} \quad \mathcal{L}(\boldsymbol{x}) + \mathcal{R}(\boldsymbol{a}, \boldsymbol{w}_0, \boldsymbol{\psi}), \tag{22}$$

subject to (3b), (3c), and (3d), where $\boldsymbol{a}$ is the vector composed of all the non-zero elements of the matrix $\boldsymbol{A} \in \mathbb{R}^{m \times m}$, i.e., $\boldsymbol{a} = \left[ \alpha_1, \cdots, \alpha_{m/2}, \beta_1, \cdots, \beta_{m/2} \right]^\top$.

In this part, we derive the regularization inherited from the spectral method to constrain the eigenvalues of $\boldsymbol{A}$ as example (Mezic & Banaszuk, 2000). Especially, stable dynamic systems require the Koopman operator $\mathcal{K}$ to have $r$ eigenvalues on a unit circle, which leads to the $r$ diagonal elements for matrix $\boldsymbol{A}$ being $0$. We utilize a quadratic loss function to penalize for the constraint, i.e., $\mathcal{R}(\boldsymbol{a}) = \boldsymbol{\lambda}^\top \boldsymbol{S}_r \boldsymbol{\lambda}$ where $\boldsymbol{S}_r = \text{diag} \sum_{i=1}^r \boldsymbol{e}_{k_i}$ is the selector of eigenvalues and $\boldsymbol{\lambda} \triangleq [\boldsymbol{A}_{11}, \cdots, \boldsymbol{A}_{mm}]^\top = [\alpha_1, \alpha_1, \cdots, \alpha_{m/2}, \alpha_{m/2}]^\top$ are the diagonal elements of the matrix $\boldsymbol{A}$.

**Theorem B.2.** *The gradient of $R(\boldsymbol{a})$ w.r.t. the vector $\boldsymbol{a}$ is,*

$$\frac{\partial \mathcal{R}(\boldsymbol{a})}{\partial \boldsymbol{a}} = 2 \, \boldsymbol{vec} \left( diag \left[ \boldsymbol{S}_r \cdot \boldsymbol{\lambda} \right] \right),$$

*where $\boldsymbol{vec} : \mathbb{R}^{m \times m} \to \mathbb{R}^m$ is a mapping that maps matrix $\boldsymbol{A} \mapsto [\boldsymbol{a}_\alpha^\top, \boldsymbol{a}_\beta^\top]^\top$ where $\boldsymbol{a}_\alpha = [\boldsymbol{A}_{1,1} + \boldsymbol{A}_{2,2}, \cdots, \boldsymbol{A}_{m-1,m-1} + \boldsymbol{A}_{m,m}]^\top$, and $\boldsymbol{a}_\beta = [\boldsymbol{A}_{1,2} - \boldsymbol{A}_{2,1}, \cdots, \boldsymbol{A}_{n-1,n} - \boldsymbol{A}_{n,n-1}]^\top$. Note that under such definition, we have $\frac{\partial \mathcal{L}}{\partial \boldsymbol{a}} = \boldsymbol{vec} \left[ \frac{\partial \mathcal{L}}{\partial \boldsymbol{A}} \right]$.*

According to Thm. B.2, the gradient w.r.t. $\boldsymbol{A}$ in Eq. (9) is converted into the gradients regarding $\boldsymbol{a}$ as

$$\frac{\partial \mathfrak{L}}{\partial \boldsymbol{a}} = \boldsymbol{vec} \left( \text{diag} \left[ 2\boldsymbol{S}_r \cdot \boldsymbol{\lambda} \right] + \int_{t_0}^{t_1} \boldsymbol{\beta}(t) \cdot \boldsymbol{w}^\top(t) \, \mathrm{d}t \right). \tag{23}$$

## C. Theoretical Supplement

### C.1. The Proof of Equivalence Between Gradient and Adjoint Trajectories

We prove the two conclusions in Thm. 3.1 in the manuscript separately by Prop. C.1 and Thm. C.2 respectively.

**Proposition C.1.** *If the partial Jacobian matrix $\frac{\partial \boldsymbol{f}(\boldsymbol{x}(t), t; \boldsymbol{\theta}(t))}{\partial \boldsymbol{x}(t)}$ is Lipschitz continuous, the gradient trajectory for $\boldsymbol{x}(t)$ equals $\boldsymbol{\alpha}(t)$, that is $\frac{\partial \mathcal{L}}{\partial \boldsymbol{x}(t)} = \boldsymbol{\alpha}(t)$.*

*Proof.* As $\mathcal{L}(\boldsymbol{x}) = \frac{1}{|T|} \sum_{t \in T} \mathcal{E}[\boldsymbol{x}(t)]$ is auto independent regarding time, we denote $\mathcal{L}_{t_s:t_e}(\boldsymbol{x}) \triangleq \frac{1}{|T|} \sum_{t \in T \cap [t_s, t_e)} \mathcal{E}[\boldsymbol{x}(t)]$.

Subsequently, the gradient trajectory $\dfrac{\partial \mathcal{L}}{\partial \boldsymbol{x}(t)}$ follows

$$\frac{d}{dt}\frac{\partial \mathcal{L}}{\partial \boldsymbol{x}(t)} = \lim_{\Delta t \to 0^+} \frac{1}{\Delta t}\left[\frac{\partial \mathcal{L}}{\partial \boldsymbol{x}(t+\Delta t)} - \frac{\partial \mathcal{L}}{\partial \boldsymbol{x}(t)}\right],$$

$$= \lim_{\Delta t \to 0^+} \frac{1}{\Delta t}\left[\frac{\partial}{\partial \boldsymbol{x}(t+\Delta t)}\mathcal{L}_{t+\Delta t:\infty} - \frac{\partial}{\partial \boldsymbol{x}(t)}\mathcal{L}_{t:\infty}\right],$$

$$= \lim_{\Delta t \to 0^+} \frac{1}{\Delta t}\left[\frac{\partial}{\partial \boldsymbol{x}(t+\Delta t)}\mathcal{L}_{t+\Delta t:\infty} - \frac{d\boldsymbol{x}(t+\Delta t)}{d\boldsymbol{x}(t)}^\top \cdot \frac{\partial}{\partial \boldsymbol{x}(t+\Delta t)}\mathcal{L}_{t+\Delta t:\infty} - \frac{\partial}{\partial \boldsymbol{x}(t)}\mathcal{L}_{t:t+\Delta t}\right],$$

$$\overset{\star}{\underset{(T\cap(t,t+\Delta t)=\emptyset)}{=}} \lim_{\Delta t \to 0^+} \frac{1}{\Delta t}\cdot\left\{\boldsymbol{I} - \left[\frac{\partial}{\partial \boldsymbol{x}(t)}\left(\boldsymbol{x}(t) + \int_t^{t+\Delta t} \boldsymbol{f}(\boldsymbol{x}(s), s; \boldsymbol{\theta}(s))\, ds\right)\right]^\top\right\} \cdot \frac{\partial \mathcal{L}}{\partial \boldsymbol{x}(t+\Delta t)} - \frac{\partial \mathcal{E}[\boldsymbol{x}(t)]}{\partial \boldsymbol{x}(t)},$$

$$\overset{*}{=} \lim_{\Delta t \to 0^+} \frac{1}{\Delta t}\cdot\left\{-\int_t^{t+\Delta t}\left[\frac{\partial \boldsymbol{f}(\boldsymbol{x}(t), t; \boldsymbol{\theta}(t))}{\partial \boldsymbol{x}(t)} + o(|s-t|)\right] ds\right\}^\top \cdot \frac{\partial \mathcal{L}}{\partial \boldsymbol{x}(t+\Delta t)} - \nabla_t \mathcal{L},$$

$$= -\lim_{\Delta t \to 0^+}\left[\frac{\partial \boldsymbol{f}(\boldsymbol{x}(t), t; \boldsymbol{\theta}(t))}{\partial \boldsymbol{x}(t)} + \frac{1}{\Delta t}\cdot o(\Delta t^2)\right]^\top \cdot \frac{\partial \mathcal{L}}{\partial \boldsymbol{x}(t+\Delta t)} - \nabla_t \mathcal{L},$$

$$= -\frac{\partial \boldsymbol{f}(\boldsymbol{x}(t), t; \boldsymbol{\theta}(t))}{\partial \boldsymbol{x}(t)}^\top \cdot \frac{\partial \mathcal{L}}{\partial \boldsymbol{x}(t)} - \nabla_t \mathcal{L},$$

where the differentiation is converted into partial derivatives in $\overset{\star}{=}$ because $\boldsymbol{\theta}(s)$ is irrelevant to $\boldsymbol{x}(t)$ for $s \geq t$ and the infinitesimal quantity in $\overset{*}{=}$ is ensured by the finite values of Jacobian and Hessian matrices due to the Lipschitz continuity of partial Jacobian matrices.

Note that, $\boldsymbol{\alpha}(t)$ satisfies $\dfrac{d\boldsymbol{\alpha}(t)}{dt} + \dfrac{\partial \boldsymbol{f}(\boldsymbol{x}(t), t; \boldsymbol{\theta}(t))}{\partial \boldsymbol{x}(t)}^\top \boldsymbol{\alpha}(t) + \nabla_t \mathcal{L} = \boldsymbol{0}$ and $\boldsymbol{\alpha}(t_1) = \nabla_{t_1}\mathcal{L}$, which has the same differential equation and the boundary condition as the gradient trajectory $\partial \mathcal{L}/\partial \boldsymbol{x}(t)$. As $\partial \boldsymbol{f}(\boldsymbol{x}(t), t; \boldsymbol{\theta}(t))/\partial \boldsymbol{x}(t)$ is finite due to the existence of the derivate, the differential function of the ODE for $\boldsymbol{\alpha}(t)$ is Lipschitz continuous. Due to the Picard-Lindelöf Theorem, we have $\boldsymbol{\alpha}(t) = \partial \mathcal{L}/\partial \boldsymbol{x}(t)$. $\qquad\square$

We then prove the latter part of Thm. 3.1.

**Theorem C.2** (Thm. 3.1, manuscript). *If the partial Jacobian matrices $\dfrac{\partial \boldsymbol{f}(\boldsymbol{x}(t), t; \boldsymbol{\theta}(t))}{\partial \boldsymbol{x}(t)}$ and $\dfrac{\partial \boldsymbol{f}(\boldsymbol{x}(t), t; \boldsymbol{\theta}(t))}{\partial \boldsymbol{\theta}(t)}$ are Lipschitz continuous, the gradient trajectories of sequence $\boldsymbol{x}(t)$ and $\boldsymbol{w}(t)$ are $\boldsymbol{\alpha}(t)$ and $\boldsymbol{\beta}(t)$ respectively, i.e.,*

$$\frac{\partial \mathcal{L}}{\partial \boldsymbol{x}(t)} = \boldsymbol{\alpha}(t),\ \ \frac{\partial \mathcal{L}}{\partial \boldsymbol{w}(t)} = \boldsymbol{\beta}(t).$$

*Proof.* According to Prop. C.1, the first conclusion is proven. On the other hand, we have

$$
\frac{d}{dt}\frac{\partial \mathcal{L}}{\partial \boldsymbol{w}(t)} = \lim_{\Delta t \to 0^+} \frac{1}{\Delta t}\left[\frac{\partial \mathcal{L}}{\partial \boldsymbol{w}(t+\Delta t)} - \frac{\partial \mathcal{L}}{\partial \boldsymbol{w}(t)}\right],
$$

$$
= \lim_{\Delta t \to 0^+} \frac{1}{\Delta t}\left[\frac{\partial \mathcal{L}}{\partial \boldsymbol{w}(t+\Delta t)} - \frac{d\boldsymbol{x}(t+\Delta t)}{d\boldsymbol{w}(t)}^{\top}\cdot\frac{\partial \mathcal{L}}{\partial \boldsymbol{x}(t+\Delta t)} - \frac{d\boldsymbol{w}(t+\Delta t)}{d\boldsymbol{w}(t)}^{\top}\cdot\frac{\partial \mathcal{L}}{\partial \boldsymbol{w}(t+\Delta t)}\right],
$$

$$
= \lim_{\Delta t \to 0^+} \frac{1}{\Delta t}\cdot\left\{\left[\boldsymbol{I} - \frac{d\boldsymbol{w}(t+\Delta t)}{d\boldsymbol{w}(t)}\right]^{\top}\cdot\frac{\partial \mathcal{L}}{\partial \boldsymbol{w}(t+\Delta t)} - \int_t^{t+\Delta t}\boldsymbol{A}^{\top}\,ds\cdot\frac{\partial \mathcal{L}}{\partial \boldsymbol{w}(t+\Delta t)}\right.
$$

$$
\left. - \int_t^{t+\Delta t}\left[\frac{\partial \boldsymbol{f}(\boldsymbol{x}(t),t;\boldsymbol{\theta}(t))}{\partial \boldsymbol{\theta}(t)}\frac{d\boldsymbol{\theta}(t)}{d\boldsymbol{w}(t)} + o(|s-t|)\right]^{\top}\,ds\cdot\frac{\partial \mathcal{L}}{\partial \boldsymbol{x}(t+\Delta t)}\right\},
$$

$$
= \lim_{\Delta t \to 0^+} \frac{1}{\Delta t}\cdot\left\{\Delta t\left[\frac{\partial \boldsymbol{f}(\boldsymbol{x}(t),t;\boldsymbol{\theta}(t))}{\partial \boldsymbol{\theta}(t)}\cdot\frac{d\boldsymbol{h}(\boldsymbol{w}(t);\boldsymbol{\psi})}{d\boldsymbol{w}(t)}\right]^{\top}\frac{\partial \mathcal{L}}{\partial \boldsymbol{x}(t+\Delta t)} - \Delta t\boldsymbol{A}^{\top}\frac{\partial \mathcal{L}}{\partial \boldsymbol{w}(t+\Delta t)}\right\},
$$

$$
= -\frac{d\boldsymbol{h}(\boldsymbol{w}(t);\boldsymbol{\psi})}{d\boldsymbol{w}(t)}^{\top}\cdot\frac{\partial \boldsymbol{f}(\boldsymbol{x}(t),t;\boldsymbol{\theta}(t))}{\partial \boldsymbol{\theta}(t)}^{\top}\cdot\frac{\partial \mathcal{L}}{\partial \boldsymbol{x}(t)} - \boldsymbol{A}^{\top}\cdot\frac{\partial \mathcal{L}}{\partial \boldsymbol{w}(t)},
$$

$$
\overset{\star}{=} -\frac{d\boldsymbol{h}(\boldsymbol{w}(t);\boldsymbol{\psi})}{d\boldsymbol{w}(t)}^{\top}\cdot\boldsymbol{\gamma}(t) - \boldsymbol{A}^{\top}\cdot\frac{\partial \mathcal{L}}{\partial \boldsymbol{w}(t)},
$$

where $\overset{\star}{=}$ holds due to the other condition $\boldsymbol{\gamma}(t) = \dfrac{\partial \boldsymbol{f}(\boldsymbol{x}(t),t;\boldsymbol{\theta}(t))}{\partial \boldsymbol{\theta}(t)}\cdot\boldsymbol{\alpha}(t)$ and Prop. C.1. As, on the other hand,

$\dfrac{d\boldsymbol{\beta}(t)}{dt} + \boldsymbol{A}^{\top}\boldsymbol{\beta}(t) + \dfrac{\partial \boldsymbol{h}(\boldsymbol{w}(t);\boldsymbol{\psi})}{\partial \boldsymbol{w}(t)}^{\top}\boldsymbol{\gamma}(t) = 0$ and $\boldsymbol{\beta}(t_1) = \boldsymbol{0}$, which has the same differential equation and the boundary condition as the gradient trajectory $\partial \mathcal{L}/\partial \boldsymbol{w}(t)$. Using the Picard-Lindelöf Theorem again allows us to obtain $\boldsymbol{\beta}(t) = \partial \mathcal{L}/\partial \boldsymbol{w}(t)$. □

### C.2. The Proof of Error Bound in the Finite-Rank Koopman Model

This section proves Thm. 3.2 in the manuscript under the guidance of Kurdila & Bobade (2018). The major conclusions in the article starts from a definition of spectral approximation space.

**Definition C.3** (Spectral Approximation Space). A spectral approximation space $A^{r,p}[H]$ is defined for a Hilbert space $H$ as

$$
A^{r,p}[H] = \left\{f \in H \,\middle|\, \exists\ \text{bases}\ \{\psi_i\}_{i\in\mathbb{N}},\,s.t.\,\left[\sum_{i=1}^{\infty}\left(i^r\langle f,\psi_i\rangle\right)^p\right]^{\frac{1}{p}} < \infty\right\}.
$$

Kurdila & Bobade (2018) estimated the bound of low-rank estimations in Example 17 which leads to our result. The major conclusion of Example 17 is summarized as follows.

**Proposition C.4** (Example 17, Kurdila & Bobade (2018)). *For any observable function $f \in A^{r,q}[L_\mu^p(\Omega)]$ where $A^{r,q}$ is the spectral approximation space of the $L_\mu^p$ space where $\mu$ is the measure on space $\Omega$. If $\mathcal{U}_j$ represents the projection of the Koopman operator $\mathcal{U}$ to the $j$-dimensional subspace we have*

$$
\|(\mathcal{U} - \mathcal{U}_j)f\|_{L_\mu^p(\Omega)} \lesssim n_j^{-r}\cdot\|f\circ w\|_{A^{r,q}[L_\mu^p(\Omega)]} = n_j^{-r}\cdot\|\mathcal{U}f\|_{A^{r,q}[L_\mu^p(\Omega)]}.
$$

*where $n_j$ is the dimension of the subspace spanned by the $j$ basis functions which are closely related to the approximation method, $w$ is the evolving function for the argument $x$ of function $f$, and $r$ is the approximation rate that reveals the smoothness of the Koopman operator.*

Then we prove Thm. 3.2.

**Theorem C.5** (Thm. 3.2, manuscript). *The estimation of Koopman space by $m$-dimensional observables results in a relative error of*

$$
\frac{\|\boldsymbol{\xi}^{\top}\boldsymbol{K}\boldsymbol{u} - \mathcal{K}g\|_{L^2(\Theta)}}{\|\mathcal{K}g\|_{L^2(\Theta)}} \lesssim m^{-r},
$$

*where $r \triangleq - \max\limits_{m < i \le N} \log_i \langle g, u_i \rangle - \frac{1}{2}$ and $\boldsymbol{\xi} \triangleq \begin{bmatrix} \langle g, u_1 \rangle & \langle g, u_2 \rangle & \cdots & \langle g, u_m \rangle \end{bmatrix}$ with $N$ being the number of data.*

*Proof.* Note that the argument of the observable function is $\boldsymbol{\theta}(t)$ in the proposed setting. If the parameters are in a manifold $\Theta$ of order $m$, the observable function should be estimated by $g(\boldsymbol{\theta}) \approx \sum_{i=1}^{m} \xi_i \, u_i(\boldsymbol{\theta})$. As $\boldsymbol{u} = \begin{bmatrix} u_1(\boldsymbol{\theta}) & u_2(\boldsymbol{\theta}) & \cdots & u_m(\boldsymbol{\theta}) \end{bmatrix}^\top$ and the order-$m$ Koopman matrix is $\boldsymbol{K} \triangleq \mathrm{e}^{\boldsymbol{A}}$. The dimension of the spanned space $n_m = m$ for a common optimization method. Therefore, we can reformulate the error bound in Prop. C.4 under our formulation of KoNODE, to Thm. 3.2 in the manuscript.

Note that according to Def. C.3, $g \in A^{r,p}[H] \iff \langle g, u_i \rangle < i^{-\frac{1}{p}-r}, \forall i = m+1, \cdots, N$ which leads to the result when $p = 2$. Note that we only consider $i = m+1, \cdots, N$ because the first $m$ (finitely many) bases do not influence the convergence of the progression and the order is less than the number of training data.

$\square$

### C.3. The Guidance of the Koopman Dimension

In this section, we refer to the rank of the data $s$ to provide guidance for the choice of hyper-dimension $m$, which is the Koopman dimension. Prop. 3.3 of the manuscript exhibits theoretical upper and lower bounds for the number of dimensions. As discussed in the manuscript, the best dimension is larger in practice, and we chose 10 for simple systems and 100 for complex ones in experiments. The proof of the proposition is provided as follows using Def. C.6 and Lms. C.7 and C.8.

**Definition C.6** (Frontier Manifold). For a dynamic in Euclidean space, a fixed time $t_0$ and origin $\boldsymbol{\theta}^*$, the frontier manifold $\mathfrak{F}_t$ is the quotient space of the equivalent class decided by the trajectories, i.e., $\mathfrak{F}_0$ satisfies (1) $\boldsymbol{\theta}^* \in \mathfrak{F}_0$, and (2) the normal vector at $\boldsymbol{\theta} \in \mathfrak{F}_0$ is $\boldsymbol{\phi}(\boldsymbol{\theta}, t_0)$. Then, we define $\mathfrak{F}_t$ as the set of evolved $\boldsymbol{\theta}(t)$ with $\boldsymbol{\theta}(t_0) \in \mathfrak{F}_0$.

**Lemma C.7** (The Theoretical Upper-Bound of $m$). *Inequality $m \le s$ holds if: (1) The data trajectories are approximately in an $s$-dimensional manifold $\mathcal{M}$. (2) The function $f$ satisfies the Lipschitz condition and is differentiable.*

*Proof.* Since $\mathcal{M}$, being an $s$-dimensional manifold, admits a bijective chart $\psi : \mathcal{M} \to \mathbb{R}^s$. For the projection $\boldsymbol{x}^\perp \triangleq \operatorname*{argmin}\limits_{\boldsymbol{y} \in \mathcal{M}} \|\boldsymbol{y} - \boldsymbol{x}\|$, the mapping

$$\phi_t: \quad \boldsymbol{\Theta} \to \mathbb{R}^s,$$
$$\boldsymbol{\theta}(t) \mapsto \frac{d\psi(\boldsymbol{x}^\perp(t))}{d\boldsymbol{x}^\perp(t)} \cdot \boldsymbol{f}(\boldsymbol{x}^\perp(t), t; \boldsymbol{\theta}(t))$$

is bijective. This, together with the uniqueness of $\boldsymbol{x}(t)$ (by Picard–Lindelöf theorem), implies that $\boldsymbol{\theta}(t)$ lies on an $s$-dimensional manifold. Hence, $m \le s$. $\square$

**Lemma C.8** (The Theoretical Observable Function). *For $\boldsymbol{\theta}_0 \in \mathfrak{F}_s$, if the trajectory $\boldsymbol{\theta}(t)$ follows the dynamic*

$$\frac{d\boldsymbol{\theta}(t)}{dt} = \boldsymbol{\phi}(\boldsymbol{\theta}(t), t), \boldsymbol{\theta}(t_0) = \boldsymbol{\theta}_0,$$

*a scalar observable function $g(\boldsymbol{\theta}) \triangleq \mathcal{C} \cdot \exp(\frac{\lambda}{D} \int inv[\boldsymbol{\phi}(\boldsymbol{\theta}, t_s(\boldsymbol{\theta}))]^\top d\boldsymbol{\theta})$ maps it into a Koopman space.*

*Proof.* Note that for $\boldsymbol{\theta}(t_0) = \boldsymbol{\theta}_0 \in \mathfrak{F}_s$, $\boldsymbol{\theta}(t)$ is uniquely identified by the Picard-Lindelöf Theorem. Moreover, all elements in a frontier manifold $\mathfrak{F}_s$ are on different trajectories, which creates a bijection between $(\boldsymbol{\theta}_0, t)$ and $\boldsymbol{\theta}(t)$. Let $t_s(\boldsymbol{\theta})$ be the mapping from $\boldsymbol{\theta}$ to time under the condition that $\boldsymbol{\theta}_0 \in \mathfrak{F}_s$.

Then, let $g(\boldsymbol{\theta})$ be a scalar observable function,

$$g(\boldsymbol{\theta}) \triangleq \mathcal{C} \cdot \sigma^{-1} \text{ where } \sigma = \exp\left(-\frac{\lambda}{D} \int inv[\boldsymbol{\phi}(\boldsymbol{\theta}, t_s(\boldsymbol{\theta}))]^\top d\boldsymbol{\theta}\right),$$

where $\lambda$ and $\mathcal{C}$ are constants and $\text{inv}[\cdot]$ is the element-wise inverse of a vector. Therefore,

$$
\begin{aligned}
\frac{dg(\boldsymbol{\theta})}{dt} - k \cdot g(\boldsymbol{\theta}) &= \nabla_{\boldsymbol{\theta}} g(\boldsymbol{\theta})^{\top} \boldsymbol{\phi}(\boldsymbol{\theta}, t) - k \cdot g(\boldsymbol{\theta}) \cdot \boldsymbol{\phi}^{\top}(\boldsymbol{\theta}, t) \cdot \text{inv}[\boldsymbol{\phi}(\boldsymbol{\theta}, t)]/D, \\
&= \boldsymbol{\phi}^{\top}(\boldsymbol{\theta}, t) \cdot [\nabla_{\boldsymbol{\theta}} g(\boldsymbol{\theta}) - \lambda/D \cdot g(\boldsymbol{\theta}) \cdot \text{inv}[\boldsymbol{\phi}(\boldsymbol{\theta}, t)]], \\
&= \frac{1}{\sigma} \nabla_{\boldsymbol{\theta}} [g(\boldsymbol{\theta}) \cdot \sigma] = 0.
\end{aligned}
$$

Hence, $g(\boldsymbol{\theta})$ lies in the Koopman space. $\qquad\square$

**Proposition C.9** (Prop. 3.3, manuscript). *If the data trajectories are in a space of rank $s$ and function $\boldsymbol{f}$ satisfies the Lipschitz condition and is differentiable, we have $\lceil \frac{D}{D-s} \rceil \leq m \leq s$.*

*Proof.* Under the assumption of differentiability of $\boldsymbol{f}$, the $\boldsymbol{\theta}$ dynamic $\frac{d\boldsymbol{\theta}(t)}{dt} = \boldsymbol{\varphi}(\boldsymbol{\theta}(t), t)$ best models the trajectory $\boldsymbol{x}(t)$ where

$$
\boldsymbol{\varphi}(\boldsymbol{\theta}(t), t) \triangleq \left[\frac{\partial \boldsymbol{f}}{\partial \boldsymbol{\theta}(t)}\right]^{\dagger} \left[\frac{d^2 \boldsymbol{x}(t)}{dt^2} - \frac{\partial \boldsymbol{f}}{\partial t} - \frac{\partial \boldsymbol{f}}{\partial \boldsymbol{x}(t)} f(\boldsymbol{x}(t), t; \boldsymbol{\theta}(t))\right], \tag{24}
$$

and $\boldsymbol{A}^{\dagger}$ being the pseudo-inverse of matrix $\boldsymbol{A}$ which is the Jacobian matrix in the formula.

Lm. C.8 infers that for $\boldsymbol{\theta}_0 \in \mathfrak{F}_{t^*}, \forall t^*$, only one dimension of $w$ is needed. Lm. C.7 indicates that the gradient $\boldsymbol{\varphi}(\boldsymbol{\theta}(t), t)$ lies in a space with the highest rank of $s$, thus the frontier manifold $\mathfrak{F}_{t^*}$ covers a dimension of at least $D - s$. To ensure a full cover of $\boldsymbol{\Theta}$-space in the initialization, the lowest dimension of $w$ is $\lceil \frac{D}{D-s} \rceil$. Together with Lm. C.7, we may obtain the result. $\qquad\square$

**Please pay attention that, without loss of generality, we choose the function bases $u$ as the observable functions $g$ in the following part of App. C for simplicity.**

### C.4. The Proof for Estimation Accuracy of the Data-Driven Method

We prove Thm. 3.4 of the manuscript and propose additionally C.15 in this section. Nüske et al. (2023) proposed an error bound for the operator in Thm. 12. The major results were summarized by Def. C.10 and Thm. C.11.

**Definition C.10.** For $N$ bases $\{\psi_i\}_{i=1}^N$ in the Koopman space, define the matrices $\boldsymbol{A}, \boldsymbol{C} \in \mathbb{R}^{N \times N}$ to satisfy

$$
\boldsymbol{C}_{ij} \triangleq \langle \psi_i, \psi_j \rangle, \quad \boldsymbol{A}_{ij} \triangleq \langle \psi_i, \mathcal{L} \psi_j \rangle.
$$

Consequently, the Koopman operator in the full Koopman space is

$$
\mathcal{L}_{\mathbb{V}} \triangleq \boldsymbol{C}^{-1} \boldsymbol{A}, \text{ where } \mathbb{V} = \text{span}\left\{\{\psi_i\}_{i=1}^N\right\}.
$$

**Theorem C.11** (Thm. 12, Nüske et al. (2023)). *Assume that the observable functions are drawn independently and identically from $L_\mu^2(\mathbb{X})$ with the variance matrix of samples $\Sigma_\Phi$ where $\mathbb{X}$ is a compact and forward invariant space and $\mu$ is the normalized Lebesgue measure. For any error bound $\tilde{\varepsilon}$ and probabilistic tolerance $\tilde{\delta} \in (0, 1)$ we hav*

$$
\mathbb{P}(\|\mathcal{L}_{\mathbb{V}} - \tilde{\mathcal{L}}_m\|_F \leq \tilde{\varepsilon}) \geq 1 - \tilde{\delta},
$$

*for any amount $m \in \mathbb{N}$ of data points such that*

$$
m \geq \frac{N^2}{\delta \varepsilon^2} \|\Sigma_\Phi\|_F^2, \quad \text{where } \varepsilon = \min\left\{1, \frac{1}{\|\boldsymbol{A}\| \|\boldsymbol{C}^{-1}\|}\right\} \cdot \frac{\|\boldsymbol{A}\| \tilde{\varepsilon}}{2\|\boldsymbol{A}\| \|\boldsymbol{C}^{-1}\| + \tilde{\varepsilon}} \text{ and } \delta = \frac{\tilde{\delta}}{3}.
$$

We can obtain a corollary of the theorem which is Thm. 3.4 in the manuscript where **we need to highlight that the notations $m$ and $N$ are switched in places**. The proof is given below.

**Theorem C.12** (Thm. 3.4, manuscript). *For $N$ data drawn independently and identically from a Hilbert space at a single time point and any probabilistic tolerance $\delta \in (0, 1)$, we have*

$$\mathbb{P}(\|\boldsymbol{K} - \hat{\boldsymbol{K}}_N\|_F \leq \varepsilon) \geq 1 - \delta,$$

$$s.t. \ \varepsilon = \frac{2\sqrt{3}\,\sigma m^2 \|\boldsymbol{K}\|}{\min\{1, \|\boldsymbol{K}\|\} \cdot \sqrt{\delta N} - \sqrt{3}\,\sigma m^2},$$

*where $\boldsymbol{K}$ is the theoretical Koopman matrix, $\hat{\boldsymbol{K}}_N$ is the matrix estimated by $N$ data and $\sigma$ is the isotropic standard deviation in the space of $\boldsymbol{w}(t)$.*

*Proof.* Note that $\boldsymbol{C} = \boldsymbol{I}$ in Thm. C.11 for orthonormal bases hence

$$\|\boldsymbol{K}\| = \max_{\boldsymbol{\zeta}} \frac{\|\boldsymbol{K}\boldsymbol{\zeta}\|}{\|\boldsymbol{\zeta}\|} = \max_{\boldsymbol{\zeta}} \frac{\|\boldsymbol{C}^{-1}\boldsymbol{A}\boldsymbol{\zeta}\|}{\|\boldsymbol{A}\boldsymbol{\zeta}\|} \frac{\|\boldsymbol{A}\boldsymbol{\zeta}\|}{\|\boldsymbol{\zeta}\|} = \max_{\boldsymbol{\zeta}} \frac{\|\boldsymbol{A}\boldsymbol{\zeta}\|}{\|\boldsymbol{\zeta}\|} = \|\boldsymbol{A}\|.$$

Therefore $\varepsilon = \min\left\{1, \frac{1}{\|\boldsymbol{K}\|}\right\} \cdot \frac{\|\boldsymbol{K}\|\tilde{\varepsilon}}{2\|\boldsymbol{K}\| + \tilde{\varepsilon}}$. We assume on the other hand, that the data is sampled isotropically with standard deviation $\sigma$, and the variance matrix satisfies $\|\boldsymbol{\Sigma}_\Phi\|_F = \sigma m$. For the number of data points $N$ we thus have

$$\varepsilon \geq \sqrt{\frac{m^2}{\delta N} \cdot \sigma^2 m^2} = \frac{\sigma m^2}{\sqrt{\delta N}}.$$

Consequently,

$$\tilde{\varepsilon} \geq \frac{2\sqrt{3}\,\sigma m^2 \|\boldsymbol{K}\|}{\min\{1, \|\boldsymbol{K}\|\} \sqrt{\tilde{\delta} N} - \sqrt{3}\,\sigma m^2}.$$

Removing the tilde notations leads to the final expression. $\qquad \square$

On the other hand, the bound estimation of $\boldsymbol{w}(t)$ is obtained in a KKR model proposed in Def. C.13.

**Definition C.13** (Koopman Kernel Regression (KKR)). Consider a discrete-time dataset $\mathbb{D}_N = \{\boldsymbol{x}_T^{(i)}, y_T^{(i)}\}_{i=1}^N$ modeled by a linear time-invariant (LTI) predictor such that

$$\hat{y} = \mathbf{1}^\top \boldsymbol{z}_T, \ \boldsymbol{z}_{t+1} = \boldsymbol{\Lambda} \boldsymbol{z}_t, \ \boldsymbol{z}_0 = \hat{\boldsymbol{\phi}}(\boldsymbol{x}_0),$$

which is trained by the optimization problem,

$$\hat{M} \triangleq \operatorname*{argmin}_{M \in \mathcal{H}} \sum_{i=1}^N \|y_T^{(i)} - M(\boldsymbol{x}_T^{(i)})\|_{\mathbb{Y}_T}^2 + \gamma \|M\|_{\mathcal{H}}^2,$$

where $\gamma \in \mathbb{R}^+$ and $\|\cdot\|_{\mathcal{H}}$ is the reproducing kernel Hilbert space (RKHS) norm.

On the other hand, Thm. 3, Bevanda et al. (2024) finds the gap between the regression error of the true model and that by $N$ data $\hat{R}_N(\hat{M}) \triangleq \frac{1}{N} \sum_{i=1}^N \|y_T^{(i)} - \hat{M}(\boldsymbol{x}_T^{(i)})\|_{\mathbb{Y}_T}^2$. The conclusion is summarized as in Thm. C.15.

**Theorem C.14** (Thm. 3, Bevanda et al. (2024)). *Let $\mathbb{D}_N = \{\boldsymbol{x}_T^{(i)}, y_T^{(i)}\}_{i=1}^N$ be a dataset consistent with a Lipschitz system on a non-recurrent domain. Then the generalization gap of a KKR model $\hat{M}$ is, with probability $1 - \delta$, upper bounded by*

$$|R(\hat{M}) - \hat{R}_N(\hat{M})| \leq 4RB\sqrt{\frac{\kappa T^2}{N}} + \sqrt{\frac{8\log\frac{2}{\delta}}{N}},$$

*where $R$ is an upper bound on the loss in the domain, $B \in \mathbb{R}^+$ is the upper bound of model $M$ such that $\|M\|_{\mathcal{H}} \leq B$ and $\kappa$ the supremum of the base kernel.*

Formulating $\boldsymbol{w}(t)$ by the KKR model and applying the theorem lead to Thm. C.15.

**Theorem C.15** (Probabilistic Gap in Regression Error for Data-Driven Koopman Model). *For $N$ data trajectories, the theoretical difference between the mean residual and true residual satisfies,*

$$\mathbb{P}\left(|R - \hat{R}_N| \leq \varepsilon\right) = 1 - \delta,$$

$$s.t.\ \varepsilon = 4RBT\sqrt{\frac{\kappa}{N}} + \sqrt{\frac{8\log\frac{2}{\delta}}{N}},$$

*where $T$ is the number of time points, $R$ is an upper bound on the loss in the domain, $B$ is the bound of model parameters $\|M\|$, and $\kappa$ the supremum of the base kernel. Here,*

$$R_0 \triangleq \frac{1}{|T|} \sum_{t \in T} \mathbb{E}_{\boldsymbol{\theta}_0(t)} \left\| \boldsymbol{w}_0(t) - \boldsymbol{g}[\boldsymbol{\theta}_0(t)] \right\|^2,$$

$$\hat{R}_N \triangleq \frac{1}{N \cdot |T|} \sum_{t \in T} \sum_{i=1}^{N} \left\| \boldsymbol{w}_i(t) - \boldsymbol{g}[\boldsymbol{\theta}_i(t)] \right\|^2,$$

*where $\boldsymbol{w}_i(t+1) = \hat{\boldsymbol{K}}_N \boldsymbol{w}_i(t),\ \boldsymbol{w}_i(0) = \boldsymbol{g}(\boldsymbol{\theta}_i(0))$ holds for any $i \in \{0, 1, 2, \cdots\}$.*

Notably, Thm. C.15 shows the fact that the error bound of the regression errors in the Koopman model also follows $\mathcal{O}(1/\sqrt{N})$ as $N \to \infty$. The theorems in this section conclude that the errors of data-driven estimations are inversely controlled by the square root of data amount $N$.

## C.5. The Error Caused by Both Finite-Order Koopman Under Data-Driven Scenario

We further combine Thm. C.5 and Thm. C.11 into Thm. C.19 to obtain the error caused by both the finite-order estimation of the Koopman operator and the finite-data-driven model. We first introduce two lemmas.

**Lemma C.16** (Probabilistic Error of $L^2$ Norm). *Let $f \in L^2(\boldsymbol{\Theta})$ where $\boldsymbol{\Theta}$ has a metric defined by the normal density. If $\|f\|_{L^2(\boldsymbol{\Theta})} < \varepsilon^2 \delta$, then $\mathbb{P}(|f(x)| \leq \varepsilon) \geq 1 - \delta$.*

*Proof.* We prove the lemma by contradiction. We provide the hypothesis that $\mathbb{P}(|f(x)| > \varepsilon) > \delta$, then

$$\int_x f^2(x)\phi(x) > \int_A f^2(x)\phi(x) > \int_A \varepsilon^2 \phi(x) = \varepsilon^2 \mathbb{P}(x \in A) > \varepsilon^2 \delta,$$

where $A \triangleq \left\{ x \,\middle|\, |f(x)| > \varepsilon \right\}$. The result contradicts with the condition $\|f\|_{L^2(\boldsymbol{\Theta})} < \varepsilon^2 \delta$, hence the hypothesis is incorrect, i.e., $\mathbb{P}(|f(x)| > \varepsilon) \leq \delta$, which leads directly to the conclusion. $\square$

**Corollary C.17** (Probabilistic Error of Relative $L^2$ Error). *If the relative $L^2$ error of function $\Delta f, f \in L^2(\boldsymbol{\Theta})$ satisfies*

$$\|\Delta f\|_{L^2(\boldsymbol{\Theta})} < \varepsilon \|f\|_{L^2(\boldsymbol{\Theta})},$$

*we have*

$$\mathbb{P}\left(|\Delta f| \leq \varepsilon^{\frac{1}{3}} \cdot \|f\|_{L^2(\boldsymbol{\Theta})}^{\frac{1}{2}}\right) \geq 1 - \varepsilon^{\frac{1}{3}}.$$

*Proof.* Setting $f = \Delta f / \|f\|_{L^2(\boldsymbol{\Theta})}$ in Lm. C.16 would lead to the result. $\square$

**Lemma C.18** (Probabilistic Error of Koopman Estimation). *For the Koopman operator $\mathcal{K}$ and its order-$m$ approximation matrix $\boldsymbol{K} \in \mathbb{R}^{m \times m}$, we have for any $\varepsilon > 0$,*

$$\mathbb{P}\left(\left| \|\boldsymbol{K}\| - r(\mathcal{K}) \right| \leq \varepsilon\right) \geq 1 - 2m^{-\frac{r}{3}} - 2\delta,$$

*where $\varepsilon = 2m^{\frac{r}{6}}(m^r - 1)^{-\frac{1}{2}} \cdot \sqrt{\frac{\tau \cdot r(\mathcal{K})}{\min\{1, r(\mathcal{K})\}}}$ with $\tau$ being the bound satisfying $\mathbb{P}\left(\|\hat{g}\| < \tau \cdot |\hat{g}(\theta)|\right) \geq 1 - \delta$, and $r$ is the convergence rate characterizing the finiteness of the Koopman space.*

*Proof.* If $r(\mathcal{K})$ denotes the spectral radius of the operator $\mathcal{K}$, $\exists g^*, s.t. \mathcal{K}g^* = r(\mathcal{K})g^*$ since $\mathcal{K} - r(\mathcal{K})\mathcal{I}$ is not invertible. Let $\boldsymbol{\xi}^* \triangleq \begin{bmatrix} \langle g^*, g_1 \rangle & \langle g^*, g_2 \rangle & \cdots & \langle g^*, g_m \rangle \end{bmatrix}$ and $\hat{g} \triangleq \boldsymbol{\xi}^{*\top} \boldsymbol{g}$ the $m$-dimensional estimation of function $g^*$ and $\tilde{g} = g^* - \hat{g}$ the residual.

Note that as $\boldsymbol{K}$ is the estimation of $\mathcal{K}$, the vector $\boldsymbol{\xi}^*$, representing the projection of the eigenfunction corresponding to the spectral radius, is the same for $\boldsymbol{K}$, i.e., $\boldsymbol{\xi}^{*\top} \boldsymbol{K} = \|\boldsymbol{K}\| \cdot \boldsymbol{\xi}^{*\top}$.

Thm. C.5 implies that $\mathbb{P}\left( \left| \boldsymbol{\xi}^\top \boldsymbol{K} \boldsymbol{g}(\boldsymbol{\theta}) - \mathcal{K}g(\boldsymbol{\theta}) \right| > m^{-\frac{r}{3}} \cdot \|\mathcal{K}g\|_{L^2(\boldsymbol{\Theta})}^{\frac{1}{2}} \right) < m^{-\frac{r}{3}}$ for random $\boldsymbol{\theta}$'s. The conclusion satisfies for any $g \in L^2(\boldsymbol{\Theta})$ and $\boldsymbol{\xi} \triangleq \begin{bmatrix} \langle g, g_1 \rangle & \langle g, g_2 \rangle & \cdots & \langle g, g_m \rangle \end{bmatrix}$. We can then obtain by setting $g = g^*$,

$$\mathbb{P}\left( \left| \|\boldsymbol{K}\| - r(\mathcal{K}) \right| \cdot |\hat{g}(\boldsymbol{\theta})| > \varepsilon^* + r(\mathcal{K}) \cdot |\tilde{g}(\boldsymbol{\theta})| \right) \leq \mathbb{P}\left( \left| \|\boldsymbol{K}\| \cdot \boldsymbol{\xi}^{*\top} \boldsymbol{g}(\boldsymbol{\theta}) - r(\mathcal{K}) \cdot \boldsymbol{\xi}^{*\top} \boldsymbol{g}(\boldsymbol{\theta}) - r(\mathcal{K}) \cdot \tilde{g}(\boldsymbol{\theta}) \right| > \varepsilon^* \right) < m^{-\frac{r}{3}},$$

where $\varepsilon^* = m^{-\frac{r}{3}} \cdot \|\mathcal{K}g\|_{L^2(\boldsymbol{\Theta})}^{\frac{1}{2}}$ hence

$$\mathbb{P}\left( \left| \|\boldsymbol{K}\| - r(\mathcal{K}) \right| \leq \frac{1}{|\hat{g}(\boldsymbol{\theta})|} \cdot \left[ m^{-\frac{r}{3}} \cdot \|\mathcal{K}g^*\|_{L^2(\boldsymbol{\Theta})}^{\frac{1}{2}} + r(\mathcal{K})(m^r - 1)^{-\frac{1}{3}} \cdot \|\hat{g}\|_{L^2(\boldsymbol{\Theta})}^{\frac{1}{2}} \right] \right),$$

$$\geq \mathbb{P}\left( \left| \|\boldsymbol{K}\| - r(\mathcal{K}) \right| \leq \frac{1}{|\hat{g}(\boldsymbol{\theta})|} \cdot \left[ m^{-\frac{r}{3}} \cdot \|\mathcal{K}g^*\|_{L^2(\boldsymbol{\Theta})}^{\frac{1}{2}} + r(\mathcal{K}) \cdot |\tilde{g}(\boldsymbol{\theta})| \right] \right) \cdot \mathbb{P}\left( |\tilde{g}| \leq (m^r - 1)^{-\frac{1}{3}} \cdot \|\hat{g}\|_{L^2(\boldsymbol{\Theta})}^{\frac{1}{2}} \right),$$

$$\overset{①}{\geq} (1 - m^{-\frac{r}{3}}) \cdot \left[ 1 - (m^r - 1)^{-\frac{1}{3}} \right] \overset{②}{\geq} 1 - 2m^{-\frac{r}{3}},$$

where $\overset{②}{\geq}$ is because $(m^r - 1)^{-\frac{1}{3}} - m^{-\frac{r}{3}} \cdot (m^r - 1)^{-\frac{1}{3}} \leq m^{-\frac{r}{3}}$ and $\overset{①}{\geq}$ holds due to the fact that

$$\frac{\|\tilde{g}\|}{\|\hat{g}\|} \overset{③}{\leq} \frac{\|\tilde{g}\|/\|g^*\|}{(\|g^*\| - \|\tilde{g}\|)/\|g^*\|} \overset{④}{\leq} \frac{m^{-r}}{1 - m^{-r}} = \frac{1}{m^r - 1}, \text{ for the } L^2(\boldsymbol{\Theta}) \text{ norm} \| \cdot \|,$$

where $\overset{③}{\leq}$ holds due to the triangle inequality in the denominator and $\overset{④}{\leq}$ holds because of Prop. C.4.

Under $L^2(\boldsymbol{\Theta})$ norm, if $\|\hat{g}\| < \tau^2 \cdot \hat{g}^2(\boldsymbol{\theta})$, we have $\|\mathcal{K}g^*\| \leq \|\mathcal{K}\| \cdot \|g^*\| = r(\mathcal{K}) \cdot \|\hat{g} + \tilde{g}\| \leq r(\mathcal{K}) \cdot (\|\hat{g}\| + \|\tilde{g}\|) < \frac{r(\mathcal{K})}{1 - m^{-r}} \cdot \tau^2 \cdot \hat{g}^2(\boldsymbol{\theta})$. As a result, for $\tau$ satisfying $\mathbb{P}(\|\hat{g}\| < \tau^2 \cdot \hat{g}^2(\boldsymbol{\theta})) \geq 1 - \delta$, $\mathbb{P}(\|\mathcal{K}g^*\| < \frac{r(\mathcal{K})}{1 - m^{-r}} \cdot \tau^2 \cdot \hat{g}^2(\boldsymbol{\theta})) \geq 1 - \delta$. Consequently,

$$\mathbb{P}\left( \left| \|\boldsymbol{K}\| - r(\mathcal{K}) \right| \leq 2m^{\frac{r}{6}} (m^r - 1)^{-\frac{1}{2}} \cdot \sqrt{\frac{\tau \cdot r(\mathcal{K})}{\min\{1, r(\mathcal{K})\}}} \right),$$

$$\geq \mathbb{P}\left( \left| \|\boldsymbol{K}\| - r(\mathcal{K}) \right| \leq m^{-\frac{r}{3}} \cdot r^{\frac{1}{2}}(\mathcal{K}) \cdot \sqrt{\tau} \cdot m^{\frac{r}{2}} \cdot (m^r - 1)^{-\frac{1}{2}} + r(\mathcal{K}) \cdot (m^r - 1)^{-\frac{1}{3}} \cdot \sqrt{\tau} \right),$$

$$\geq (1 - 2m^{-\frac{r}{3}}) \cdot (1 - \delta)^2 > 1 - 2m^{-\frac{r}{3}} - 2\delta.$$

The result directly leads to the conclusion. $\qquad \square$

**Theorem C.19.** *For $N$ data drawn independently and identically from a Hilbert space at a single time point and any observable function $g$ and its corresponding $\boldsymbol{\xi}$, with any probabilistic tolerance $\delta \in (0, 1)$ we have*

$$\mathbb{P}\left( \frac{|\boldsymbol{\xi}^\top \hat{\boldsymbol{K}}_N \boldsymbol{w}(t) - \mathcal{K}g(\boldsymbol{\theta}(t))|}{\|\boldsymbol{\xi}\| \cdot \|\boldsymbol{w}(t)\|} \leq \varepsilon \right) \geq 1 - \delta,$$

$$s.t. \ \varepsilon = 2\|\boldsymbol{\xi}\| \cdot \|\boldsymbol{w}(t)\| \cdot \frac{\sqrt{3}\,\sigma m^2 \cdot r(\mathcal{K})}{\sqrt{N(\delta - 2m^{-\frac{r}{3}})} \cdot \min\{1, r(\mathcal{K})\} - \sqrt{3}\,\sigma m^2} + o(m^{-\frac{r}{3}}),$$

*where $\hat{\boldsymbol{K}}_N$ is the order-$m$ Koopman matrix estimated by the $N$ data.*

*Proof.* We find the following inequality of the mapped observable functions at argument $\boldsymbol{\theta}(t)$.

$$|\boldsymbol{\xi}^\top \hat{\boldsymbol{K}}_N \boldsymbol{w}(t) - \mathcal{K}g(\boldsymbol{\theta}(t))| \leq |\boldsymbol{\xi}^\top \hat{\boldsymbol{K}}_N \boldsymbol{w}(t) - \boldsymbol{\xi}^\top \boldsymbol{K} \boldsymbol{w}(t)| + |\boldsymbol{\xi}^\top \boldsymbol{K} \boldsymbol{w}(t) - \mathcal{K}g(\boldsymbol{\theta}(t))|,$$

$$\overset{\star}{\leq} \|\boldsymbol{\xi}\| \cdot \|\hat{\boldsymbol{K}}_N - \boldsymbol{K}\|_F \cdot \|\boldsymbol{w}(t)\| + |\boldsymbol{\xi}^\top \boldsymbol{K} \boldsymbol{w}(t) - \mathcal{K}g(\boldsymbol{\theta}(t))|,$$

where $\overset{\star}{\leq}$ is true due to the Cauchy-Schwartz inequality indicating that $\|A\boldsymbol{x}\| \leq \|A\|_F \cdot \|\boldsymbol{x}\|$.

Note that Cor. C.17 indicates $\mathbb{P}\left(|\boldsymbol{\xi}^\top \boldsymbol{K}\boldsymbol{g}(\boldsymbol{\theta}) - \mathcal{K}g(\boldsymbol{\theta})| \leq m^{-\frac{r}{3}} \cdot \|\mathcal{K}g\|_{L^2(\boldsymbol{\Theta})}^{\frac{1}{2}}\right) \geq 1 - m^{-\frac{r}{3}}$ for random $\boldsymbol{\theta}$'s. Consequently,

$$
\begin{aligned}
&\mathbb{P}(|\boldsymbol{\xi}^\top \hat{\boldsymbol{K}}_N \boldsymbol{w}(t) - \mathcal{K}g(\boldsymbol{\theta}(t))| \leq \tilde{\varepsilon}), \\
&\geq \mathbb{P}\left(\|\boldsymbol{\xi}\| \cdot \|\hat{\boldsymbol{K}}_N - \boldsymbol{K}\|_F \cdot \|\boldsymbol{w}(t)\| \leq \tilde{\varepsilon} - m^{-\frac{r}{3}} \cdot \|\mathcal{K}g\|_{L^2(\boldsymbol{\Theta})}^{\frac{1}{2}}\right) \cdot \mathbb{P}\left(|\boldsymbol{\xi}^\top \boldsymbol{K}\boldsymbol{w}(t) - \mathcal{K}g(\boldsymbol{\theta}(t))| \leq m^{-\frac{r}{3}} \cdot \|\mathcal{K}g\|_{L^2(\boldsymbol{\Theta})}^{\frac{1}{2}}\right), \\
&\geq (1 - m^{-\frac{r}{3}}) \cdot \mathbb{P}\left(\|\hat{\boldsymbol{K}}_N - \boldsymbol{K}\|_F \leq \frac{\tilde{\varepsilon} - m^{-\frac{r}{3}} \cdot \|\mathcal{K}g\|_{L^2(\boldsymbol{\Theta})}^{\frac{1}{2}}}{\|\boldsymbol{\xi}\| \cdot \|\boldsymbol{w}(t)\|}\right).
\end{aligned}
$$

Letting $\mathbb{P}\left(\|\hat{\boldsymbol{K}}_N - \boldsymbol{K}\|_F \leq \frac{\tilde{\varepsilon} - m^{-\frac{r}{3}} \cdot \|\mathcal{K}g\|_{L^2(\boldsymbol{\Theta})}^{\frac{1}{2}}}{\|\boldsymbol{\xi}\| \cdot \|\boldsymbol{w}(t)\|}\right)$ be $1 - \tilde{\delta}$ and applying Thm. C.11 obtains

$$
\tilde{\varepsilon} = \|\boldsymbol{\xi}\| \cdot \|\boldsymbol{w}(t)\| \cdot \frac{2\sqrt{3}\,\sigma m^2 \|\boldsymbol{K}\|}{\min\{1, \|\boldsymbol{K}\|\} \cdot \sqrt{N\tilde{\delta}} - \sqrt{3}\,\sigma m^2} + m^{-\frac{r}{3}} \cdot \|\mathcal{K}g\|_{L^2(\boldsymbol{\Theta})}^{\frac{1}{2}}.
$$

Let $\mathcal{B}(k) \triangleq 2\|\boldsymbol{\xi}\| \cdot \|\boldsymbol{w}(t)\| \cdot \max\left\{\frac{k}{p-1}, \frac{k}{pk-1}\right\} + m^{-\frac{r}{3}} \cdot \|\mathcal{K}g\|_{L^2(\boldsymbol{\Theta})}^{\frac{1}{2}}$ for $p = \frac{1}{\sigma m^2}\sqrt{\frac{N\tilde{\delta}}{3}}$, then $\tilde{\varepsilon} = \mathcal{B}(\|\boldsymbol{K}\|)$.

Note that Lm. C.18 shows that
$$
\mathbb{P}\left(\left|\|\boldsymbol{K}\| - r(\mathcal{K})\right| \leq \varepsilon^*\right) \geq 1 - 2m^{-r} - 2\delta^*,
$$

where $\varepsilon^* = 2m^{\frac{r}{6}}(m^r - 1)^{-\frac{1}{2}}\sqrt{\frac{\tau \cdot r(\mathcal{K})}{\min\{1, r(\mathcal{K})\}}}$.

Let $I$ be the interval $[r(\mathcal{K}) - \varepsilon^*, r(\mathcal{K}) + \varepsilon^*]$ we have

$$
\mathbb{P}\left(\left|\mathcal{B}(\|\boldsymbol{K}\|) - \mathcal{B}(r(\mathcal{K}))\right| \leq \Delta\mathcal{B}\right) \geq \mathbb{P}\left(\left|\|\boldsymbol{K}\| - r(\mathcal{K})\right| \leq \varepsilon^*\right) \geq 1 - 2m^{-\frac{r}{3}} - 2\delta^*,
$$

where $\Delta\mathcal{B} \triangleq \varepsilon^* \cdot \sup_{k \in I} |\mathcal{B}'(k)|$ and $\frac{\sup_{k \in I} |\mathcal{B}'(k)|}{2\|\boldsymbol{\xi}\| \cdot \|\boldsymbol{w}(t)\|} = \begin{cases} 1/(pI_{\min} - 1)^2, & \text{if } I_{\min} = r(\mathcal{K}) - \varepsilon^* < \frac{\sqrt{p-1}+1}{p}, \\ 1/(p-1), & \text{otherwise.} \end{cases}$

Consequently,

$$
\begin{aligned}
&\mathbb{P}\left(|\boldsymbol{\xi}^\top \hat{\boldsymbol{K}}_N \boldsymbol{w}(t) - \mathcal{K}g(\boldsymbol{\theta}(t))| \leq \mathcal{B}(r(\mathcal{K})) + \Delta\mathcal{B}\right), \\
&\geq \mathbb{P}\left(|\boldsymbol{\xi}^\top \hat{\boldsymbol{K}}_N \boldsymbol{w}(t) - \mathcal{K}g(\boldsymbol{\theta}(t))| \leq \mathcal{B}(\|\boldsymbol{K}\|)\right) \cdot \mathbb{P}\left(|\mathcal{B}(\|\boldsymbol{K}\|) - \mathcal{B}(r(\mathcal{K}))| \leq \Delta\mathcal{B}\right), \\
&\geq (1 - \tilde{\delta}) \cdot (1 - 2m^{-\frac{r}{3}} - 2\delta) > 1 - 2m^{-\frac{r}{3}} - 2\delta^* - \tilde{\delta},
\end{aligned}
$$

where $\mathcal{B}(r(\mathcal{K})) = 2\|\boldsymbol{\xi}\| \cdot \|\boldsymbol{w}(t)\| \cdot \frac{r(\mathcal{K})}{p\min\{1, r(\mathcal{K})\} - 1} + m^{-\frac{r}{3}} \cdot \|\mathcal{K}g\|_{L^2(\boldsymbol{\Theta})}^{\frac{1}{2}}$.

Note that $\hat{g}(\boldsymbol{\theta}(t)) = \boldsymbol{\xi}^\top \boldsymbol{w}(t)$ we have

$$
\begin{aligned}
\mathcal{B}(r(\mathcal{K})) + \Delta\mathcal{B} = &2\|\boldsymbol{\xi}\| \cdot \|\boldsymbol{w}(t)\| \cdot \frac{r(\mathcal{K})}{p\min\{1, r(\mathcal{K})\} - 1} + m^{-\frac{r}{3}} \cdot \sqrt{\frac{r(\mathcal{K})}{1 - m^{-r}}} \cdot \tau d \\
&+ 4\|\boldsymbol{\xi}\| \cdot \|\boldsymbol{w}(t)\| \cdot m^{\frac{r}{6}}(m^r - 1)^{-\frac{1}{2}} \cdot \sqrt{\frac{\tau \cdot r(\mathcal{K})}{\min\{1, r(\mathcal{K})\}}} \cdot \max\left\{\frac{1}{(p(r(\mathcal{K}) - \varepsilon^*) - 1)^2}, \frac{1}{p-1}\right\},
\end{aligned}
$$

where $d$ is the bound of the observable function. Let the RHS be $1 - \delta$, then $\tilde{\delta} = \delta - 2m^{-\frac{r}{3}} - 3\delta^*$ and $p = \sqrt{N(\delta - 2m^{-\frac{r}{3}} - 3\delta^*)}/(\sqrt{3}\,\sigma m^2)$ which leads directly to the conclusion by converting the last two terms into the infinitesimal term and omitting $\delta^*$. $\qquad\square$

## C.6. The Tightness of the Overall Estimation Compared to the ODE Models

For the overall error of the proposed model, we dig deeper to find the theoretical reasoning for its superiority. Thm. C.22 provides the theoretical advantage of the proposed method in estimating complex systems compared to conventional ones. The theorem will not be included in the manuscript for three reasons, i.e., (1) the inequality of theory is not significantly tight, (2) the numerical experiment better proves the effectiveness of the proposed method, and (3) better estimations of the error may require methods similar to those for Neural ODE evaluation such as boundings by the eigenvalues of the feedback gain matrix.

We propose two definitions for the theorem.

**Definition C.20** (The Static Dynamic Model Condition). The dynamic satisfies the condition if and only if for the function $g$ and any $\tilde{\varepsilon}$, $\exists \tilde{\delta} > 0$,

$$\mathbb{P}(|g(\boldsymbol{\theta}^*) - g[\tilde{\boldsymbol{\theta}}(t)]| < \tilde{\varepsilon}) \geq 1 - \tilde{\delta},$$

where $\boldsymbol{\theta}^* \triangleq \underset{\boldsymbol{\theta}}{\operatorname{argmin}} \int_t \left\| \boldsymbol{f}(\boldsymbol{x}(t), t; \boldsymbol{\theta}) - \boldsymbol{f}(\boldsymbol{x}(t), t; \tilde{\boldsymbol{\theta}}(t)) \right\|$ is the LSE estimation of a static $\boldsymbol{\theta}$ and $\tilde{\boldsymbol{\theta}}(t)$ is randomly drawn from the ideal trajectory defined in Eq. (24). The condition holds when the true dynamic behind the trajectory is static, which ensures a good fit of the NODE.

**Definition C.21** (Estimation Bound of Trajectory). We define an upper bound for the trajectory estimation as,

$$\mathcal{B}[\hat{\boldsymbol{\theta}}(t)] \triangleq \sup_{\boldsymbol{\theta}} \left\| \frac{\partial f(\boldsymbol{x}(t), t; \boldsymbol{\theta})}{\partial \boldsymbol{\theta}} \right\| \cdot \|\hat{\boldsymbol{\theta}}(t) - \tilde{\boldsymbol{\theta}}(t)\|, \tag{25}$$

where $\tilde{\boldsymbol{\theta}}(t)$ is the ideal trajectory in Eq. (24). It is clear that the overall estimation error of the trajectory using $\hat{\boldsymbol{\theta}}(t)$ can be bounded by $\mathcal{B}[\hat{\boldsymbol{\theta}}(t)]$.

Consequently, We prove the theorem as Thm. C.22.

**Theorem C.22** (The Superiority of the Proposed Model). *During the modeling of trajectory $x(t)$, the estimation error upper bound of the proposed model is smaller than that of a conventional ODE method if the observable function $g$ does NOT satisfy* the static dynamic model condition, *i.e.,*

$$\mathbb{P}(\mathcal{B}[\boldsymbol{\theta}^\dagger(t)] \leq \mathcal{B}[\boldsymbol{\theta}^*]) \geq 1 - \delta^*,$$

*where $\boldsymbol{\theta}^\dagger(t)$ is the trajectory predicted by the proposed method while $\boldsymbol{\theta}^*$ consists of the static parameters in the conventional ODE method.*

*Proof.* We use the notations $\boldsymbol{\theta}^*$, $\tilde{\boldsymbol{\theta}}(t)$, $\tilde{\delta}$ and $\tilde{\varepsilon}$ defined in Def. C.20. Apparently, Eq. (25) gives a bound of the estimation error where the "sup" exists as $\boldsymbol{f}$ is Lipschitz continuous, and the bound is reached for a linear mapping $\boldsymbol{f}$.

Note that $\hat{\boldsymbol{\theta}}(t) = \boldsymbol{\theta}^*$ in NODE and $\hat{\boldsymbol{\theta}}(t) = \boldsymbol{\theta}^\dagger(t) \triangleq \boldsymbol{h}(\hat{\boldsymbol{w}}(t); \boldsymbol{\psi}) \approx \boldsymbol{g}^{-1}(\boldsymbol{\xi}^\top \hat{\boldsymbol{w}}(t))$ in the proposed Koopman model where $\hat{\boldsymbol{w}}(t)$ satisfies the Koopman model of matrix $\hat{\boldsymbol{K}}_N$. Thm. 3.4, Manuscript shows that

$$\mathbb{P}(|\boldsymbol{\xi}^\top \hat{\boldsymbol{w}}(t + \Delta t) - \boldsymbol{g}[\tilde{\boldsymbol{\theta}}(t + \Delta t)]| < \varepsilon^* \cdot \|\boldsymbol{\xi}\| \cdot \|\hat{\boldsymbol{w}}(t)\|) \geq 1 - \delta,$$

with $\varepsilon^*$ representing the RHS. As the base $u_i(t) \in L^2(\boldsymbol{\Theta})$ for observables $\hat{\boldsymbol{w}}$, Lemma C.12, Appendices implies $\mathbb{P}(|\hat{\boldsymbol{w}}_i(t)| \leq \frac{1}{\sqrt{\delta}}) \geq 1 - \delta$ if $\|u_i(t)\|_{L^2(\boldsymbol{\Theta})} = 1$ and $\max |\boldsymbol{\xi}_i| \leq r(\mathcal{K})$,

$$\mathbb{P}(\|\boldsymbol{\xi}\| \cdot \|\hat{\boldsymbol{w}}(t)\| \leq \frac{m\sqrt{m} \cdot r(\mathcal{K})}{\sqrt{\delta}}) \geq 1 - \delta.$$

Consequently, if we let

$$\tilde{\varepsilon} = \frac{2\sqrt{3}\sigma m^{\frac{7}{2}} r^2(\mathcal{K})}{\sqrt{N\delta(\delta - 2m^{-\frac{r}{3}})} \cdot \min\{1, r(\mathcal{K})\} - \sqrt{3}\delta\sigma m^2} + o(m^{-\frac{r}{3}}),$$

then $\mathbb{P}(|\boldsymbol{\xi}^\top \hat{\boldsymbol{w}}(t + \Delta t) - \boldsymbol{g}[\tilde{\boldsymbol{\theta}}(t + \Delta t)]| < \tilde{\varepsilon}) \geq 1 - 2\delta$.

*Table 3.* Test MSEs of ablation study across Hypernet, CONDE, OURS, and OURS$_{\text{SPARSE}}$ on NON-LINEAR OSCILLATOR (01) and DAMPED OSCILLATOR FAMILY (02). Results are mean across 5 runs with different seeds. The best results are highlighted in **bold**, while the second-best is underlined.

| DATA | HYPERNET | CNODE | OURS$_{\text{SPARSE}}$ | OURS |
|---|---|---|---|---|
| 01 | 0.381 | 0.614 | 0.114 | **0.085** |
| 02 | 0.023 | 0.446 | 0.009 | **0.008** |

*Table 4.* Results of different ODE solvers on non-linear oscillator dataset.

| METHODS | EULER | RK4 | DOPRI5 |
|---|---|---|---|
| NODE | 0.333 | 0.361 | 0.366 |
| KONODE | 0.053 | 0.098 | 0.099 |

On the other hand, for $g$ failed to satisfy the static dynamic model condition at error $\tilde{\varepsilon}$ we have

$$\mathbb{P}(\|\boldsymbol{\theta}^{\dagger}(t) - \tilde{\boldsymbol{\theta}}(t)\| \leq \|\boldsymbol{\theta}^* - \tilde{\boldsymbol{\theta}}(t)\|),$$
$$=\mathbb{P}(|\boldsymbol{g}[\boldsymbol{\theta}^{\dagger}(t)] - \boldsymbol{g}[\tilde{\boldsymbol{\theta}}(t)]| \leq |g(\boldsymbol{\theta}^*) - \boldsymbol{g}[\tilde{\boldsymbol{\theta}}(t)]|)$$
$$- \mathbb{P}(|\boldsymbol{g}[\boldsymbol{\theta}^{\dagger}(t)] - g[\tilde{\boldsymbol{\theta}}(t)]| \leq |g(\boldsymbol{\theta}^*) - g[\tilde{\boldsymbol{\theta}}(t)]| \text{ and } \tilde{\boldsymbol{\theta}}(t) \in Q),$$
$$\geq\mathbb{P}(|\boldsymbol{g}[\boldsymbol{\theta}^{\dagger}(t)] - \boldsymbol{g}[\tilde{\boldsymbol{\theta}}(t)]| \leq |g(\boldsymbol{\theta}^*) - \boldsymbol{g}[\tilde{\boldsymbol{\theta}}(t)]|) - \mathbb{P}(\tilde{\boldsymbol{\theta}}(t) \in Q) \geq 1 - \delta^*,$$

where $Q \triangleq \{\boldsymbol{\theta} \mid |g(\boldsymbol{\theta}^*) - \boldsymbol{g}(\boldsymbol{\theta})| < \tilde{\varepsilon}\}$ and $\delta^* \triangleq 2\delta + \tilde{\delta} + \mathbb{P}(\tilde{\boldsymbol{\theta}}(t) \in Q)$.

As a result, with a probability of $1 - \delta^*$, the error bound in Eq. (25) for the proposed method is lower than that of the conventional method. $\qquad\square$

## D. Experimental Supplement

### D.1. Additional Experimental Results

#### D.1.1. THE ABLATION STUDY

To analyze the impact of each part in our framework, we conduct an ablation study on the non-linear oscillator and damped oscillator family data comparing four framework settings: (1) $\boldsymbol{\theta}$ are modeled as $\boldsymbol{\theta}(t)$, as in Hypernet; (2) $\boldsymbol{\theta}$ are further constrained as the solution of another ODE parameterized by a neural network, i.e., $\frac{d\boldsymbol{\theta}(t)}{dt} = \boldsymbol{q}(\boldsymbol{\theta}(t), t; \boldsymbol{\Phi})$, as in a normal coupled neural ODE framework (we refer this framework as CNODE); and (3) ours, which further models parameter dynamics using the Koopman model. To further highlight the advantage of intrinsic linear modeling, we also impose sparsity by setting $80\%$ of the parameters in $\boldsymbol{h}$ to zero. This ensures that the model's behavior is predominantly driven by the intrinsic linear modeling component.

From Tab. 3, the poor performance of CNODE compared to Hypernet suggests that strictly constraining $\boldsymbol{\theta}(t)$ as an ODE process may be overly restrictive and increase the training burden of the network, limiting flexibility and making the network more challenging to train. In contrast, using the Koopman model for $\boldsymbol{\theta}(t)$ enables the excellent performance, even with a sparse $\boldsymbol{h}$, demonstrating the effectiveness of our approach in capturing underlying dynamics efficiently.

#### D.1.2. ADDITIONAL RESULTS ABOUT NON-LINEAR OSCILLATOR

Tab. 4 presents the performance of various ODE solvers—`euler`, `rk4`, and `dopri5`—on the non-linear oscillator dataset, comparing our method with NODE. As is well known, `RK4` and `dopri5` are higher-order solvers, with `dopri5` using adaptive step sizes. Our approach demonstrates strong performance even with the simplest `euler` method.

We compare the computational efficiency of our framework and NODE by reporting both the average forward and backward propagation times over 200 training epochs in Tab. 5. Additionally, we report the total cumulative time spent on forward and backward propagation for each method from the beginning of training until convergence. The convergence criterion is

*Table 5.* Results of running time regarding average forward propagation time (01), average backward propagation time (02), total forward propagation time from the beginning of training until convergence (03), total backward propagation time from the beginning of training until convergence (04), and convergence epoch between KoNODE(ours) and NODE on the non-linear oscillator dataset.

| METHODS | 01(S) | 02(S) | 03(S) | 04(S) | CONVERGENCE EPOCH |
|---------|-------|-------|-------|-------|-------------------|
| NODE | 0.0147±0.0027 | 0.0588±0.0073 | 43.5089 | 171.9932 | 48 |
| KONODE | 0.0382±0.0053 | 0.1656±0.0190 | 20.6565 | 88.5394 | 10 |

based on monitoring the stability of the training loss over consecutive batches. Specifically, after each batch, the absolute difference between the current loss and the previous loss is computed. If this difference remains below a predefined threshold ($5 \times 10^{-4}$) for 3 consecutive batches, the training is considered converged. The training loss curves for our framework and NODE are also presented in Fig. 6, showing that our method converges faster than NODE.

From Tab. 5, we observe that although our method requires more time per iteration for both forward and backward propagation compared to NODE, its superior expressive power enables it to converge faster. As a result, the total forward and backward propagation times at convergence are lower for our method than for NODE. In addition, we note that in our formulation, $\boldsymbol{w}(t)$ has an explicit solution, has an explicit solution, which is also one of its advantages.

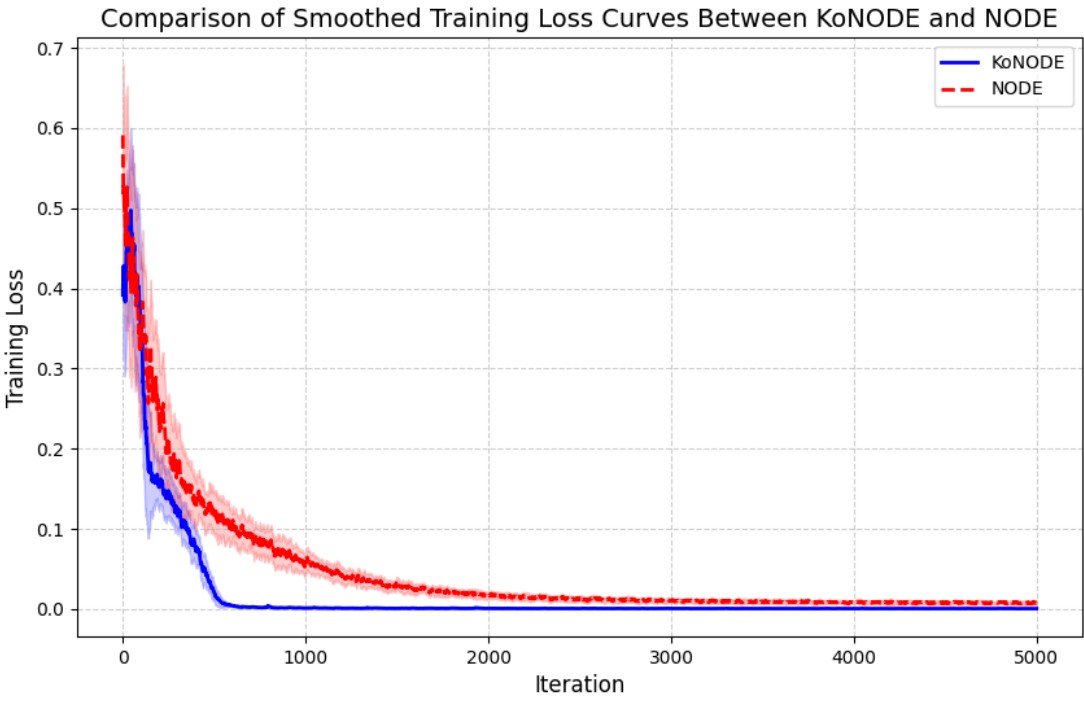

*Figure 6.* Comparison of smoothed training loss curves between KoNODE and NODE.

In Fig. 7, we visualize trajectory results on the nonlinear oscillator data, comparing with the top-performing Hypernet and ANODEV2.

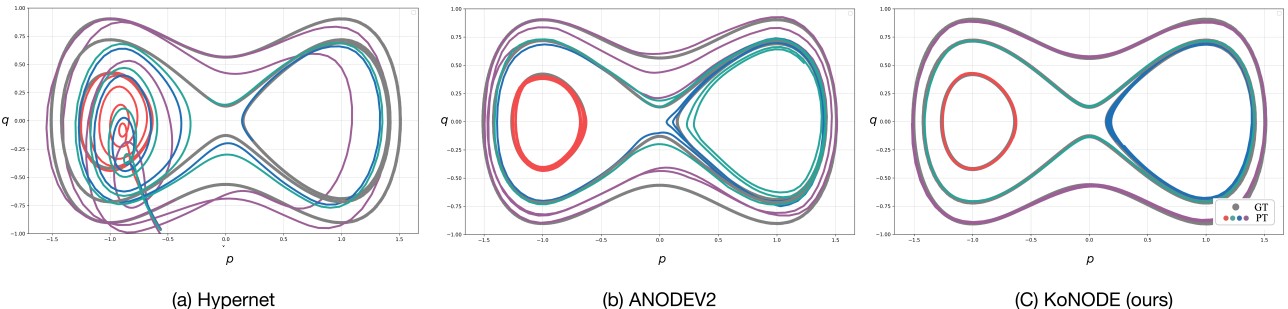

*Figure 7.* Trajectories on nonlinear oscillator data across Hypernet, ANODEV2, and KoNODE (ours).

### D.2. Details About Experiment in Sec. 5.1

#### D.2.1. IMPLEMENTATION DETAILS

All the experiments throughout the article were trained on a single GPU of model GTX 3090 Ti.

**Data Generation.** The dataset is created by solving an ODE

$$
\begin{bmatrix} \frac{dx}{dt} \\ \frac{dy}{dt} \end{bmatrix} = \boldsymbol{A}(t) \begin{bmatrix} x^3 \\ y^3 \end{bmatrix}, \quad \begin{bmatrix} x_0 \\ y_0 \end{bmatrix} = \begin{bmatrix} 2 \\ 0 \end{bmatrix}, \tag{26}
$$

where $\boldsymbol{A}(t) = \begin{bmatrix} -0.1 + 0.5\sin(t) & 2.0 + \cos(t) \\ -2.0 + 0.5\cos(2t) & -0.1 - 0.5\sin(t) \end{bmatrix}$.

Following (Chen et al., 2018), the initial state $\boldsymbol{x}_0 = [2, 0]^\top$ is used as the starting condition. The time range for trajectory generation is set as $t \in [0, 25]$, and the sampling resolution is set to 1000 time steps. The reference trajectory is computed using `dopri5` from `torchdiffeq`.

**Training Procedure.** In this experiment, we aim to single trajectory fitting task. During training, each training batch consists of a randomly sampled set of initial conditions from the reference trajectory. A batch contains 10 consecutive time steps, that is, the model predicts the remaining 9 time points regarding initial points during training. We set the batch size to 5. The optimizer is `Adam`, with an initial learning rate of 0.01. Learning rate scheduling following an exponential decay. Decay occurs at iterations 500, 4000, with scaling factors 1.0, 0.1, and 0.01. We train runs for 5000 iterations. The loss function is MAE (Mean Absolute Error) between predicted and ground-truth trajectories. After training, the trained model is evaluated on the full test trajectory to measure the trajectory modeling performance.

#### D.2.2. ARCHITECTURE AND HYPERPARAMETER DETAILS

We adopt `dopri5` as ODE solver, with `atol=`$10^{-7}$,`rtol=`$10^{-9}$ in this experiment to learn the ODE system. Gradient computation is performed using the adjoint sensitivity method. The network architecture for $\boldsymbol{h}$ is MLP. The dimension of $\boldsymbol{w}$ is 2. $\boldsymbol{w}_0$ is randomly sampled from Gaussian distribution. The differential function network $\boldsymbol{f}$ architecture is MLP, consisting of three layers with 300 neurons. We adopt the same architecture, training procedure, hyperparameters, and ODE solver for KoNODE and NODE.

### D.3. Details about Experiment in Sec. 5.2

#### D.3.1. COMPARISION METHODS

To evaluate the effectiveness of our framework, we compare it against several existing neural ODE-based models, each representing different design choices and extensions of the standard neural ODE (NODE). Below, we provide a brief overview of each comparison model.

**Vanilla NODE** (Chen et al., 2018). Standard Neural ODE $\frac{d\boldsymbol{x}(t)}{dt} = f(\boldsymbol{x}(t), t; \boldsymbol{\theta})$, where $\boldsymbol{f}$ is chosen to be a feedforward

fully connected network with four hidden layers of size 80 just as our framework.

**Hypernet** (Ha et al., 2016). Hypernet extends the vanilla NODE by allowing the parameters of the ODE function to evolve over time, introducing additional flexibility in modeling complex dynamics. Instead of using a static parameter set $\boldsymbol{\theta}$, the model employs a hyper network that generates time-dependent parameters:

$$\frac{d\boldsymbol{x}(t)}{dt} = f(\boldsymbol{x}(t), t; \boldsymbol{\theta}(t)), \tag{27}$$

where $\boldsymbol{\theta}(t)$ is dynamically generated based on an auxiliary network, namely, hypernetwork, which is implemented as a multilayer perceptron (MLP). The function $\boldsymbol{f}$ governing the ODE is chosen to be a fully connected feedforward network, consistent with our framework.

**ANODEV2** (Zhang et al., 2019). ANODEV2 introduces a novel coupled neural ODE framework in which the parameters of the ODE function, $\boldsymbol{\theta}(t)$, are not fixed but instead evolve over time. This makes ANODEV2 the first method to integrate parameter dynamics directly into the Neural ODE formulation. Additionally, ANODEV2 employs the Gershgorin function to model $\boldsymbol{\theta}(t)$ further to ensure numerical stability and interpretability. In our experiments, the function $\boldsymbol{f}$ governing the ODE is chosen to be a fully connected feedforward network as our framework.

**Heavy Ball Neural ODE (HBNODE)** (Xia et al., 2021). HBNODE incorporates second-order dynamics inspired by momentum-based optimization. Instead of modeling first-order differential equations, HBNODE considers a second-order formulation:

$$\frac{d^2\boldsymbol{x}(t)}{dt^2} + \gamma \frac{d\boldsymbol{x}(t)}{dt} + \nabla_{\boldsymbol{x}} f(\boldsymbol{x}(t)) = 0, \tag{28}$$

where $\gamma$ represents a damping term. This design mimics heavy-ball acceleration techniques in optimization and enhances the stability and efficiency of learning in NODE-based systems.

**LEADS** (Yin et al., 2021). LEADS is a framework for learning dynamical systems that generalize across different environments by capturing both shared and environment-specific dynamics. Instead of training a single global model or separate models per environment, LEADS learns a common model for shared dynamics and augments it with components tailored to each environment. This approach reduces sample complexity and improves generalization to both seen and unseen environments, with theoretical guarantees and strong empirical performance on linear and nonlinear systems.

**CoDA** (Kirchmeyer et al., 2022). CoDA is a framework designed to enable fast and efficient generalization of dynamical system models to new, unseen environments. It addresses distributional shifts by conditioning a shared dynamics model on environment-specific context vectors, inferred from data using a hypernetwork. This approach constrains the hypothesis space, allowing rapid adaptation and improved generalization with limited data. CoDA is theoretically grounded and achieves state-of-the-art results across diverse nonlinear dynamical systems, with the ability to infer system-specific parameters from context with minimal supervision.

**ODE via Invertible Neural Networks (OINNs)** (Zhi et al., 2022). OINNs leverage the invertibility property of normalizing flows to construct a NODE formulation that preserves bijective mappings between states. The system is governed by:

$$\frac{d\boldsymbol{x}(t)}{dt} = g(f(\boldsymbol{x}(t))), \tag{29}$$

where $\boldsymbol{g}(\cdot)$ is an invertible transformation. This approach allows for more stable training and enables efficient computation of the inverse mapping.

**Modulated Neural ODE (MoNODE)** (Auzina et al., 2024). MoNODE introduces external control mechanisms into the NODE framework, allowing for dynamic modulation of the ODE function. This design is particularly useful in scenarios where external factors influence system dynamics. The governing equation takes the form:

$$\frac{d\boldsymbol{x}(t)}{dt} = f(\boldsymbol{x}(t), u(t)), \tag{30}$$

where $\boldsymbol{u}(t)$ is an external modulation signal.

As described above, Hypernet and ANODEV2 explicitly incorporate time-dependent ODE parameters, enhancing adaptability in dynamic environments. We implement NODE, Hypernet, and ANODEV2 using the `torchdiffeq` library, with their detailed architectures described above. For HBNODE, OINNs, and MoNODE, we leverage publicly available code to implement them.

D.3.2. TRAINING SETUP

The following reports the training setup of two experiments in Sec. 5.2. All models are implemented using PyTorch. We employ the Adam optimizer for training with an initial learning rate of $1 \times 10^{-4}$ , which follows an exponential decay schedule with a decay rate of 0.95. The batch size is set to 10, and the model is trained for 200 epochs. For solving the ordinary differential equation (ODE), we utilize the `euler` solver with `atol` $= 10^{-7}$, and `rtol` $= 10^{-9}$. Gradient computation is performed using the adjoint sensitivity method, which allows for memory-efficient backpropagation through ODE solvers. The loss function during training is MSE between the predicted trajectory and the ground truth.

D.3.3. IMPLEMENTATION DETAILS ABOUT NON-LINEAR OSCILLATOR

**System details.** In this section, we consider an undamped and unforced nonlinear oscillator. We set $\alpha = -1, \beta = 1, \gamma = 0$. The system equations are as follows:

$$\frac{dq}{dt} = p, \quad \frac{dp}{dt} = q - q^3,$$

where $q$ is the position, $p$ is the momentum, and the parameters are $\alpha = -1$, $\beta = 1$, and $\gamma = 0$. We give the characteristics analysis of this system in the following.

(1) No Damping Term ($\gamma = 0$): The damping term typically causes the system's energy to dissipate gradually, leading to a decrease in oscillation amplitude over time. However, in this system, due to the absence of a damping term, the system will continue to oscillate without any energy dissipation.

(2) Nonlinear Restoring Force: In this system, the restoring force consists of two parts: a linear term $-\alpha q$ and a nonlinear term $-\beta q^3$.

- **Linear Term** ($q$): This is the standard spring restoring force term, describing the basic elastic behavior of the system.

- **Nonlinear Term** ($-q^3$): This is the key term in the Duffing equation, reflecting the nonlinear characteristics of the system. Typically, this term causes the restoring force to depend not only on the linear displacement but also on the cubic power of the displacement.

(3) System's Motion Behavior: Due to the nonlinear restoring force, the system's motion can exhibit various complex behaviors, such as periodic motion, chaotic motion, or quasiperiodic oscillations. The nonlinear term usually causes the system to deviate from a simple stable trajectory and may lead to different phase trajectories and periods. In particular, when $\beta = 1 > 0$, the system exhibits sensitivity to initial conditions.

**Data generation.** We first consider an undamped and unforced nonlinear oscillator, setting the parameters as $\alpha = -1, \beta = 1, \gamma = 0$. The training set includes 600 trajectories, each starting from an initial state uniformly sampled from an annular region in $[0.2, 1]$. Each trajectory spans 30 time steps with a fixed step size of 0.1, and Gaussian noise $0.01n, n \sim \mathcal{N}(0, 1)$ is added to simulate real-world data. The test set consists of 200 trajectories from the same system, with identical initial state sampling but extending to 300 time steps. This setup enables us to assess both the ability of long-term prediction and generalization. All trajectories are generated using `dopri5`. We visualize the trajectory data of four types under our parameter settings as shown in Fig. 8.

**Model architecture and hyperparameter.** The neural ODE function is parameterized as a fully connected network with four hidden layers, each containing 80 neurons with ReLU activation. $\boldsymbol{w}_0$ is randomly sampled from Gaussian distribution. The dimension of $\boldsymbol{w}$ is 10.

D.3.4. IMPLEMENTATION DETAILS ABOUT THE DUFFING OSCILLATORS

**System details.** The system family represents a **weakly damped linear oscillator**, governed by the equations:

$$\frac{d\mathbf{q}}{dt} = \mathbf{p}, \quad \frac{d\mathbf{p}}{dt} = -\alpha\mathbf{q} - \gamma\mathbf{p} \tag{31}$$

which can be rewritten as a second-order differential equation:

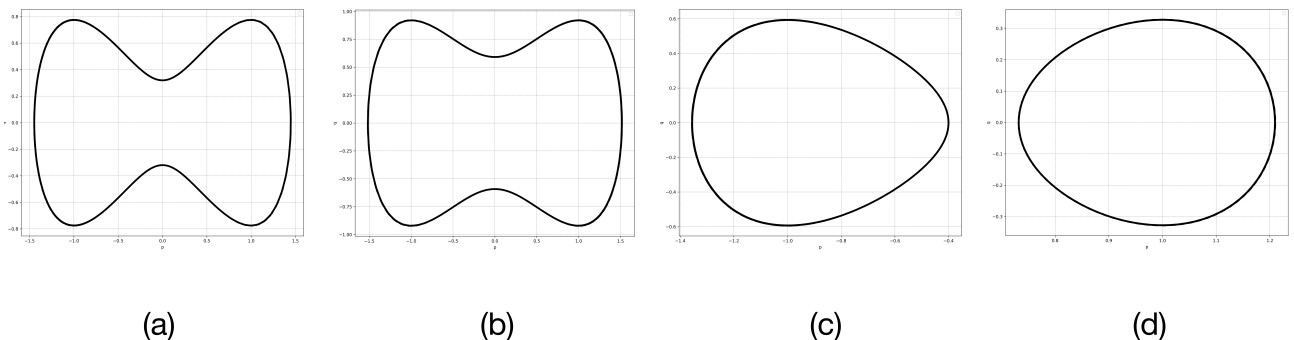

*Figure 8.* Four types trajectory data generated from non-linear oscillator system.

$$\frac{d^2\mathbf{q}}{dt^2} + \gamma\frac{d\mathbf{q}}{dt} + \alpha\mathbf{q} = 0. \tag{32}$$

Key characteristics:

- **Underdamped oscillation**: The system exhibits oscillatory motion with gradual amplitude decay due to small damping.

- **Frequency dependence on** $\alpha$: The oscillation frequency is approximately in the range $[0.7, 1.0]$.

- **Exponential decay controlled by** $\gamma$: Weak damping leads to slow energy dissipation.

- **Parameter sensitivity**: Small variations in $\alpha$ and $\gamma$ induce minor shifts in oscillation frequency and decay rate, making the system useful for studying stability under small perturbations.

**Data generation.** We consider a weakly damped linear oscillator, setting the parameters as $\alpha \in [0.9, 1.1]$, $\beta = 0$, and $\gamma \in [0.09, 0.11]$. The training set consists of 300 trajectories, each initialized from a random state uniformly sampled within an annular region $r \in [0.5, 1]$. Each trajectory spans 100 time steps with a fixed step size of 0.1, and Gaussian noise of magnitude $0.01n$, $n \sim \mathcal{N}(0, 1)$ is added to simulate real-world variability. The test sets contain **100** trajectories, following the same initial state sampling process as the training set but extending to **200** time steps, allowing for the evaluation of long-term predictive capabilities. All trajectories are numerically integrated using the Dopri5 solver (a fourth-order Runge-Kutta adaptive method). To visualize the system's behavior, we randomly sample and plot several trajectory examples in the phase space (momentum $p$ vs. position $q$), as shown in Fig. 9. This dataset serves as a benchmark for evaluating the ability of learning models to capture damped oscillatory dynamics and generalize across different time horizons.

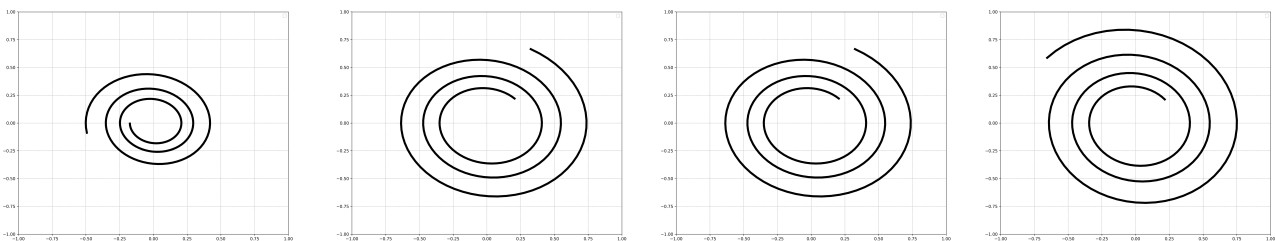

*Figure 9.* Several trajectory examples in the phase space of duffing oscillators family data.

**Model architecture and hyperparameter.** The Neural ODE function is parameterized as a fully connected network with four hidden layers, each containing 500 neurons with ReLU activation. $\boldsymbol{w}_0$ is initialized using method II, that is, using a

*Table 6.* Details about robot motion datasets regarding "S", "CUBE", and "C".

| DATA | NUMBER OF TRAJECTORIES | TIME POINTS | DATA DIMENSION |
|------|------------------------|-------------|----------------|
| "C" | 12 | 1000 | 3 |
| "CUBE" | 14 | 1000 | 3 |
| "S" | 7 | 1000 | 2 |

RNN, which processed the first 5 observed trajectory points to infer an initial latent state. Note that all of the comparison methods in this experiment use five observed trajectory points as inputs. The dimension of the $w$ is set to 100. $h$ is also the fully connected network.

### D.4. Details about Experiment in Sec. 5.4

**Training setup.** The training setup of Experiment IV is the same to Experiment II in App. D.3.2.

D.4.1. DATASET

**Dataset Description** The dataset, proposed by Khansari-Zadeh & Billard (2011), consists of three types of trajectory data, all derived from real-world motion demonstrations, including:

(1) **Drawing "S" shapes.** This trajectory represents an object or a robot executing an "S"-shaped motion, which may be used for handwritten character recognition, path planning, or robot trajectory imitation learning.

(2) **Placing a cube on a shelf.** This task involves manipulating objects (e.g., a robotic arm grasping and placing an object), which is commonly used in robotic grasping and placement tasks to evaluate the precision and stability of robotic operations.

(3) **Drawing out a large "C".** This trajectory involves performing a large-scale "C"-shaped motion, which may be used for handwritten character recognition or path planning. It can also serve as a test case for trajectory generation models.

**Data generation** For preprocessing, we follow the approach of Zhi et al. (2022), where B-spline interpolation is applied to smooth the trajectories and standardize the number of time points to 1000. Detailed information about these three datasets is provided in Tab. 6 and the visual results are presented in Fig. 10.

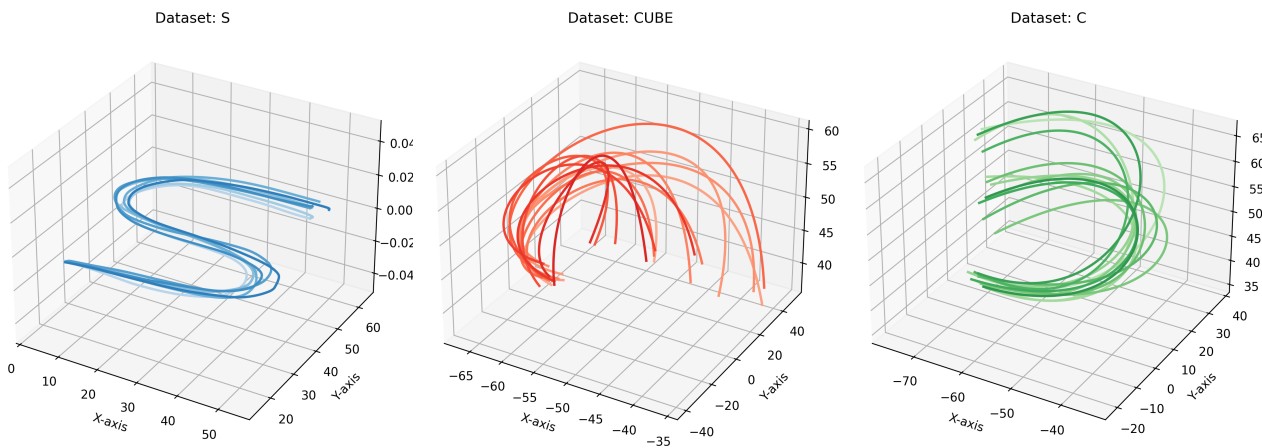

*Figure 10.* Trajectory example of robot motion datasets.

For the "S" dataset, 4 trajectories are used for training, while 1 trajectory is allocated for validation and 1 for testing. For the "CUBE" dataset, 10 trajectories are selected for training, with 1 trajectory for validation and 3 for testing. For the "C"

*Table 7.* Details about datasets in Experiment V regarding Type and Data Characteristics.

| DATASET | TYPE | DATA CHARACTERISTICS |
|---------|------|----------------------|
| CUP | AIR QUALITY | LONG-TERM DEPENDENCIES, NONLINEAR POLLUTANT DIFFUSION |
| WEATHER | WEATHER VARIABLE | SHORT-TERM DYNAMICS DOMINANT, NON-PERIODIC FLUCTUATIONS |
| ETTH1 | TRANSFORMER TEMPERATURE | LONG-TERM DEPENDENCIES, PERIODIC TRENDS |
| ETTH2 | TRANSFORMER TEMPERATURE | LONG-TERM DEPENDENCIES, COMPLEX PERIODICITY |
| ETTM1 | TRANSFORMER TEMPE. (MINUTE) | STRONG SHORT-TERM DYNAMICS, HIGH-FREQUENCY DATA |
| ETTM2 | TRANSFORMER TEMPE. (MINUTE) | HIGH-FREQUENCY DYNAMICS, STRONG SHORT-TERM FLUCTUATIONS |

*Table 8.* Datails about datasets in Experiment V regarding Frequency and Dimension.

| DATASET | FREQUENCY | DIMENSION |
|---------|-----------|-----------|
| CUP | 1 HOUR | 270 |
| WEATHER | 10 MIN | 21 |
| ETTH1 | 1 HOUR | 7 |
| ETTH2 | 1 HOUR | 7 |
| ETTM1 | 15 MIN | 7 |
| ETTM2 | 15 MIN | 7 |

dataset, 10 trajectories are designated for training, while 1 trajectory is assigned for validation and 1 for testing. During training, a random starting point is selected from the trajectories in the training set, and the subsequent 100 data points are extracted as training samples. The model's performance is then evaluated across the entire trajectory.

D.4.2. IMPLEMENTATION DETAILS ABOUT ROBOT MOTIONS: MODEL ARCHITECTURE AND HYPERPARAMETERS.

For dataset "S" and dataset "CUBE", the Neural ODE function is parameterized as a fully connected network with four hidden layers, each containing 200 neurons with ReLU activation, while for dataset "C" it contains 350 neurons. The initial observable $w_0$ is initialized using method II, that is, using a RNN, which processed the first 50 observed trajectory points to infer an initial latent state. Note that all of the comparison methods in this experiment use 50 observed trajectory points as inputs. The dimension of the $w$ is both set to 10. $h$ is also the fully connected network.

## D.5. Details about Experiment in Sec. 5.5

D.5.1. DATA DESCRIPTION

In this study, we use the following six datasets to evaluate the predictive performance of different models. These datasets cover multiple application domains, including air quality forecasting, weather prediction, and electricity load forecasting, each exhibiting distinct time series characteristics. The details of the datasets are summarized in Tabs. 7 and 8.

D.5.2. IMPLEMENTATION DETAILS

We give the details about model architecture and hyperparameters across different datasets in this experiment in Tab. 9. Both our method and the baseline models take 10 time steps as input and predict the subsequent 100 time steps.

D.5.3. DETAILS ABOUT COMPARISION METHODS

All the results of the comparison methods we reproduced are implemented based on the original paper or official code.

**DeepVAR**: A deep vector autoregression (VAR) model with variational inference, suitable for multivariate time series forecasting and probabilistic modeling.

**Autoformer**: Utilizes auto-correlation attention and trend-seasonality decomposition for improved long-term time series forecasting.

**PatchTST**: A Transformer-based model with patching mechanisms, effectively capturing local patterns and long-range dependencies.

*Table 9.* Details about model architecture and hyperparameters.

| MODEL SETTINGS | CUP | WEATHER | ETTH1 | ETTH2 | ETTM1 | ETTM2 |
|---|---|---|---|---|---|---|
| **ODE NETWORK** | | | | | | |
| NETWORK TYPE | MLP | MLP | MLP | MLP | MLP | MLP |
| HIDDEN STATE DIMENSION | 300 | 300 | 250 | 300 | 150 | 300 |
| HIDDEN LAYER | 4 | 4 | 5 | 4 | 4 | 5 |
| ACTIVATION | RELU | RELU | RELU | RELU | RELU | RELU |
| | | | W SETTINGS | | | |
| $w_0$ | II(RNN) | II(RNN) | II(RNN) | II(RNN) | II(RNN) | II(RNN) |
| DIMENSION | 100 | 100 | 100 | 100 | 10 | 100 |
| **TRAINING SETTINGS** | | | | | | |
| OPTIMIZER | ADAM | LEARNING RATE | $1e^{-3}$ | | | |
| LR SCHEDULER | EXP DECAY | DECAY RATE | 0.95 | | | |
| BATCH SIZE | 32 | EPOCH | 10 | | | |
| NUMERICAL SOLVER | EULER | BACKPROPAGATION | ADJOINT METHOD | | | |

**Koopa**: Employs Koopman operator theory for nonlinear dynamic system modeling, ideal for periodic and quasi-periodic time series.

**SST**: Combines state-space modeling (SSM) with Transformer architecture, excelling at both short-term dynamics and long-term dependencies.

