# OpenReview forum: "KoNODE: Koopman-Driven Neural Ordinary Differential Equations with Evolving Parameters for Time Series Analysis"
_ICML.cc/2025/Conference — ICML 2025 poster_

### Official Review · Reviewer_waH1 · 2025-02-18

**Overall Recommendation:** 4

**Summary:**

**Edit**: Having read through the rebuttals and the other reviews, I am satisfied with the paper and have increased my score from 3 (weak accept) to 4 (accept). I am grateful to the authors for taking the time to respond, and have left remaining specific thoughts in the Rebuttal Comment.

**Original Review**:

This paper introduces Koopman-Driven Neural Ordinary Differential Equations (KoNODEs). The motivation is that Neural ODEs given by:

$\frac{d\mathbf{x}}{dt} = f(\mathbf{x}, t, \theta)$

Can benefit from making the parameters time-dependent:

$\frac{d\mathbf{x}}{dt} = f(\mathbf{x}, t, \theta(t))$.

One example being that the underlying dynamics might remain unchanged, but the parameters of the dynamics do, for example in engineering the wear and tear of instruments can change the observed dynamics, even though the underlying physics remains unchanged.

An existing solution is ANODE-V2 that models $\frac{d\mathbf{x}}{dt} = f(\mathbf{x}, t, \theta(t))$ using a neural network and $\frac{d\theta}{dt} = g(\theta, t, \phi)$, so the network parameters are given by another ODE.

This work adds further modeling restrictions to the ODE with the aim of modeling more complex behaviors, for longer time series. This is achieved by using Koopman theory, which at a high level says that non-linear dynamics can be achieved by using linear dynamics in a different space. This paper explores how this can be achieved in practice for Neural ODEs, provides an implementation, theoretical analysis and experimental evidence showing modeling Neural ODEs using Koopman operators is an effective way to restrict the dynamics of the parameters to actually improve the model.

**Claims And Evidence:**

The claims presented are supported by experimental evidence. However, there are some relevant baselines that would improve the strength of the evidence. For example ODE-RNN or Latent ODE (https://arxiv.org/abs/1907.03907), or Neural Processes (https://arxiv.org/abs/1807.01622) as another method. These are common baselines that would improve the experimental results.

I am also concerned with Figure 3, which shows vanilla NODE not performing on the spiral synthetic dataset. Why is this the case since it does perform in the original paper?

**Essential References Not Discussed:**

There are none as far as I can tell.

**Experimental Designs Or Analyses:**

The experiments and analyses associated are sound. But as mentioned would be improved with stronger ODE baselines (e.g. ODE-RNN, Latent ODE) and stronger baselines for generalising to new dynamics (e.g. Neural Processes).

**Methods And Evaluation Criteria:**

The experiments carried out make sense for the application. The datasets are a good mix of synthetic to demonstrate key points and real to show that the method works on real data.

**Other Comments Or Suggestions:**

This is a list of contained writing changes I would make. They are suggestions so are not required:

- Line 17 Right: "The Ordinary Differential Equations" should be just "Ordinary Differential Equations"
- Line 19 Right: "of ODEs" should be "of the ODE"
- Line 38-39 Right: "encodes the deeper underlying dynamical principles of the evolution (namely deep-level information) inherently" should be "inherently encodes the underlying dynamics of the evolution (namely deep-level information)"
- Line 59 Left: "Based on the insight" should be "Base on this insight"
- Line 80-81 Left: "and ultimately governed by a deeper-leveled, intrinsic linear core - the Koopman linear dynamics" should be "and ultimately governed by the underlying Koopman linear dynamics"
- Line 66 Right: "Applying the Koopman Theory" should be just "Applying Koopman Theory"
- Line 94-96 Right: ", rather than propagating activations through discrete layers as in recurrent or deep networks. Neural ODEs employ numerical solvers" should be ". Rather than propagating activations through discrete layers as in recurrent or deep networks, Neural ODEs employ numerical solvers"
- Line 124 Left: "The Koopman Theory" should be "Koopman Theory"
- Line 124-125 Left "dynamic system analysis" should be "dynamical system analysis"
- Line 141 Left: "commonly finitely approximated" can be just "commonly approximated" since the phrase "finite number" is used later in the sentence
- Line 144 Left "The Koopman operator are" should be "The Koopman operator is" or "The Koopman operators are"
- Line 112 Right: "within the NODE setups" should be "within the NODE setup"
- Line 117-120 Right: "Besides, we assume the trajectories of the intrinsic linear dyanmics are represented as w(t)" should be "We let the intrinsic linear dynamics be represented as w(t)"
- Line 126-144 Right: Describing the three hierarchical levels should refer to the "dynamics" not the "dynamic"
- Line 158 Right: The sentence "We model the h by the neural network due to its flexibility" is not needed, but should be "We model h using a neural network due to its flexibility"
- Line 169 Left: "in replace of the conjugated imaginary eigenvalues of A" should be "that represent the imaginary eigenvalues of A"
- Figure 2 caption: Should refer to the "dynamics" not the "dynamic"
- Figure 2 caption: "over altogether N time points" should be just "over N time points"
- Section 4 description: "just as the red line" and "just as the green line" should be "shown by the red line" and "shown by the green line"
- Line 287 Left: "At last, we get the loss between the predicted trajectory and the true trajectory", should be "Using the true trajectory and the predicted trajectory, we can calculate the loss"
- Line 295-300 Left: "We serve w0 as an encoding" should be "w0 is an encoding" and "Learnable setting when given sequence as input" should be "Learnable setting when given a sequence as input"
- Line 375 Left: "Preformance metric" should be "Performance metric"
- Line 405 Left: "should inherently low-dimensional" should be "should inherently be low-dimensional"
- Lines 433-439 Left: When using quotation marks in Latex they can be properly displayed by changing "S" to ``S''
- Line 420 Right: "ETTh1" should be "ETTH1"
- Line 427 Right: "and the Koopman modeling" should be "and Koopman modeling"

**Other Strengths And Weaknesses:**

Stengths:

- The idea is robust and theoretically motivated
- The datasets are diverse and relevant
- The experiments support the claims

Weaknesses:

- The writing needs some work, see the Other Comments or Suggestions section of the review for specific cases. The description of KoNODE at points is overly complicated, or unnecessary. For example, at the beginning of page 4, one possible $D$ matrix is described in detail, before describing a second $D$ matrix with $2\times2$ block diagonals, the first version is never used and therefore is not necessary to describe. Section 3.2.2 is another section that overly complicates the description, where the point is to say that the adjoint sensitivity method is used to calculate gradients, which is already described in detail in the original Neural ODE paper

**Questions For Authors:**

1. Is it possible to represent KoNODE as another larger first order ODE:

$z = [x, \theta, w]$, $\frac{dz}{dt} = [f(x, t, \theta), \frac{\partial h}{\partial w}Aw, Aw]$, $z\_0 = [x\_0, h(w\_0), w\_0]$

If this is the case, it might be easier to write $\frac{\partial h}{\partial w}Aw$ as $h\_1(w)$ and $h(w\_0)$ as $h\_2(w\_0)$. That way the whole system can be solved together in one ODE solve, which will likely reduce the computation required per solve. This also makes the adjoint sensitivity derivation trivial since the standard adjoint sensitivity can be applied to the extended ODE system.

2. In Figure 4C, why does error increase as the dimension of $w$ increases. Is it hard for KoNODE to learn redundancy? Or is it encouraged to use all of its dimensions to learn the underlying Koopman dynamics? The theoretical bound presented says that the error decreases with more dimensions, why is this not the case? Does the model overfit when it has more capacity, i.e. $N$ is not large enough, so $r$ is negative in Theorem 3.2 so the error bound does not decrease with $m$?

3. Can you rephrase your argument around Table 4, the argument is that the Euler method can benefit from KoNODEs, with more accurate dynamics. However this does not make sense to me. Firstly Table 4 shows that the Euler method is also better using NODE, so it does not empirically hold. Additionally, RK4 and DOPRI5 should be more accurate than the Euler method during the ODE solve, so should also benefit from improved modeling of the dynamics.

Based on the answers to these questions and the others mentioned I will increase my evaluation.

**Relation To Broader Scientific Literature:**

The key contributions are framed appropriately in relation to the scientific literature.

**Theoretical Claims:**

The theoretical claims as far as I can tell are correct. I have not checked the proofs in the appendix in significant detail. However, I am confused by Line 254 on the left:

$-\text{max}\_{m < i \leq N}\log\_i \xi\_i - 1$

What does it mean for a logarithm to be subscripted? Previously it is given that $\xi = [\langle g, u\_1 \rangle, \langle g, u\_2 \rangle ... \langle g, u\_m \rangle]$, so how can $i$ be larger than $m$ and $\leq N$? Especially given that $n$ is the amount of data, but $\xi$ seems to be the dimension $m$ of the Koopman space?

---

> ### Author Rebuttal · Authors · 2025-04-01
>
> Thank you for your deep understanding of our method and insightful suggestions.
>
> **[1. comparison to baselines]**
>
> We have included the baselines mentioned, Latent ODE (ODE enc) and Neural Processes, in `supplementary_table_reviewers_Snqd_waH1.pdf`, https://anonymous.4open.science/r/KoNODE-D8F2/. We didn't compare ODE-RNN due to its official code not supporting prediction tasks.
>
> ---
>
> **[2. failure on spiral synthetic dataset]**
>
> Lines 283-285 to the right clarified that the spiral dataset we were using is obtained by $[\frac{dx}{dt}\ \frac{dy}{dt}]^⊤= A^⊤(t)[x^3\ y^3]^⊤$ where $A(t)=\begin{bmatrix}-0.1+0.5\sin t&2.0+\cos t\\\\-2.0+0.5\cos 2t&-0.1-0.5\sin t\end{bmatrix}$, which is more complicated than the one used in vanilla NODE, i.e., $ A(t) = \begin{bmatrix}-0.1&2.0\\\\-2.0&-0.1\end{bmatrix}$. We did successfully reproduce fittings for the spiral proposed in both ours and NODE yet the modified data is more complicated so would demonstrate the limitation of the conventional model.
>
> ---
>
> **[3. $\xi_i$ in line 254]**
>
> Sorry for the negligence in the notation. We will clarify in the revision that $\xi=[\xi_1\ \xi_2\ \cdots\ \xi_m]^⊤$ while the coordinates $\xi_i≜\langle g, u_i\rangle$ even for $i>m$ as the true Koopman space has an infinite number of bases $\left\\{u_i\right\\}_{i=1}^\infty$ (though no more than $N$-dimensions due to the data-driven scenario).
>
> ---
>
> **[W1. writing]**
>
> Thank you for the valuable suggestions. We will modify accordingly (including the minor fixes and the simplifications of the KoNODE description and Section 3.2.2).
>
> ---
>
> **[Q1. larger ODE expression]**
>
> It is a promising insight that will definitely be easier to write. However, this pipeline sacrifices the consistency of function $h$ and does not significantly reduce the computational cost (per solve). The reasons are: (1) In the forward integration, both pipelines should go through function $f$, compute $Aw$, and another network being either $h$ or $h_1$; (2) In the adjoint method, both pipelines require the gradient through network $h$ (or $h_1$) which requires a backpropagation through them; (3) The two networks, $h$ and $h_1$, both require the potential to map from the $w$-space to the $θ$-space, hence the sizes of the networks do not differ much.
>
> ---
>
> **[Q2. Fig. 4C]**
>
> Thank you for your deep understanding of our method. Fig. 4(c) shows a curve dropping in $[2, 10]$ and rising if $m > 10$ with a tolerable relative perturbation. Experimentally, more weights are introduced in networks, which may add extra uncertainty and instability during training when $m$ is large and [W2 & Q2. selection of Koopman dimension], *Response to Reviewer iauo* explains the reasons why the practical dimension is not that stable. Theoretically, the error bound in Thm. 3.4, Manuscript is majorly dominated by two terms, one with a factor of $\frac{m^2}{\sqrt N}$ in coefficient $p$ and the other with the order of $m^{-\frac r3}$, which does indicate the instability when $m$ is too large and may infer overfits. On the other hand, the order $r$ is negative only when $m$ is too small that the first $m$ bases $\\{u_i\\}_{i=1}^m$ fail to successfully model the true observable function $g$, hence this scenario only influences the left-hand side of the graph when $m$ is below the theoretical lower bound given in Thm. R.4, [W2 & Q2. selection of Koopman dimension], *Response to Reviewer iauo*.
>
> ---
>
> **[Q3. Tab. 4]**
>
> Sorry for the negligence of the phenomenon as the table aimed at showing the more significant improvement from the KoNODE framework using the Euler method. We will rephrase the interpretation in the revision. Regarding the lateral comparison, the integration by RK4 and DOPRI5 is undoubtedly more accurate yet the superiority of Eurler is caused by the different intrinsic objectives under the regularly sampled trajectory, i.e., the network for Euler estimates $\int_t^{t+Δt}f(x(t), t)\ dt$ for a fixed $Δt$ but the others for a mutable $Δt$. The fixed $Δt$ reduces the complexity of the target function and thus simplifies the task using Euler.

---

> > ### Comment · Reviewer_waH1 · 2025-04-03
> >
> > Thank you for taking the time to respond. I have read the other reviews and responses, and am generally happy with the paper now, I will raise my score accordingly. Please see remaining comments below:
> >
> > **Additional Baselines and ODE-RNN**: Thank you for running the additional baselines, as well as the additional baselines for Reviewer Snqd. The results are still good for KoNODE, which is reassuring. I'm not particularly convinced by the reason for not including ODE-RNN, the prediction can simply be a function of the hidden state $y(t) = g(h(t))$, where $h$ follows an ODE-RNN. This should not be too hard to implement. **However**, that being said I also recognise there is already a large variety of baselines, and another baseline is not necessary.
> >
> > **Spiral Dataset**: Thank you for the clarification, my apologies for not noticing this difference. The paper would be improved with clear communication around this point. That is, at the start of Section 5.1, saying something along the lines of "We adapt the fitting task proposed in Chen et al. 2018, by adding a time dependency to the matrix $A$. We add sines and cosines to produce the following new time-dependent matrix: ..."
> >
> > **$\xi\_i$ Line 254**: My apologies, but this still is not clear to me. Please can my error in the following reasoning be identified and explained:
> >
> > - Lines 140-144 Left: In practice the infinite dimensional Koopman space is finitely approximated by modeling a finite number of bases $\mathbf{u} = [u\_1, u\_2,..., u\_m]$, we have $m$ basis functions.
> > - Line 253 Left: $\mathbf{\xi}=[\langle g, u\_1 \rangle, ..., \langle g, u\_m \rangle]$. These are inner products between the function of interest in Theorem 3.2 $g$ and the finite Koopman bases $u\_i$. Each of these give scalars, and we have to have finitely many since we are doing a practical implementation. So there are $m$ components of $\bf{\xi}$ since there are $m$ basis functions.
> > - If $\mathbf{\xi}$ is $m$ dimensional then $i$ can only be between $1$ and $m$, not $m+1$ and $N$.
> > - This theorem is about the "Order-m Koopman Operator", so we have $m$ basis functions, $g$ is the function of interest, and we have $N$ data points. Why do we have new basis functions per data point? The true Koopman space has infinite, but this theorem is about the approximation, where there are finitely many?
> >
> > It is unclear what I am not understanding, and more broadly, as we have discussed, the paper will benefit significantly in the long-term by making the theory clearer.
> >
> > **Larger ODE**: This point has been misunderstood, I was really asking about the specific implementation, i.e. the code. It wasn't clear to me if the ODE for $w$ was solved first, and then the ODE for $x$, requiring two ODE solves in series and storing the $w(t)$ trajectory. I made a mistake and realised the combined ODE could be written as:
> >
> >  $\frac{d}{dt}[x, w] = [f(x, t, h(w, \psi)), Aw]$, $[x, w]\_{0} = [x\_0, w\_0]$
> >
> > Without an unnecessary ODE for $\theta$. Implementing this way requires only one ODE solve and not storing a dense trajectory $w(t)$. Having now looked through the supplementary code I realise this is exactly how KoNODE has been implemented, so I accept I got this wrong. In this case I'm curious what you think would happen if KoNODE is implemented differently:
> >
> >  $\frac{d}{dt}[x, w] = [f(x, t, w(t), \psi), Aw]$, $[x, w]\_{0} = [x\_0, w\_0]$
> >
> > That is, rather than using $h$ to generate parameters for $f$,  could $w$ be used as a concatenated input to $f$ which has fixed parameters $\psi$? The universal approximation theorem would make this valid, but how do you think it would affect learning, parameter efficiency (no $h$ network, not so many evolving $\theta$s), computation?
> >
> > **Figure 4C**: Thank you for your answer. It sounds like it is a case of overfitting am I correct? How would you suggest tuning $m$, are there good values to begin with relative to $N$ and the dimension of $x$?
> >
> > **Table 4**: This is not a major concern (it's in the Appendix), and I am still happy with the paper. But I'm still not convinced by this reasoning. No matter the ODE solver, they are predicting $x(t)$ for a fixed set $[t\_1, ... t\_f]$. All that should change theoretically is the accuracy of the solve. In practice of course this affects the gradients and therefore the training, because they produce different trajectories. But if gradient descent is run long enough they should reach similar results. Is it because Euler produces "simpler" trajectories, i.e. it will not focus on significantly complicated parts of the trajectory, and therefore despite being a worse solve it is more stable? Maybe training with Euler at first, then changing to a more accurate solver at test time or even part way through training could be better? This is very much a side point, and I know was not the point of the Table (which was to show that KoNODE is robust to the solver), I just think it is curious, and goes against what I've experienced between solvers.

---

> > > ### Author Response · Authors · 2025-04-07
> > >
> > > Thank you sincerely for raising the score and for the time spent. Your intelligent feedback has significantly enhanced the depth and clarity of our manuscript.
> > >
> > > **[1. Additional Baselines and ODE-RNN]** Thank you for your thoughtful suggestion. We agree that, theoretically, the ODE-RNN is capable of making predictions. However, we omitted it because the official code raises an error: "Exception: Extrapolation for ODE-RNN not implemented," which prevents a direct comparison. Due to the limited time of the rebuttal, we could not re-implement it from scratch, and we sincerely apologize for this.
> > >
> > > **[2. Spiral Dataset]** Thank you for the helpful suggestion and for pointing this out. We will modify the manuscript accordingly.
> > >
> > > **[3. $ξ_i$ in line 254]** We apologize for the confusion caused by the notation and thank you for the patient analysis. The first three reasonings are correct. For the fourth one, the basis functions for the true Koopman space is infinite, hence we are able to identify infinite number of $⟨g, u_i⟩$. The coordinate $ξ_i$ in line 254 is the only one in which $i$ can be greater than $m$, we will replace it with $⟨g, u_i⟩$ instead to avoid confusion.
> > >
> > > Regarding the questions in reasoning four, we do not have new basis functions per data point. The reason why $i\le N$ is that the space the observable function $g$ in has a maximal dimension of $N$. As Lemma R.3, Response to Reviewer iauo, defines a possible observable function for any $θ$ trajectory to model the dataset, these functions span an observable function space with a dimension no more than $N$. Consequently, $⟨g, u_i⟩=0, ∀ i>N$.
> > >
> > > **[4. larger ODE]** Sorry for the misunderstanding and thank you for the clarification. Your current interpretation of the KoNODE implementation is very precise.
> > >
> > > We are very much thrilled by the acute insights you possess. Regarding your proposed alternative formulation (Imp. II), we agree that this is a theoretically valid and feasible variant. This formulation resembles the idea behind ANODEs (https://arxiv.org/abs/1904.01681), where additional variables are appended to the state to enhance expressivity.
> > >
> > > While both implementations are in principle functionally equivalent under the universal approximation theorem, their motivation differ. Our design (Imp. I) is motivated by a desire to extracte deep-level dynamic structures through $w(t)$. In contrast, Imp. II treats $w(t)$ as a refined input to enhance the model’s capacity, rather than explicitly encoding the system’s intrinsic dynamics. Technically speaking, any output neurons link to $w(t)$ in Imp. I while they may disconnect to it and only be activated by $x(t)$ using Imp. II.
> > >
> > > The following discussed practical factors.
> > >
> > > (1) Learning Dynamics. When $w(t)$ is concatenated with $x(t)$, its influence on the dynamics must be learned implicitly through a fixed network $f$. Without architectural constraints, this may often lead to learned representations that are dominated by $x(t)$, with $w(t)$ playing a minor role, especially when $w(t)$ is low-dimensional.
> > >
> > > (2) Parameter and Computational Efficiency. Although Imp. II appears simpler, it places the full burden of modeling complex, time-varying interactions on a single network $f$, which may increase training time or convergence difficulty. As highlighted in [1], effectively modeling intricate dependencies from low-dimensional auxiliary inputs (like $w(t)$) often requires large networks. By separating the responsibilities—$h$ modeling latent evolution and $f$ focusing on observable dynamics—Imp. I achieves better parameter specialization.
> > >
> > > **[5. Figure 4C]** Yes, "overfitting" is indeed precise. A suggested selection criterion of $m$ is increasing it from $⌈\frac D{D-r}⌉$ where $D$ is the model size and $r<\min\{n, N\}$ is the rank of data (described thoroughly in Theorem R.4, Response to Reviewer iauo) until the performance does not improve. An empirical choice is 10 for simple dynamics in 2D space and 100 for high-dimensional ones.
> > >
> > > **[6. Table 4]** We believe the superiority of the Euler solver is caused by the technical reason that the regression w.r.t. low-dimensional inputs (such as $t$) tends to be more difficult for neural networks to model [1]. Specifically, the Euler method only models $f(x(t),t;θ(t))$ but high-order methods RK4 and DOPRI5 require the model of $f(x^*,t^*;θ(t))$ where $x^\*≠x(t^\*)$ which rely on more sensitive modeling of $t$, making these models suffer more from the intractability caused by the low-dimension of time. The reason why the correspondence of $t$ and $x(t)$ relies less on the modeling of $t$ is implied by the mapping $t_s(⋅)$ from state to time in Lemma R.3, Response to Reviewer iauo.
> > >
> > > As for the further questions,  (1) yes, we agree that the Euler solver may yield more stable results; (2) regarding switching solvers later, we believe this is interesting and worth exploring.
> > >
> > > [1] See link: https://link.springer.com/chapter/10.1007/978-3-030-47358-7_27

---

### Official Review · Reviewer_1pn6 · 2025-03-12

**Overall Recommendation:** 3

**Summary:**

An architecture based on neural ODE with time-varying parameters is proposed. The idea is to model the dynamics of the parameters with latent linear dynamics, which the authors motivate by referring to the Koopman operators. The superior prediction performance of the proposed method is empirically demonstrated.

### Update after rebuttal

Thanks for the clarification, the additional discussion will make the paper more convincing. I keep my originally positive score.

**Claims And Evidence:**

I found two points in the claims. One is about its practical utility, i.e., better performance in prediction. I think this aspect is nicely demonstrated with the empirical results.

The other point of the claims I found is about "understanding" of data or models. For example, the authors claim:

> We propose a three-level hierarchical architecture ... that deepens our understanding of system behavior and its governing laws.

> By leveraging Koopman operators, our framework could use spectral tools for system analysis ...

Although I do not deny such a possible utility of the method in general, these aspects do not seem supported by concrete empirical observations.

**Essential References Not Discussed:**

N/A

**Experimental Designs Or Analyses:**

As mentioned in the "Methods And Evaluation Criteria" section, the experiments make sense.

**Methods And Evaluation Criteria:**

The time-series prediction experiments seem valid.

**Other Comments Or Suggestions:**

N/A

**Other Strengths And Weaknesses:**

N/A

**Questions For Authors:**

I do not have major questions that strongly affect my evaluation.

**Relation To Broader Scientific Literature:**

Time-series prediction is widespread in any domain of science and engineering.

**Theoretical Claims:**

A bunch of theoretical claims are presented in Section 3.3 and the appendices. They may be okay, but I do not see how relevant they are to the proposed method. The presented theories are on the estimation error of the Koopman operators, but the proposed method consists of not only the Koopman part but also other components such as the map from $w$ to $\theta$. It would be nice if the authors could elaborate more on the motivation of the theoretical arguments: for example, the overall picture (purpose) of the analyses, key assumptions, and remaining gap to fully analyze the prediction errors of the proposed method, if any.

---

> ### Author Rebuttal · Authors · 2025-04-01
>
> Thank you for your insightful comments.
>
> **[1. empirical observation supports]**
>
> We have added empirical evidence and analysis to support the two quoted claims. We will include concrete empirical observations in the revision. Please refer to [W1. lack sufficient interpretability], *Response to Reviewer Snqd* for further details, and `spectrum_reviewers_Snqd_1pn6.pdf`, https://anonymous.4open.science/r/KoNODE-D8F2/ for the visualization.
>
> ---
>
> **[2. theoretical claims]**
>
> Thank you for the suggestion and we are sorry for the confusion. We will clarify two motivations in Section 3.3 in the revised manuscript, i.e., the presented error provides (1) a theoretical reference for the choice of dimension $m$, and (2) proof for the minority of errors caused by the additional Koopman module compared to conventional ODE models, apart from the potential improvement due to the advanced fitting ability.
> The theoretical claims are given under the assumption that the parameters $θ(t)$ indicate the running dynamics, which are intrinsically determined for a given trajectory $x(t)$. Consequently, an accurate model of $θ(t)$ directly leads to the accurate estimation of $x(t)$.
> Thm. R.6 below conducts a theoretical comparison of the overall prediction error between the proposed method and a conventional ODE based on the given theories to indicate the superiority of our method.
>
> **Definition R.5 (The Static Dynamic Model Condition)**
>
> The dynamic satisfies the condition if and only if for the function $g$ and any $\tildeε$, $∃\tildeδ>0$,
> $$
> ℙ(|g(θ^*) - g[\tildeθ(t)]|<\tildeε)\ge 1-\tildeδ,
> $$
> where $\tildeθ(t)$ is randomly drawn from the ideal trajectory defined in Eq. (1), Theorem R.4, [W2 & Q2. selection of Koopman dimension], *Response to Reviewer iauo* and $θ^*≜\underset{θ}{\text{argmin}}\int_t\left\Vert f(x(t), t;θ)-f(x(t), t;\tildeθ(t))\right\Vert$ is the LSE estimation of a static $θ$. The condition holds when the true dynamic behind the trajectory is static which ensures a good fit of the NODE.
>
> **Theorem R.6**
>
> During the modeling of trajectory $x(t)$, the estimation error upper bound of the proposed model is smaller than that of a conventional ODE method if the observable function $g$ does NOT satisfy *the static dynamic model condition*, i.e., for the bound
> $$
> \mathcal B[\hat{θ}(t)]≜\sup_{θ}\left\Vert\frac{∂f(x(t), t;θ)}{∂θ}\right\Vert⋅\Vert\hat{θ}(t) -\tilde{θ} (t)\Vert,\tag{2}
> $$
> there exists,
> $$
> ℙ(\mathcal B[{θ}^†(t)]\le\mathcal B[θ^*])\ge1-δ^*,
> $$
> where $θ^†(t)$ is the trajectory predicted by the proposed method while $θ^*$ consists of the static parameters in the conventional ODE method.
>
> *Proof:* We use the notations in Def. R.5. Apparently, Eq. (2) gives a bound of the estimation error where the "sup" exists as $f$ is Lipchitz continuous, and the bound is reached for linear mapping $f$.
>
> Note that $\hat{θ}(t)=θ^*$ in NODE and $\hat{θ}(t)=θ^†(t)≜h(\hat{w}(t);ψ)≈g^{-1}(\xi ^⊤\hat{w}(t))$ in the proposed Koopman model where $\hat{w}(t)$ satisfies the Koopman model of matrix $\hat{K}_N$. Thm. 3.4, Manuscript shows that
> $$
> ℙ(|\xi ^⊤\hat{w}(t+Δt) - g[\tilde{θ}(t+Δt)]|<ε^*⋅\Vert\xi\Vert⋅\Vert\hat{w}(t)\Vert )\ge 1-δ,
> $$
>
> with $ε^*$ representing the RHS . As the base $u_i(t)\in L^2(Θ)$ for observables $\hat{w}$, Lemma C.12, Appendices implies $ℙ(|\hat{w}_i(t)|\le\frac{1}{\sqrtδ} )\ge 1-δ$ if $\Vert u_i(t)\Vert _{L^2(Θ)} = 1$ and $\max|\xi _i|\le r(\mathcal K)$,
> $$
> \textstyleℙ(\Vert\xi\Vert⋅\Vert\hat{w}(t)\Vert\le\frac{m\sqrt m⋅r(\mathcal K)}{\sqrtδ})\ge 1-δ.
> $$
> Consequently, if we let
> $$
> \tildeε=\frac{2\sqrt3σm^{\frac{7}{2}}r^2(\mathcal K)}{\sqrt{Nδ(δ-2m^{-\frac{r}{3}})}⋅\min\\{1, r(\mathcal K)\\}-1}+o(m^{-\frac{r}{3}}),
> $$
> then $ℙ(|\xi ^⊤\hat{w}(t+Δt) - g[\tilde{θ}(t+Δt)]|<\tildeε)\ge 1-2δ$.
>
> On the other hand, for $g$ failed to satisfy the static dynamic model condition at error $\tildeε$ we have
> $$
> \begin{aligned}
> &ℙ(\Vertθ^†(t) -\tilde{θ}(t)\Vert\le\Vertθ^*-\tilde{θ}(t)\Vert ),\\\\
> = &ℙ(|g[θ^†(t)] - g[\tilde{θ}(t)] |\le|g(θ^*)-g[\tilde{θ}(t)]|)\\\\
> &-ℙ(|g[θ^†(t)] - g[\tilde{θ}(t)] |\le|g(θ^*)-g[\tilde{θ}(t)]|\text{ and }\tilde{θ}(t)\in Q),\\\\
> \ge &ℙ(|g[θ^†(t)] - g[\tilde{θ}(t)] |\le|g(θ^*)-g[\tilde{θ}(t)]|)-ℙ(\tilde{θ}(t)\in Q)\ge 1-δ^*,
> \end{aligned}
> $$
> where $Q≜\\{θ|~|g(θ^*)-g(θ)|<\tildeε\\}$ and $δ^*≜2δ+\tildeδ+ℙ(\tilde{θ}(t)\in Q)$.
>
> As a result, with a probability of $1-δ^*$, the error bound in Eq. (2) for the proposed method is lower than that of the conventional method.

---

### Official Review · Reviewer_Snqd · 2025-03-13

**Overall Recommendation:** 3

**Summary:**

This paper explores the challenge of modeling time series with NODEs. The authors propose a Koopman-driven framework named KoNODE that hierarchically encodes system dynamics through evolving ODE parameters and Koopman linear operators. Specifically, they introduce a three-level architecture—spanning observed state dynamics, parameter dynamics, and intrinsic Koopman linear dynamics—to disentangle surface-level behaviors from fundamental governing rules. Extensive experiments on synthetic and real-world datasets validate the effectiveness of the proposed approach.

## update after rebuttal
All my concerns have now been adequately addressed. Overall, this is a good work. However, considering the limited application scenarios, I will maintain my original score of "weak accept."

**Claims And Evidence:**

N/A

**Essential References Not Discussed:**

Pls refer to Weaknesses

**Experimental Designs Or Analyses:**

N/A

**Methods And Evaluation Criteria:**

N/A

**Other Comments Or Suggestions:**

N/A

**Other Strengths And Weaknesses:**

**Advantages:**

1. The proposed KoNODE framework is both theoretically grounded and empirically effective in capturing intrinsic system dynamics.
2. The experiments are very detailed and thorough, clearly demonstrating the effectiveness and efficiency of the proposed method.
3. The manuscript is well-structured and clearly written, facilitating easy comprehension for readers.

**Weakness:**

1. From my opinion, the first contribution which is claimed as “uncovers the fundamental principles driving system evolution” appears somewhat overstated, as the identified deepest dynamics still lack sufficient interpretability.
2. This framework is very generic, but its primary focus on time-evolving ODE parameters is not yet a common paradigm. Could the author compare their settings with other generalizable NODEs [1][2][3]?

---
**Reference:**

[1] LEADS: Learning Dynamical Systems that Generalize Across Environments. NeurIPS, 2021.

[2] Generalizing to New Physical Systems via Context-Informed Dynamics Model. ICML, 2022.

[3] Generalizing Graph ODE for Learning Complex System Dynamics across Environments. KDD, 2023.

**Questions For Authors:**

1. Line 178-179, the left half of the page. Could the author clarify why the assumption $A \equiv D$ makes sense?

2. In Section 3.3.2, Theorems 3.3 and 3.4 are formulated in Hilbert space rather than in the observation space of the data. Should $f$ and $h$ be maintained as characteristic functions to ensure these two theorems valid?

**Relation To Broader Scientific Literature:**

This work is beneficial for the development of AI for Science research.

**Theoretical Claims:**

N/A

---

> ### Author Rebuttal · Authors · 2025-04-01
>
> Thank you for your constructive feedback.
>
> **[W1. lack sufficient interpretability]**
>
> We apologize for the unclear description of "revealing the fundamental driving forces of system evolution" and will clarify it in the Introduction and Experiment sections in the revision.
>
> In our framework, the deepest dynamics, i.e., Koopman linear dynamics $dw/dt = Aw$, are obtained through Koopman operators. Fundamental evolution principles are revealed by the spectrum of the operators, which correspond element-wise to matrix $A$ and are given by $λ_j(K)=e^{α_jΔt}(\cosβ_jΔt±\rm i\it\sinβ_jΔt)$ in our framework. Specifically, the real part of the eigenvalues indicates the evolution speed, while the imaginary part corresponds to intrinsic frequencies and periodicity. Analyzing the elements of $A$ could help identify the system’s dominant driving modes and frequencies and offer insights into stability, periodicity, or other inherent characteristics, thus gaining a mathematically interpretable understanding of the system's evolution.
>
> To further demonstrate this, we provide a more detailed analysis and interpretation of the learned dynamics in Sec. 5.2 by visualizing the spectrum of approximate Koopman operator for the Oscillator dataset in `spectrum_reviewers_Snqd_1pn6.pdf`, https://anonymous.4open.science/r/KoNODE-D8F2/. First, the magnitudes of spectrum are approximately one for both two dynamic systems, indicating boundary stability, where the state remains on a stable trajectory without diverging or converging. Second, the dominant spectrum of the nonlinear oscillator (on the left in the figure) has similar imaginary parts (i.e., the frequency components are the same), indicating that the system exhibits clear periodicity. Modes with the same imaginary part have the same periodic components.
>
> ---
>
> **[W2. comparison to generalizable NODEs]**
>
> We have added the comparison between our framework with other generalizable NODEs, LEADS [1], and CoDA [2] in `supplementary_table_reviewers_Snqd_waH1.pdf`, https://anonymous.4open.science/r/KoNODE-D8F2/. Due to the lack of publicly available code for [3], we did not compare with it.
>
> ---
>
> **[Q1. A≡D]**
>
> As stated in lines 174-178 on the left, $\frac{dg(θ)}{dt}=Ag(θ)\iff\frac{d\tilde{g}(θ)}{dt}=PAP^{-1}\tilde{g}(θ)=D\tilde{g}(θ)$ hence we can assume $A\equiv D$ by modeling $\tilde{g}$ instead of $g$. We will clarify the definition of $P$ to avoid confusion. The assumption $A≡D$ is not only mathematically rigorous but, as addressed in [W1. lack sufficient interpretability], allows us to analyze the system's behavior directly by observing the elements of matrix $A$.
>
> ---
>
> **[Q2. Hilbert space]**
>
> The theorems are formulated in Hilbert spaces for generality, and this is a very mild assumption. The observation space of the data naturally satisfies this, as we only require the data space to be complete and have an inner product. The two functions are not necessarily characteristic functions. As clarified in line 156 on the right, function h is maintained as the inverse of the characteristic function, while f is the differential function in the NODE framework shown in Eq. (2).

---

### Official Review · Reviewer_iauo · 2025-03-13

**Overall Recommendation:** 4

**Summary:**

This paper proposes KoNODE, a hierarchical framework that integrates Koopman operators into Neural Ordinary Differential Equations (NODEs) to learn time-evolving parameters.The authors provide theoretical error bounds for the finite-dimensional approximation of the Koopman operator and show how the proposed method improves long-horizon prediction and generalization on both synthetic oscillators and real-world time series tasks.

## update after rebuttal
Given the fact, that authors addresed all the raised question and provided a new theorem with the proof, I raise the recommendation to Accept (4).

**Claims And Evidence:**

The paper states that modeling time-evolving ODE parameters via Koopman operators enables a deeper representation of underlying system dynamics, improves long-horizon forecasting, and generalizes better to unseen conditions. It has both theoretical evidence with finite-dimensional Koopman approximation error bound and experimental evidence with superior performance on the synthetic and real world data.

**Essential References Not Discussed:**

n/a

**Experimental Designs Or Analyses:**

I reviewed both the synthetic and real-world experiments in detail, and the overall design and selection of baselines are sound. There are two  minor concerns. The first one relates to the potential computational overhead introduced by the added Koopman operator layer. While the appendix does include some runtime analysis, it would be helpful to include a concise overview in the main text to clarify any changes in memory or runtime requirements. The second one is that while the authors provide some insight into Koopman operator dimensionality in synthetic settings, it remains unclear how these dimensional choices generalize to real-world scenarios or whether any systematic criterion (e.g., data-driven rank selection) could be used. Addressing these points would further strengthen the paper’s clarity and applicability. Aside from that, the experimental setup appears valid, with no major issues that would detract from the paper’s conclusions.

**Methods And Evaluation Criteria:**

The authors conducted several experiments on synthetic and real-world data. They selected a wide range of methods including classic Neural ODEs as well as the models with evolving parameters (ANODEV2), that makes the comparison reasonable.

**Other Comments Or Suggestions:**

n/a

**Other Strengths And Weaknesses:**

The paper’s primary strengths lie in its novel integration of Koopman operators with Neural ODEs, supported by comprehensive theoretical analysis and strong empirical results across both synthetic and real-world datasets. Meanwhile, potential weaknesses include limited discussion of computational overhead in the main text (despite some runtime data in the appendix), and a need for deeper examination of how the chosen Koopman dimensionality affects performance in real-world scenarios and whether a systematic selection method could be applied.

**Questions For Authors:**

1. How does KoNODE handle irregularly sampled data?
2. Is there a systematic way to select the dimension of the Koopman space m? Can any data-driven rank estimation be integrated?

**Relation To Broader Scientific Literature:**

The paper builds on Neural ODE literature (Chen et al. 2018) and prior Koopman approaches (Lusch et al. 2018, among others). It extends ANODEV2 by modeling theta through a linear Koopman system rather than a general ODE, thus adding interpretability and improved long-horizon stability.

**Theoretical Claims:**

There are several theoretical claims on the error bounds. Proofs, that are provided in the appendix, look reasonable, however, they may be a bit hard to follow, specifically with these equal signs with stars and asterisks, it is better to avoid such “nonlinearities” and be more consequent.

---

> ### Author Rebuttal · Authors · 2025-04-01
>
> Thank you for your valuable feedback.
>
> **[W1. changes in memory or runtime]**
>
> We will move the runtime analysis from the appendix to the main text to show our advantage in convergence rate and add a discussion on memory complexity. Regarding memory, we clarify that the memory of the proposed framework is $O(\max\\{n, m, h\\}⋅D)$ compared to $O(nD)$ of NODE for $n$, $m$, $h$ being the dimensions of $x(t)$, $θ(t)$, and the hidden layer respectively and $D$ the model size for the differential function $f$.
>
> ---
>
> **[W2 & Q2. selection of Koopman dimension]**
>
> Thank you for the concern about the choice of the Koopman dimension $m$. We provide theoretical lower and upper bounds for $m$. Although other factors may influence the best picks, the bounds provide references for the systematic criterion.
> Specifically, Thm. R.1 shows that $m\le r$ for the rank of trajectory space $r$ and Thm. R.4 shows that $m$ has a tight lower bound $⌈\frac{D}{D-r}⌉$ for second-order derivable $x(t)$, where $D$ is the dimension of $Θ$ (the model complexity of $f$) and $r$ is the rank of data. Note that the lower bound is commonly small for a network that is strong enough, yet the hyper-parameter may not be static in practice due to (1) the auxiliary dimensions caused by the designed form of the Koopman matrix, (2) the disturbance caused by the multiple choice of the dynamic $φ(θ(t), t)$ as $D\gg n$, and (3) the attempt to more accurate dynamic regression by multiple $\mathfrak F_t$ sets. The theorems are given as follows.
>
> **Theorem R.1 (The Theoretical Upper-Bound of $m$)**
>
> Inequality $m\le r$ holds if:
>
> (1) The data trajectories are approximately in an $r$-dimensional manifold $\mathcal M$, i.e.,
> $$
> ∀ε>0,∃~δ> 0,ℙ_{x(t)\sim\text{data}}[\Vert x(t)-x^\perp(t)\Vert <ε] > 1-δ,
> $$
> where $x^\perp≜\underset{y\in\mathcal M}{\text{argmin}}\Vert x-y\Vert $ is the projection of $x$ to the manifold.
>
> (2) Function $f$ satisfies the Lipchitz condition and is differentiable.
>
> *Proof:* Since $\mathcal M$ admits a bijective chart $ψ:\mathcal M\toℝ^r$, the map $\phi_t(θ(t)) = \frac{dψ(x^\perp(t))}{dx^\perp(t)}f(x^\perp(t), t;θ(t))$ is bijective. This, together with the uniqueness of $x(t)$ (by Picard–Lindelöf), implies that $θ(t)$ lies on an $r$-dimensional manifold. Hence, $m\le r$.
>
> **Definition R.2 (Frontier Manifold)**
>
> For a dynamic in Euclidean space, a fixed time $t_0$ and origin $θ^*$, the frontier manifold $\mathfrak F_t$ is the quotient space of the equivalent class decided by the trajectories, i.e., $\mathfrak F_0$ satisfies (1) $θ^*\in\mathfrak F_0$, and (2) the normal vector at $θ\in\mathfrak F_0$ is $φ(θ, t_0)$. Then, we define $\mathfrak F_t$ as the set of evolved $θ(t)$ with $θ(t_0)\in\mathfrak F_0$.
>
> **Lemma R.3**
>
> For $θ_0\in\mathfrak F_s$, if the trajectory $θ(t)$ follows the dynamic
> $$
> \frac{dθ(t)}{dt} =φ(θ(t), t),θ(t_0) =θ_0,
> $$
> a scalar observable function $g(θ)≜𝓒⋅\exp(\frac{λ}{D}\int\text{inv}[φ(θ, t_s(θ))]^⊤dθ)$ maps it into a Koopman space.
>
> *Proof:* Note that for $θ(t_0) =θ_0\in\mathfrak F_s$, $θ(t)$ is uniquely identified by the Picard-Lindelöf Theorem. Moreover, all elements in a frontier manifold $\mathfrak F_s$ are on different trajectories, which creates a bijection between $(θ_0, t)$ and $θ(t)$. Let $t_s(θ)$ be the mapping from $θ$ to time under the condition that $θ_0\in\mathfrak F_s$.
>
> Then, let $g(θ)$ be a scalar observable function,
> $$
> g(θ)≜𝓒⋅σ^{-1}\text{ where }σ=\exp\left(-\frac{λ}{D}\int\text{inv}[φ(θ, t_s(θ))]^⊤dθ\right),
> $$
> where $λ$ and $𝓒$ are constants and $\text{inv}[⋅]$ is the element-wise inverse of a vector. Therefore,
> $$
> \begin{aligned}
> \frac{dg(θ)}{dt} - k⋅g(θ) =&∇_{θ}g(θ)^⊤φ(θ, t) - k⋅g(θ)⋅φ^⊤(θ, t)⋅\text{inv}[φ(θ, t)]/D,\\\\
> =&φ^⊤(θ, t)⋅[∇_{θ}g(θ) -λ/D⋅g(θ)⋅\text{inv}[φ(θ, t)]], \\\\
> =&\frac{1}{σ}∇_{θ}[g(θ)⋅σ]= 0.
> \end{aligned}
> $$
> Hence, $g(θ)$ lies in the Koopman space.
>
> **Theorem R.4 (The Theoretical Lower-Bound of $m$)**
>
> To model the trajectory of $x(t)$ if it is second-order derivable, the dimension of $w(t)$ is ideally $⌈\frac{D}{D-r}⌉$ where $r$ is the rank of data.
>
> *Proof:* For a differentiable $f$, the dynamic $\frac{dθ(t)}{dt} =φ(θ(t), t)$ best models the trajectory $x(t)$ where
> $$
> φ(θ(t), t)≜\left[\frac{∂f}{∂θ(t)}\right]^{†}\left[\frac{d^2x(t)}{dt^2} -\frac{∂f}{∂t} -\frac{∂f}{∂x(t)}f(x(t), t;θ(t))\right],\tag{1}
> $$
> and $A^†$ being the pseudo-inverse of the Jacobian matrix.
>
> Lamma R.3 infers that for $θ_0\in\mathfrak F_s$, only one dimension of $w$ is needed. Thm. R.1 indicates that the gradient $φ(θ(t), t)$ lies in a space with the highest rank of $r$, thus the frontier manifold $\mathfrak F_s$ covers a dimension of at least $D-r$. To ensure a full cover of $Θ$-space in the initialization, an ideal dimension of $w$ is $⌈\frac{D}{D-r}⌉$.
>
> ---
>
> **[Q1. irregularly sampled data]**
>
> NODE-based methods model the system continuously, allowing them to handle irregularly sampled data by interpolating for intermediate time points.

---

### Decision · Program_Chairs · 2025-05-01

**Decision:**

Accept (poster)

**Comment:**

This paper combines Neural ODEs with Koopman operators in a hierarchical approach to model non-stationary dynamics. It provides theoretical error bounds and demonstrates long-term prediction and generalization on various synthetic and real-world examples. All referees agreed that this is a solid and novel contribution, with thorough evaluation and theoretical analysis, and all four recommended acceptance, with which I concur.

Besides what was stated in the reviews and related discussion, the authors may want to take the following additional remarks on board when drafting their revision:

- One other point that came up and should be clarified in the revision is that Theorem 3.2 is considering the infinite basis case. Also, one referee felt that the presentation of the theory could be further improved.

- Modeling non-stationary dynamics by assuming parameters to be time-dependent is a quite common approach in time series and dynamical systems modeling. I would like to encourage the authors to more thoroughly check the related lit. (maybe perplexity or just google), especially papers that discuss non-autonomous dynamical systems, and extend the coverage of related lit. which in my mind is a bit thin.

- Likewise, LEADS and CoDA are somewhat older frameworks. There is much more recent work for hierarchical modeling with time-dependent parameters and testing generalization across domains. I would recommend checking the most recent NeurIPS and ICLR publications on this.